# Two Calm Ends and the Wild Middle:
# A Geometric Picture of Memorization in Diffusion Models

Nick Dodson [* 1]  Xinyu Gao [* 1]  Qingsong Wang [* 2]  Yusu Wang [2]  Zhengchao Wan [1]

## Abstract

Diffusion models generate high-quality samples but can also memorize training data, raising serious privacy concerns. Understanding the mechanisms governing when memorization versus generalization occurs remains an active area of research. In particular, it is unclear where along the noise schedule memorization is induced, how data geometry influences it, and how phenomena at different noise scales interact. We introduce a geometric framework that partitions the noise schedule into three regimes based on the coverage properties of training data by Gaussian shells and the concentration behavior of the posterior, which we argue are two fundamental objects governing memorization and generalization in diffusion models. This perspective reveals that memorization risk is highly non-uniform across noise levels. We further identify a danger zone at medium noise levels where memorization is most pronounced. In contrast, both the small and large noise regimes resist memorization, but through fundamentally different mechanisms: small noise avoids memorization due to limited training coverage, while large noise exhibits low posterior concentration and admits a provably near linear Gaussian denoising behavior. For the medium noise regime, we identify geometric conditions through which we propose a geometry-informed targeted intervention that mitigates memorization.

## 1. Introduction

Diffusion models have achieved remarkable success in generative modeling, producing high-quality images but some-

times memorizing the training data, raising serious privacy and copyright concerns (Somepalli et al., 2023a; Carlini et al., 2023; Wen et al., 2024). This memorization risk is inherent to the training objective, as it admits a unique global minimum, namely the *empirical optimal* solution, which can only generate the training data (Gu et al., 2025). Consequently, it is both theoretically interesting and practically important to understand how and when a learned model can deviate from the empirical optimal to generalize beyond its training data.

Some recent works analyze how and where the learned model deviates from the empirical optimal. Vastola (2025) argues that the stochastic nature of diffusion model training helps prevent fitting the empirical optimal denoiser, and hence promoting generalization. Later, Bertrand et al. (2026) contests this view, showing that for a large range of training times the regression target is essentially unique while model still generalizes. Recently, Song et al. (2025) point out that supervised training in diffusion models is limited to small regions which fails to cover most parts of the inference trajectory, and argues that the generalization behavior of the model stems from extrapolation.

**Our work.** A key aspect to understand when models *generalize* is to understand when they *memorize*, which we focus on in this paper. To this end, we distinguish between *trajectory-level memorization*, which concerns whether full sampling trajectories collapse to the training set, and *per-noise-level memorization*, which characterizes memorizing denoiser behavior at a fixed noise scale. Empirically, we find that models exhibiting strong trajectory-level memorization may nonetheless behave benignly at many individual noise levels, indicating that memorization is not induced uniformly across the noise schedule (see Section 3 for more details).

This observation motivates our central question: *at which noise levels is memorization formed, and what geometric mechanisms govern its emergence?*

To address these questions, we analyze diffusion training through a geometric lens based on two quantities: *posterior weight concentration* and *Gaussian shell coverage* of the training data. This perspective naturally partitions the noise

*Equal contribution [1]Department of Mathematics, University of Missouri, Columbia, Missouri, USA [2]Halıcıoğlu Data Science Institute, University of California San Diego, La Jolla, California, USA. Correspondence to: Zhengchao Wan <zwan@missouri.edu>, Yusu Wang <yusuwang@ucsd.edu>.

*Proceedings of the 43rd International Conference on Machine Learning*, Seoul, South Korea. PMLR 306, 2026. Copyright 2026 by the author(s).

schedule into three regimes. At both small and large noise levels, the training lies on "calm ends" that are resistant to memorization, but for fundamentally different reasons: in the small noise regime, memorization is suppressed due to limited training coverage, while in the large noise regime due to weak posterior concentration and effectively linear denoising behavior. In contrast, a "danger zone" emerges at intermediate noise level – the "wild middle" – where these two effects align and memorization risk peaks sharply. Crucially, memorization formed in this regime can propagate along the inference trajectory and persist into the small noise stage, leading the model to memorize even when both ends of the schedule remain benign. This observation suggests that, to avoid memorization, it is most effective to control the behavior of diffusion models specifically within this intermediate "danger zone". By analyzing how coverage and posterior concentration interact, we can predict the location of this danger zone directly from the dataset. Experiments in Section 5 confirm this picture and show that selectively undertraining the intermediate regime can substantially mitigate memorization while preserving generation quality.

**More related work.** A related theoretical line studies sharp transition times in diffusion dynamics. From a statistical physics viewpoint, Biroli et al. (2024) characterize three regimes of the backward process, separated by speciation and collapse times; their analysis is explicit for high-dimensional Gaussian mixtures and is supported by numerical experiments on real datasets. Sclocchi et al. (2025) empirically show that diffusion dynamics on hierarchical data exhibits a phase transition at which high-level feature reconstruction changes abruptly, while low-level features vary more smoothly. Related critical-window analyses in Gaussian-mixture settings formalize how particular output features become determined over narrow time intervals of the generative process (Li & Chen, 2024). Several recent works have investigated generalization in diffusion models from complementary perspectives. Li et al. (2024) show that models in the generalization regime can remain close to a Gaussian linear model, suggesting an inductive bias toward learning covariance structure rather than memorizing samples. Zhang et al. (2026) connect generalization with balanced internal representations, offering another view of how learned networks depart from empirical optimal solutions. In addition, neural-network spectral bias (Rahaman et al., 2019; Wang & Pehlevan, 2025; Kadkhodaie et al., 2024) may influence which components of the data distribution diffusion models learn most readily. Beyond these mechanisms, practical factors that promote generalization include early stopping, limiting capacity relative to dataset size (Li et al., 2024), increasing training data when memorization and generalization scale differently (Bonnaire et al., 2025), and auxiliary noise-class conditioning for distributional contrast (Yoon et al., 2023).

## 2. Background

Let $p$ denote a data distribution on $\mathbb{R}^d$. Let $\boldsymbol{X} \sim p$ denote a random variable following $p$ and $\boldsymbol{Z} \sim \mathcal{N}(0, I_d)$ a standard Gaussian random variable independent of $\boldsymbol{X}$. For the training and sampling of diffusion models, we mostly follow the description given in (Karras et al., 2022).

**Training.** In diffusion models, one of the central tasks is to denoise a noisy observation $\boldsymbol{X}_\sigma = \boldsymbol{X} + \sigma \boldsymbol{Z}$ for various noise levels $\sigma > 0$ in order to obtain $\boldsymbol{X}$. In particular, one popular diffusion training loss is defined as

$$\mathcal{L}(\theta) := \mathbb{E}_{\substack{\boldsymbol{X} \sim p \\ \boldsymbol{Z} \sim \mathcal{N}(0, I_d) \\ \sigma \sim \lambda}} \left\| m_\sigma^\theta(\boldsymbol{X}_\sigma) - \boldsymbol{X} \right\|^2 \tag{1}$$

where $\lambda$ denotes certain distribution on some chosen $[\sigma_{\min}, \sigma_{\max}] \subset (0, \infty)$, and $m_\sigma^\theta : \mathbb{R}^d \to \mathbb{R}^d$ is a neural network parameterized by $\theta$ that aims to approximate the *(optimal) denoiser* $m_\sigma$ that we will introduce below. We refer to $m_\sigma^\theta$ as a *trained denoiser* in what follows.

**Optimal solution.** The above loss admits a unique global minimum given by the posterior mean which we call the *optimal denoiser*: $m_\sigma(x) := \mathbb{E}[\boldsymbol{X} \mid \boldsymbol{X}_\sigma = x]$. In practice, a model is trained with finite samples $\mathcal{D} = \{x_i\}_{i=1}^N$ assumed to be drawn i.i.d. from some data distribution $p$. In this case, the expectation over $\boldsymbol{X} \sim p$ in the training loss Eqn. (1) can be replaced by $\boldsymbol{X} \sim p_\mathcal{D}$ where $p_\mathcal{D} := \frac{1}{N} \sum_{i=1}^N \delta_{x_i}$, and the resulting optimal denoiser, which we now call the *empirical optimal denoiser*, has a closed form given by

$$m_\sigma(x) = \sum_{i=1}^N w_i(x, \sigma) x_i \tag{2}$$

where $w_i(x, \sigma) = \mathbb{P}(\boldsymbol{X} = x_i \mid \boldsymbol{X}_\sigma = x)$ is called the *posterior weight* on $x_i$ at the noise level $\sigma$ and is given by

$$w_i(x, \sigma) = \frac{\exp\left(-\frac{\|x - x_i\|^2}{2\sigma^2}\right)}{\sum_{j=1}^N \exp\left(-\frac{\|x - x_j\|^2}{2\sigma^2}\right)}. \tag{3}$$

**Sampling.** After training, one samples $z \sim \mathcal{N}(0, I_d)$ and lets $x_{\sigma_{\max}} = \sigma_{\max} z$ for some large $\sigma_{\max}$. New samples are generated by solving the following ODE:

$$dx_\sigma / d\sigma = -(m_\sigma^\theta(x_\sigma) - x_\sigma)/\sigma \tag{4}$$

from $\sigma_{\max}$ to $\sigma_{\min}$. For any $\sigma \in [\sigma_{\min}, \sigma_{\max}]$, we denote by $\Psi_\sigma^\theta : \mathbb{R}^d \to \mathbb{R}^d$ the flow map induced by the denoiser $m_\sigma^\theta$, which maps $x_{\sigma_{\max}}$ to $x_\sigma$ by solving the ODE above.

## 3. Empirical Investigation of Memorization

To understand where and how memorization emerges during diffusion model training, we begin with an empirical investigation that motivates our subsequent geometric framework.

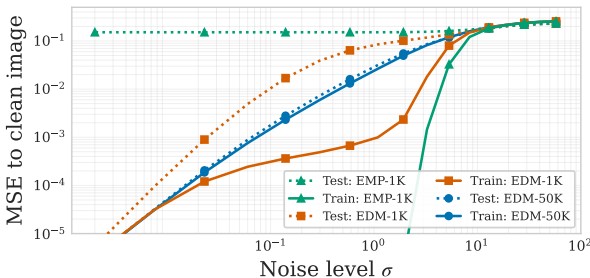

*Figure 1.* **MSE to Clean Image.** Comparison of denoising quality across noise levels. Solid lines: training data; dotted lines: test data. **EDM-1K** shows a generalization gap in the mid-$\sigma$ region.

By comparing models with different memorization behaviors across the noise schedule, we find the intermediate noise level is the most susceptible to memorization.

We start by comparing trained denoisers and the empirical optimal denoiser on the CIFAR-10 dataset (Krizhevsky et al., 2009). We adopt the EDM from (Karras et al., 2022) to model $m_\sigma^\theta$ and consider three denoisers: a full-data EDM denoiser trained on 50k images (**EDM-50K**), an EDM denoiser trained on only 1k images (**EDM-1K**)[1], and the empirical optimal denoiser under the same 1k training images (**EMP-1K**).

One straightforward way to compare the three models is to evaluate their denoising losses via Eqn. (1). For a fixed noise level $\sigma$, this objective reduces to the denoising mean squared error (MSE). We therefore evaluate the fixed-$\sigma$ denoising MSE of the three models over $\sigma \in [0.002, 80]$ on both training and test data from CIFAR-10:

$$\mathrm{MSE}_\sigma(p_\bullet) := \mathbb{E}_{\substack{\boldsymbol{X} \sim p_\bullet \\ \boldsymbol{Z} \sim \mathcal{N}(0, I_d)}} \| m_\sigma^\bullet(\boldsymbol{X}_\sigma) - \boldsymbol{X} \|^2 \quad (5)$$

where $m_\sigma^\bullet$ denotes either a trained or an optimal denoiser and $p_\bullet$ is either the training distribution $p_\mathcal{D}$ or the test distribution $p_\mathcal{T}$.

Intuitively, $\mathrm{MSE}_\sigma(p_\mathcal{D})$ measures how well a model can recover a training image from its noisy version at noise level $\sigma$, while $\mathrm{MSE}_\sigma(p_\mathcal{T})$ measures the corresponding performance on unseen test images. Taken together, these two quantities provide a coarse diagnostic of generalization behavior across noise scales: a small train–test gap suggests similar behavior on training and test data, whereas a large gap indicates degradation on test samples and may signal memorization. We emphasize, however, that train–test MSE alone is not a definitive measure of memorization, but serves here as a useful summary statistic.

The results are shown in Figure 1. Interestingly, we observe three distinct noise regions in which the three models exhibit

---

[1]Training on 1k CIFAR-10 images is known to lead to memorization (Gu et al., 2025). We also validate this in Table 1. The 1k data is the first 1k in CIFAR-10 which is randomly indexed.

*Table 1.* **Memorization Rates in Denoiser Swapping.** Memorization measured as fraction of 256 samples where $d_{1\mathrm{NN}} < d_{2\mathrm{NN}}/3$. Swapping in medium region flips behavior.

| Condition | Noise Range ($\sigma$) | Mem. Rate |
| --- | --- | --- |
| EDM-1K (default sample) | $[0.002, 80]$ | 92.2% |
| EDM-50K (default sample) | $[0.002, 80]$ | 0.0% |
| *EDM-1K → EDM-50K swap:* | | |
| large | $\sigma > 8.4$ | 93.0% |
| **medium** | $[0.14, 8.4]$ | **0.0%** |
| small | $\sigma < 0.14$ | 91.0% |
| *EDM-50K → EDM-1K swap:* | | |
| large | $\sigma > 8.4$ | 0.0% |
| **medium** | $[0.14, 8.4]$ | **92.2%** |
| small | $\sigma < 0.14$ | 0.0% |

qualitatively different behavior. At this stage, this partition is purely observational; in subsequent sections, we provide a geometric characterization that explains the emergence of these large-, intermediate-, and small noise regimes.

1. At large $\sigma$ (i.e., large noise), all three denoisers behave similarly on both train and test. We will later explain in Section 4.2.2 that this is due to the fact that high noise denoising is dominated by coarse statistics.

2. At middle $\sigma$, **EMP-1K** sharply diverges from **EDM-50K**, achieving near-zero training error but large test error due to its nearest-neighbor nature, while **EDM-50K** continues to generalize well.

3. Interestingly, at small $\sigma$, although **EDM-1K** generates training data during sampling, it differs from **EMP-1K** (the empirical optimal denoiser) in its per $\sigma$-level denoising behavior on test data.

This three-regime pattern motivates a closer examination of memorization for the three denoisers. In particular, instead of only considering the classic notion of memorization, we should also consider memorization risk *at each noise level*. While the MSE gap between train and test data may be a good metric for generalization, it is not directly capturing memorization. To resolve this issue, we propose the following metrics for evaluating memorization.

**Trajectory-Level memorization.** For a denoiser $m_\sigma^\bullet$ with noise schedule $\sigma \in [\sigma_{\min}, \sigma_{\max}]$, let $\Psi_\sigma^\bullet$ be the resulting flow map given by solving Eqn. (4). Intuitively, the denoiser $m_\sigma^\bullet$ is said to have memorized the data set $\mathcal{D}$ if its generated samples through the flow map are close to one of the data points in $\mathcal{D}$. While there could be many ways of quantifying this intuition, we adopt the idea from (Yoon et al., 2023), in which they proposed to classify a generated sample as memorized if it is significantly closer

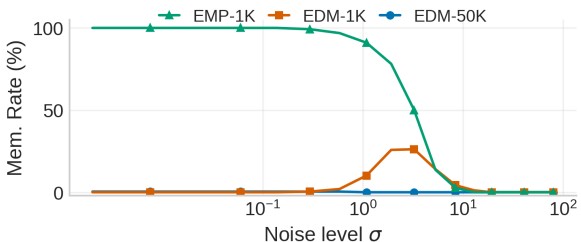

*Figure 2.* **Per-Noise-Level Memorization Rate.** Fraction of denoised test images classified as memorized at each noise level.

to its nearest training neighbor than to the second nearest. Formally, we say $\{m_\sigma^\bullet\}_{\sigma \in [\sigma_{\min}, \sigma_{\max}]}$ is memorizing if for a large portion of noise samples $\boldsymbol{Z} \sim N(0, I_d)$, $d_{1\mathrm{NN}}(\Psi_{\sigma_{\min}}^\bullet(\sigma_{\max}\boldsymbol{Z})) < d_{2\mathrm{NN}}(\Psi_{\sigma_{\min}}^\bullet(\sigma_{\max}\boldsymbol{Z}))/3$, where $d_{K\mathrm{NN}}$ denotes the distance to the $K$th nearest point in $\mathcal{D}$. In particular, we compute the ratio of samples satisfying this condition to quantify the trajectory-level memorization rate.

**Per-Noise-Level memorization.** Motivated by the definition above, we identify a finer-grained notion of memorization at each noise level $\sigma$. Given a training set $\mathcal{D} \subset \mathbb{R}^d$ sampled from a population distribution $p$, a denoiser $m_\sigma^\bullet$ is said to have memorized $\mathcal{D}$ at a noise level $\sigma$ if for (nearly) all sample test points $x_{\text{test}} \sim p$ and a large portion of noise samples $\boldsymbol{Z} \sim \mathcal{N}(0, I_d)$, we have $d_{1\mathrm{NN}}(m_\sigma^\bullet(x_{\text{test}} + \sigma\boldsymbol{Z})) < d_{2\mathrm{NN}}(m_\sigma^\bullet(x_{\text{test}} + \sigma\boldsymbol{Z}))/3$. Under this notion we can examine denoiser performance at a specific noise level.

With these two metrics, we now analyze where memorization arises for **EDM-1K** and **EDM-50K**. As expected and shown in Table 1, **EDM-1K** exhibits strong trajectory-level memorization (92.2% of samples), while **EDM-50K** generalizes well (0% memorization); we omit **EMP-1K**, which exhibits 100% memorization. We now examine per-noise-level memorization across the three models. Figure 2 shows that **EDM-50K** exhibits no memorization for any $\sigma$, whereas **EMP-1K** increasingly approaches nearest-neighbor behavior as $\sigma \downarrow$, reaching $\approx 100\%$ for $\sigma < 0.3$. In contrast, **EDM-1K** attains its highest memorization rate in an intermediate regime (peaking at $\sim 26\%$ around $\sigma \in [1.1, 5.3]$) and drops to nearly 0 memorization rate at very small $\sigma$. We include qualitative visualizations of one-step denoising outputs across different noise levels in Appendix F.2, which further illustrate this phenomenon. In summary, while both **EDM-1K** and **EMP-1K** exhibit strong trajectory-level memorization, the diffusion model behind **EDM-1K** in fact deviates significantly from the empirical optimal **EMP-1K**. In particular, the per-noise-level memorization behaviors are quite different from mid-to-small levels of $\sigma$s.

Finally, we would like to investigate how per-noise-level memorization affects the trajectory-level memorization. We conduct the *denoiser swapping experiments* (Table 1, see

Appendix F.3 for details) which show that swapping the denoiser in the medium region ($\sigma \in [0.14, 8.4]$) flips memorization behavior. Specifically, replacing **EDM-1K** with **EDM-50K** in the medium region drops memorization from 92.2% to 0%, while the reverse (swapping **EDM-50K**'s denoiser with **EDM-1K**'s in this range) increases it from 0% to 92.2%. In contrast, swapping in large or small regions has negligible effect.

Taken together, these results reveal a qualitative mismatch between empirical optimality (**EMP-1K**) and learned memorization (**EDM-1K**): while the empirical optimal memorizes strongly across a broad noise range, the learned diffusion model memorizes primarily in an intermediate regime. This suggests that to fully understand memorization in diffusion models, we not only need to analyze the trajectory-level behavior but also the per-noise-level contributions. In particular, if one wants to mitigate memorization, it is crucial to understand and control the denoiser behavior at per-noise-level, especially in the medium noise regime.

*Remark* 3.1. Notably, the **EDM-1K** shows 92.2% trajectory-level memorization, however its peak per-noise-level memorization is only around 26.2%. This suggests that repeated denoising along the sampling trajectory exhibits an accumulation effect that can amplify memorization risk. We verify this empirically in Section F.4 on ODE trajectory windows using EDM-1K: we start a Euler discretization of (4) at different noise levels $\sigma$ to find that the intermediate noise level along the trajectory can significantly boost the memorization rate. For example, even if the one-step memorization rate at the noise level $\sigma = 3.3$ is only 14.1%, after four ODE steps, the memorization rate jumps to 91%.

## 4. Geometric Interpretation of Noise Regimes

Section 3 shows that per-noise-level memorization concentrates at medium noise levels and contributes most to trajectory-level memorization (as shown by the denoiser swapping experiments in Table 1). In this section, we seek to provide a geometric interpretation of this phenomenon. In particular, we characterize the three regimes and explain why the two ends are "calm" while the middle is "wild". We further identify a "danger zone" of memorization risk in the medium noise regime and provide a practical mitigation strategy in Section 5.

### 4.1. Key Players for Per-Noise-Level Memorizations

To analyze per-noise-level memorization we examine the training loss Eqn. (1) when $\sigma$ is fixed, which reduces to $\mathrm{MSE}_\sigma(p_\mathcal{D})$ (cf. Eqn. (5)). Two major players arise from this loss: the empirical optimal denoiser $m_\sigma$, which the model attempts to fit, and the distribution of noisy training samples $\boldsymbol{X}_\sigma$, which determines the supervised region. With sufficient model capacity, the trained denoiser $m_\sigma^\theta$ can match

$m_\sigma$ on this region; but if this supervision is not of sufficient coverage of test samples, the trained denoiser might not generalize beyond that. Consequently, whether this induces per-noise-level memorization depends on how these two players interact as the noise level varies.

To understand these two players, we analyze two quantities below: the *posterior weight*, induced from the empirical optimal denoiser, and the so-called *Gaussian shell coverage*, induced from the distribution of training samples. Their combined behavior at different noise levels leads to our characterization of three distinct noise regimes (cf. Section 4.2).

**Posterior Weight.** Recall that the functions $w_i(x, \sigma)$ defined in Eqn. (3) are the posterior weights on the data samples $x_i$ at a noise level $\sigma$. From Eqn. (2) it is clear that the posterior weights determine how close the empirical optimal denoiser is to a given data point. In particular if at a given noise level one weight is always substantially larger than the rest, the empirical optimal denoiser becomes concentrated near the corresponding data points, resulting in per-noise-level memorization of the training data. As a trained model attempts to fit the empirical optimal denoiser, it is worth investigating which noise levels yield this effect. To make this more precise, we define the following quantity.

**Definition 4.1.** The *posterior weight for the dataset* $\mathcal{D} = \{x_i\}_{i=1}^N$ at a noise level $\sigma$ is given by the quantity

$$W_\sigma(\mathcal{D}) := \mathbb{E}_{\substack{\boldsymbol{X} \sim p_{\mathcal{D}} \\ \boldsymbol{Z} \sim \mathcal{N}(0, I_d)}} \max_{1 \leq i \leq N} w_i(\boldsymbol{X}_\sigma, \sigma)$$

Note that we only take expectation over samples from a fixed dataset $\mathcal{D}$. This is due to the fact that during training, the model only has access to $\boldsymbol{X}_\sigma$ where $\boldsymbol{X}$ is drawn from the training data distribution $p_{\mathcal{D}}$. This definition captures the expected maximum posterior weight over all noisy samples at a given noise level sigma. It can be estimated directly from a dataset; see e.g., Figure 3, which shows a sharp moderate-noise transition where the weight concentrates. Such a transition phenomenon is not surprising and has been observed and characterized in prior works, e.g., (Biroli et al., 2024). Below, we provide explicit and data-dependent probabilistic bounds for describing this posterior concentration at a fixed noise level, with quantities directly computable from datasets.

First of all, consider any sample data point, and w.l.o.g, assume we consider $x_1 \in \mathcal{D}$. Although the maximum weight $\max_i w_i(x_1 + \sigma \boldsymbol{Z}, \sigma)$ is present in the definition above, this maximum is usually attained at moderate noise levels for $i = 1$, i.e., the maximum is attained at the data point itself. See Figure 19 for an empirical validation, where the two quantities are compared across a full noise schedule. For this reason we provide the following theorem that gives noise thresholds for high probability weight concentration;

see Section B.1 for a proof.

**Theorem 4.2.** *Let $x_1 \in \mathcal{D}$ and condition on $\boldsymbol{X} = x_1$. Let $w_1(\boldsymbol{X}_\sigma, \sigma)$ denote the posterior weight on $x_1$ at a noise level $\sigma$. Then for any $\delta \in (0, 1)$ and $q \in (\frac{1}{2}, 1)$, with probability at least $1 - \delta$, we have that:*

*1. If $\sigma \geq \min_{K>1} \frac{d_{K\mathrm{NN}}(x_1)}{a_{K-1,\delta,q}}$, then $w_1(\boldsymbol{X}_\sigma, \sigma) \leq q$.*

*2. If $\sigma \leq \frac{d_{2\mathrm{NN}}(x_1)}{b_{\delta,q}}$, then $w_1(\boldsymbol{X}_\sigma, \sigma) \geq q$.*

*where the constants $a_{K,\delta,q}$ and $b_{\delta,q}$ are defined by*

$$a_{K,\delta,q} := F^{-1}(\delta/K) + \sqrt{\left(F^{-1}(\delta/K)\right)^2 + 2\log\left(\frac{Kq}{1-q}\right)}$$

$$b_{\delta,q} := \tilde{F}^{-1}(\delta/N) + \sqrt{\left(\tilde{F}^{-1}(\delta/N)\right)^2 + 2\log\left(\frac{Nq}{1-q}\right)}$$

*and where $F$ is the CDF of the standard normal distribution and $\tilde{F} := 1 - F$. (Note that $d_{1\mathrm{NN}}(x_1) = 0$.)*

*Remark* 4.3. We consider a dataset consisting of 1K images randomly sampled from the CIFAR-10 dataset. Using the bounds given by Theorem 4.2, over a random sample of 20 points, the average lower bound needed to guarantee posterior concentration below 0.6 with probability 0.95 was $\sigma \geq 22.54$. The average upper bound needed to guarantee posterior concentration above 0.95 with probability 0.95 was $\sigma \leq 2.25$, which compares well with Figure 3: in the same setting $\sigma = 2.25$ lies close to where the average max weight reaches 0.95.

*Remark* 4.4. In Section B.2 we use a variant of Theorem 4.2 to theoretically explain the sharp alignment between the optimal and conditional vector fields observed in high dimensional flow matching settings by (Bertrand et al., 2026).

**Gaussian Shell Coverage.** (Song et al., 2025) proposes using Gaussian shells around training data points to model supervised regions during training. Following their intuition, we will use a theoretically tighter bound of Gaussian shells (cf. Lemma 4.5) and propose a novel notion of data coverage utilizing the supervised regions via Gaussian shells. The following high dimensional concentration result is key to our analysis in this paper (see proof in Section C).

**Lemma 4.5** (Gaussian concentration (Laurent & Massart, 2000)). *Let $d \geq 2$ and fix $c > 0$. Define $r_{c,d}^{\mathrm{in}} := \sqrt{d - 2\sqrt{cd}}$, and $r_{c,d}^{\mathrm{out}} := \sqrt{d + 2\sqrt{cd} + 2c}$. Let $\boldsymbol{Z} \sim \mathcal{N}(0, I_d)$. Then, $\mathbb{P}\left(\|\boldsymbol{Z}\| \in [r_{c,d}^{\mathrm{in}}, r_{c,d}^{\mathrm{out}}]\right) \geq 1 - 2e^{-c}$.*

This result states that $Z \sim \mathcal{N}(0, I_d)$ will concentrate in a thin shell around the origin with radius $\approx \sqrt{d}$ and thickness controlled by $c$. If we let $c = 5$, then the concentration probability is at least 0.9865. So we will simply let $c = 5$ through this paper unless otherwise stated.

Based on the concentration result, for any $x \in \mathbb{R}^d$ and $\sigma > 0$, we define the *Gaussian shell* centered at $x$ by

$$S_\sigma(x) := \left\{ x' \in \mathbb{R}^d : \sigma r_{c,d}^{\text{in}} \leq \|x' - x\| \leq \sigma r_{c,d}^{\text{out}} \right\}. \quad (6)$$

As a direct consequence of Lemma 4.5, we have that

$$\mathbb{P}\big(\boldsymbol{X}_\sigma \in S_\sigma(x) \mid \boldsymbol{X} = x\big) \geq 1 - 2e^{-c}.$$

This shows that a single Gaussian shell provides a high-probability description of where a noisy data point lies. Based on the notion of Gaussian shells, we next introduce the notion of data coverage.

**Definition 4.6** (Gaussian Shell Coverage). Suppose $\mathcal{D} = \{x_i\}_{i=1}^N$ is a finite dataset sampled from the data distribution $p$. We define the *Gaussian shell coverage* of the distribution $p$ relative to $\mathcal{D}$ at the noise level $\sigma$ as follows:

$$C_\sigma(p, \mathcal{D}) := \mathbb{P}\left( \boldsymbol{X}_\sigma \in \bigcup_{i=1}^N S_\sigma(x_i) \right), \quad \text{where } \boldsymbol{X} \sim p$$

The quantity approximately quantifies the likelihood that a diffusion model encounters a noisy test $x_{\text{test}} + \sigma z$ during training. In other words, $C_\sigma$ measures how large a portion does the training data covers the underlying ground-truth distribution after convolution $p_\sigma = p * \mathcal{N}(0, \sigma^2 I)$. For brevity, in the remainder of the paper we will refer to the Gaussian shell coverage $C_\sigma(p, \mathcal{D})$ simply as "coverage" when no confusion can arise. Note that while in applications the data distribution $p$ is not known, this quantity can be empirically estimated if a test dataset is available (or one could simply use a held-out subset of the training data which does not require access to external test data); see Figure 3 where we estimate $C_\sigma$ for the CIFAR-10 dataset.

Figure 3 also exhibits an interesting sharp phase transition for the coverage as the noise level varies. To provide a theoretical understanding why this happens, we find it better to treat the dataset as random and study the coverage in expectation. Interestingly, it turns out that controlling this coverage is closely related to the probability mass of intersection between just two shells. Below we provide a clean description of such a probability mass.

**Definition 4.7.** Let $e_1 = [1, 0, \cdots, 0]^T \in \mathbb{R}^d$ and define

$$\Phi_{d,c}(t) := \mathbb{P}\big(\|\boldsymbol{Z}\|, \|\boldsymbol{Z} + te_1\| \in [r_{c,d}^{\text{in}}, r_{c,d}^{\text{out}}]\big).$$

The function $\Phi_{d,c}(t)$ measures the probability of a Gaussian noisy observation for one data point lying also in the Gaussian shell over a data point at distance $t$ and hence roughly represents the "probability of seeing a noisy data in the intersection of two Gaussian shells". The following result shows that the coverage is closely controlled by $\Phi_{d,c}$.

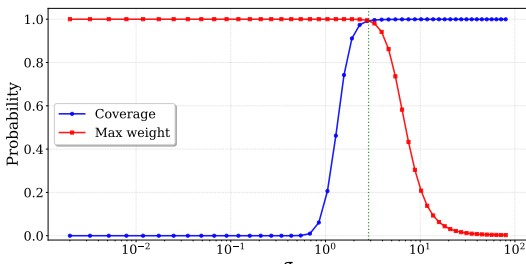

*Figure 3.* **Gaussian shell coverage and max posterior weight on CIFAR-10.** We plot empirical estimates of the Gaussian shell coverage $C_\sigma(p, \mathcal{D})$ and the max posterior weight $W_\sigma(\mathcal{D})$ as functions of the noise level $\sigma$, using a 1k CIFAR-10 training subset and 1k held-out test images.

**Theorem 4.8.** *Let* $\boldsymbol{X}, \boldsymbol{X}_1, \ldots, \boldsymbol{X}_N \overset{i.i.d.}{\sim} p$, *and let* $\boldsymbol{Z} \sim \mathcal{N}(0, I_d)$ *be independent of* $(\boldsymbol{X}, \boldsymbol{X}_1, \ldots, \boldsymbol{X}_N)$. *Fix a noise level* $\sigma > 0$. *Define the union of shells* $\mathcal{U}_\sigma := \bigcup_{i=1}^N S_\sigma(\boldsymbol{X}_i)$. *Then the coverage probability satisfies*

$$\mathbb{P}(\boldsymbol{X}_\sigma \in \mathcal{U}_\sigma) \geq \mathbb{E}\left[\Phi_{d,c}\left(\frac{d_{1\text{NN}}(\boldsymbol{X})}{\sigma}\right)\right],$$

$$\mathbb{P}(\boldsymbol{X}_\sigma \in \mathcal{U}_\sigma) \leq 2e^{-c} + N \, \mathbb{E}\left[\Phi_{d,c}\left(\frac{\|\boldsymbol{X} - \boldsymbol{X}'\|}{\sigma}\right)\right]$$

*where* $\boldsymbol{X}'$ *is an independent copy of* $\boldsymbol{X}$.

See Section C.1 for the proof. The two bounds in the theorem are governed by the distribution of pairwise distances in the data as well as the ambient dimension $d$ reflected by $\Phi_{d,c}$. The lower bound actually provides a very accurate estimation empirically and one may directly use it as a proxy to estimate the coverage; see Figure 15 for experimental results on CIFAR-10.

### 4.2. Characterization of the Noise Regimes

With the two quantities of posterior weight and data coverage defined and explored, we are ready to characterize the noise regimes. In Figure 3 we plot two curves of the two quantities against noise levels, based on a 1k subset of the CIFAR-10 dataset.

Note that in Figure 2, **EDM-1K** exhibits a positive memorization ratio for noise levels range $\sigma \in [0.6, 12]$, which closely aligns with the transition interval observed in Figure 3. Based on all these observations, we now informally identify the three noise regimes as follows using the two quantities:

| Regime | Posterior weight | Shell coverage |
|---|---|---|
| Small noise | High | Low |
| Medium noise | Phase transition | Phase transition |
| -Danger zone- | High | High |
| Large noise | Low | High |

While informal this description may serve as a guiding prin-

ciple for classifying training dynamics across a full noise schedule. Precise estimation of noise ranges requires further work, but our Theorem 4.2 and Theorem 4.8 serve as a starting point. Appendix F.6 provides additional dataset-size and latent-space scaling diagnostics, showing that the relevant nearest-neighbor distance distributions and coverage–weight transition structure remain stable as the dataset size increases. In what follows, we provide rationales for this characterization and further analysis within each regime.

### 4.2.1. SMALL NOISE REGIME

The small noise (small $\sigma$) regime is characterized by high posterior weight and low coverage. This is where the empirical optimal denoiser behaves almost like a nearest neighbor map. If a trained model well approximates the empirical optimal one, it will absolutely produce memorized training data and hence it is natural to suggest mitigation for memorization in this region (Wan et al., 2025; Baptista et al., 2025). However, although the empirical optimal denoiser exhibits strong memorization behavior (which is almost locally constant), the trained denoiser is unlikely to fit the empirical optimal denoiser since the supervision region during training, described by shell coverage, is very limited.

We have empirically observed in Figure 2 that the small noise regime is quite safe from memorization risk in per-noise-level: while the empirical optimal denoiser exhibits nearly 100% memorization for $\sigma < 0.3$, the learned model (**EDM-1K**) shows minimal memorization in this regime. We also validate in the swapping experiment in Table 1 that the small noise regime in a trained diffusion model does not contribute to memorization.

Within the small noise regime, when $\sigma$ is small enough (e.g., $< 0.2$ for the CIFAR-10), all Gaussian shells around data points will be disjoint. In this setting, we have the following description of the optimal solution to the training objective. See Theorem C.6 for the complete statement and proof.

**Theorem 4.9** (Informal theorem). *Suppose the training is only restricted to the union of shells $\bigcup_{i=1}^{N} S_\sigma(x_i)$, which are also pairwise disjoint for small $\sigma$. Then, there are infinitely many global minimizers of the denoising objective sending $S_\sigma(x_i)$ to $x_i$ for $i = 1, \ldots, N$.*

In the small-$\sigma$ regime, training supervision is concentrated within the small region consisting of the union of Gaussian shells. Then by Theorem 4.9, the training objective alone does not constrain the denoiser outside this small region, so extrapolation behavior is largely determined by the model's inductive bias. In one dimension, shallow network denoisers have been shown to interpolate the data with piecewise linear behavior (Zeno et al., 2023). In high dimensional and deep network settings, this behavior may not hold. Below we try to provide some evidence that certain locality

inductive bias will appear for image datasets.

For an image data such as CIFAR-10, we hypothesize that the locality increases when $\sigma$ decreases: each output pixel depends mainly on a local patch. The task therefore reduces to patch denoising in a much lower-dimensional subspace, where the effective training data, the extracted patches, are plentiful, making the problem easier. This aligns with empirical findings in (Kamb & Ganguli, 2025; Lukoianov et al., 2025; Niedoba et al., 2025): trained neural denoisers exhibit shrinking receptive fields as the noise level decreases. (Kamb & Ganguli, 2025; Lukoianov et al., 2025; Wang et al., 2025) finds the optimal denoiser under locality bias and (Lukoianov et al., 2025) connects the localization to the data statistics. However, it was not clear what affects the spatial decay rate of the sensitivity. To study this analytically, we model the data as a Gaussian with circulant covariance—a natural idealization, since natural images exhibit approximately translation-invariant second-order statistics and stationary Gaussian models are a standard tool in their analysis (Field, 1987; Simoncelli & Olshausen, 2001). We propose the following theorem; formal statements and proofs are in Appendix E.

**Theorem 4.10** (Informal theorem). *For a cyclic stationary Gaussian data distribution, at any noise level $\sigma$, the sensitivity between two pixels is bounded by $\frac{T_\sigma}{m}$, where $m$ is their wrap-around distance and $T_\sigma$ is a constant depending only on the data spectrum and $\sigma$. A more concentrated spectrum leads to faster decay of the sensitivity.*

### 4.2.2. LARGE NOISE REGIME

We characterize the large noise regime by low posterior weight and high coverage. The latter implies that the model $m_\sigma^\theta$ can learn the empirical optimal on a region that covers most of the mass of $p_\sigma$, where inference takes place. In this regime, the empirical optimal denoiser does not exhibit per-noise-level memorization behavior, and hence even if the trained denoiser closely approximates the empirical optimal denoiser, per-noise-level memorization risk remains low.

In fact, we can provide some more refined description of the empirical optimal denoiser within this regime.

It is known that when $\sigma$ is large, the denoiser $m_\sigma$ will be close to the mean $\mu$ of the data distribution $p$ (Wan et al., 2025). Empirically, it has also been observed that $m_\sigma$ behaves like a linear map when $\sigma$ is large (Li et al., 2024). In what follows, we reconcile these two observations by establishing the limiting behavior of $m_\sigma$ when $\sigma$ is large.

Let $\Sigma$ denote the covariance of $p$, and recall that $\mu$ is the mean of $p$. Consider now the Gaussian distribution $\mathcal{N}(\mu, \Sigma)$ that matches the mean and covariance of $p$. The denoiser $m_\sigma^G$ of this Gaussian distribution is linear and has the closed

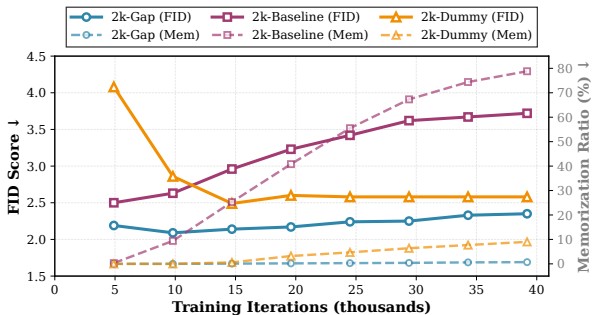

*Figure 4.* **Comparison of anti-memorization methods.** Training curves showing FID score (solid lines) and memorization ratio (dashed lines) for three methods: 2k-Baseline, 2k-Dummy, and 2k-Gap. Gap training achieves the lowest memorization (0.7%) and best FID (2.35) at final checkpoint.

form (Li et al., 2024):

$$m_\sigma^G(x) = \mu + \Sigma(\Sigma + \sigma^2 I)^{-1}(x - \mu). \qquad (7)$$

Note that when $\sigma$ is large, the linear term will be small and hence $m_\sigma^G \approx \mu$. It turns out that this optimal linear denoiser for Gaussian captures the limiting behavior of $m_\sigma$ when $\sigma$ is large; see our Theorem 4.11 below and see Section D for a proof. We note that Wang & Vastola (2024) has established a similar result for the empirical distribution. Our result not only applies to any bounded-support distribution, including both the empirical training distribution and the ground-truth data distribution, but also gives an explicit decay rate for the residual term.

**Theorem 4.11.** *Assume* $\text{supp}(p) \subset B_R(0)$. *Then, for any* $\sigma_0 > 0$, *there exists a bounded continuous function* $H$ *on* $\overline{B_R(0)} \times [\sigma_0, \infty)$ *such that*

$$m_\sigma(x) = m_\sigma^G(x) + H(x, \sigma)(1 + \sigma)^{-2}.$$

This theorem indicates that for large $\sigma$, the denoiser (the one either for the true data distribution $p$ or for the empirical distribution over a dataset) is close to being a linear map determined by the mean and covariance of $p$. This validates the empirical observations in (Li et al., 2024). In particular, applying the theorem to both the empirical and population distributions shows that, at high noise, both denoising targets are governed by their respective first two moments, up to an $O((1 + \sigma)^{-2})$ residual. Consequently, when the empirical mean and covariance are close to their population counterparts, the empirical optimal denoiser is already close to the population optimal denoiser in this regime. This explains why memorization is not driven by high-noise denoising in general. Furthermore, this near-Gaussian structure at high noise can also be used for conditional generation: Wang et al. (2026) exploit it to align unconditional diffusion trajectories with subclass or object directions, enabling guided generation without retraining the model.

### 4.2.3. MEDIUM NOISE REGIME

We characterize the medium noise regime by simultaneous transitions in posterior weight concentration and data coverage: as sigma decreases from large to small, posterior weights concentrate on nearest neighbors while data coverage becomes incomplete. The interplay between these two effects creates a volatile training environment.

There are two key observations we have regarding the medium noise regime. The first is the discovery of a *danger zone* for memorization: note that in Figure 3 there is a small region of $\sigma$ in the medium regime where both coverage and posterior weights are high. In this case, the denoiser $m_\sigma^\theta$ will tend to learn the empirical optimal $m_\sigma$ in a region of $\mathbb{R}^d$ where $p_\sigma$ is concentrated (due to high Gaussian shell coverage) and furthermore, since this empirical optimal $m_\sigma$ exhibits high per-noise-level memorization due to the high posterior concentration, $m_\sigma^\theta$ will likely exhibit high per-noise-level memorization.

The other observation is that generalization appears to arise primarily in the medium sigma regime. Although we do not yet have a theoretical explanation for this behavior, our empirical results provide strong evidence: swapping denoisers in this regime flips memorization behavior (Table 1), with qualitative examples shown in Figure 12. This highlights the medium region as the key regime governing both memorization and generalization in diffusion models.

More theoretical understanding of this regime is left to future work. In the section below, we provide interesting empirical findings regarding this regime: since the medium sigma regime is the most dangerous for memorization, we conjecture that avoiding it during training can mitigate memorization. We validate this idea experimentally in Section 5.

## 5. Targeted Noise Region Undertraining

Our theoretical analysis has identified a "danger zone" in the medium-$\sigma$ regime where memorization risk is highest due to the confluence of high posterior concentration and significant data coverage. A natural question arises: can we mitigate memorization by targeting this specific regime during training? While the swap experiments in Table 1 already provides some positive answer, in this section, we further test a rejection training idea on CIFAR-10 (Figure 4) and CelebA as well as on stable diffusion finetuning.

We emphasize that, however, our removal of intermediate noise regime during training or finetuning should be viewed as a controlled intervention to validate the underlying mechanism, rather than a generally optimal training strategy. In practice, one may instead consider softer or data-dependent modifications of the noise schedule that reduce memorization while preserving the benefits of training in this regime.

**CIFAR-10 car subset.** We consider a CIFAR-10 car subset (2k images) where both generalization and memorization occur (Yoon et al., 2023). As a baseline, we train standard EDM (Karras et al., 2022) with the full noise schedule $\sigma \in [0.002, 80]$ (**2k-Baseline**). Building on the baseline, we compare **2k-Dummy** (Yoon et al., 2023), which adds 2k random noise images as a second class to provide distributional contrast, and **2k-Gap** (ours), which excludes the danger zone $\sigma \in [1.0, 5.0]$ identified empirically via coverage and posterior weight analysis (Appendix F.8). We train for 40k iterations, evaluate at 8 checkpoints, and generate 1,024 samples per checkpoint to compute FID against 5k CIFAR-10 training cars and the memorization ratio.

Figure 4 shows training trajectories. The 2k-Baseline exhibits severe memorization (78.8%) with FID 3.72. The 2k-Dummy method substantially reduces memorization to 9.0% with FID 2.58. Our 2k-Gap achieves the best results: 0.7% memorization and FID 2.35. Appendix F.9 provides an ImageNet goldfish replication: the same coverage–weight diagnostic identifies a danger zone (Figure 21), and excluding this region similarly suppresses memorization during training (Figure 22).

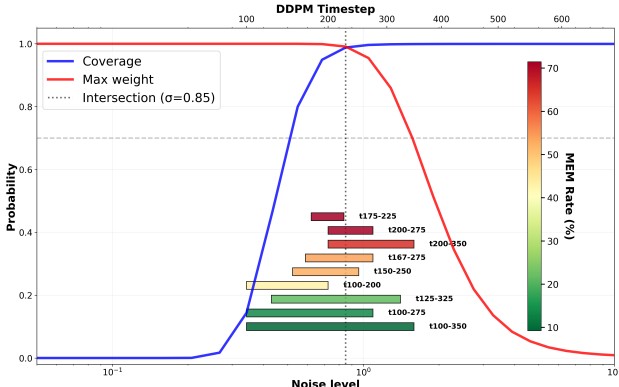

*Figure 5.* **Danger zone and gap configurations.** The intersection of coverage and max weight curves identifies the danger zone. Memorization rates for different gap configurations are overlaid, showing that gaps targeting this region minimize memorization.

**Grayscale CelebA training data.** Following (Bonnaire et al., 2025), we use downscaled grayscale CelebA data with 1024 training samples, which diffusion models can memorize. We compute the coverage and max weight curves (Figure 5) to identify the danger zone around $\sigma \in [0.55, 1.3]$ (approximately DDPM timesteps 125–325). We test multiple timestep gap configurations around this region to study the effect on memorization. We utilize the same DDPM architecture as in (Bonnaire et al., 2025) and train for 100k iterations with batch size 512 and the baseline achieves 74.97% memorization. We then test nine timestep gap configurations, removing different ranges within [100–350] (the danger zone plus some buffer) during training.

Results are summarized in Figure 5. Skipping the full range [100–350] significantly reduces memorization from 75% to 9%. High max weight regions are more influential: the gap [100–200] spanning only 100 timesteps reduces memorization to 43%, while [100–275] achieves 13%. We include in Appendix F.11 a Pareto frontier analysis over all gap configurations, showing the trade-off between image quality (FID) and memorization; mid-range gaps (150–250, 167–275) achieve the best balance. Visual comparisons of generated samples and their nearest training neighbors are also provided in Appendix F.11, showing high sample quality and meaningful generalization under skip training.

**Stable Diffusion (SD) v1.4 finetuning.** We also test the diagnostic in SD v1.4 finetuning on 200 memorized image-prompt pairs. In this prompt-conditioned finetuning set, each prompt is paired with only one image, so the conditioned max weight is always 1 for every $\sigma$ and the danger-zone criterion reduces to estimating the coverage onset. We estimate this onset with a broader LAION data (Schuhmann et al., 2022) lower-bound proxy in SD latent space, obtaining nontrivial coverage near $\sigma \approx 3$ (Figure 23; details in Appendix F.10). The prompt-conditioned setting therefore predicts a one-sided gap that removes training noise levels above this shell-overlap onset. The finetuning results follow this prediction: the final mean SSCD[2] decreases from $0.42$ under the unmodified schedule to $0.29$, $0.22$, and $0.10$ with caps $\sigma < 5$, $\sigma < 3$, and $\sigma < 1$, respectively (Figure 24), with the same qualitative trend in generated samples (Figures 25–27). These results support targeted noise-regime control for memorization mitigation.

## 6. Discussion

Our results indicate that trajectory-level memorization in diffusion models arises from a specific medium range of noise levels, rather than uniformly across the sampling schedule. This per-noise-level view explains why the small and large noise ends can be benign for distinct reasons, while an intermediate danger zone aligns posterior concentration with sufficient training coverage. Practically, this points to targeted controls that act selectively on risky noise scales, such as schedule gaps, reweighting, or data-dependent caps. More broadly, the framework highlights the importance of data geometry and noise-scale interactions in the design and analysis of diffusion models.

## Acknowledgements

This work is partially supported by the National Science Foundation (NSF) under grants CCF-2112665, MFAI-

---

[2]SSCD is a near-duplicate copy-detection descriptor (Pizzi et al., 2022) commonly used to quantify copying in diffusion models (Somepalli et al., 2023b).

2502083, and MFAI-2502084, and by the Defense Advanced Research Projects Agency (DARPA) under Contract No. HR001125CE020.

## Impact Statement

This paper studies the theoretical and empirical mechanisms of memorization in deep generative models, with a focus on diffusion models. Understanding where memorization arises can help improve privacy and copyright risk assessment for models trained on sensitive or proprietary data. Our findings may also inform mitigation strategies that reduce unintended copying by targeting the noise regimes most associated with memorization. The work does not introduce a new extraction attack or release sensitive data; its primary intended impact is to support safer training, auditing, and deployment of generative models.

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

# Appendix

**Table of Contents for Appendix**

# A. Flow Matching Preliminaries

Throughout this paper, we have considered the noisy observation $X_\sigma = X + \sigma Z$ where $X$ is drawn from a data distribution $p$ and $Z \sim \mathcal{N}(0, I_d)$. In this appendix, however, it will sometimes be useful to adopt the notation and perspective of flow-matching. For this reason, we present this section, which gives a brief overview of flow matching and then connects it back with the $\sigma$-parameterization used within the main text.

In flow matching (Lipman et al., 2023), instead of considering $X_\sigma$, people consider another type of interpolation between $X$ and $Z$, which is given by

$$X_t = \alpha_t X + \beta_t Z$$

where $\alpha_t$ and $\beta_t$ are such that $\alpha_0 = \beta_1 = 0$ and $\alpha_1 = \beta_0 = 1$. Letting $p_t := \mathrm{Law}(X_t)$, then $(p_t)_{t \in [0,1]}$ provides a probability path interpolating between the data distribution $p_1 = p$ and the prior distribution $p_0 = \mathcal{N}(0, I_d)$.

One can construct a velocity field $u_t : \mathbb{R}^d \to \mathbb{R}^d$ and then use the continuity equation to regenerate the probability path and hence perform the sampling process. We first define the conditional velocity field for any $x, x_1 \in \mathbb{R}^d$:

$$u_t(x|x_1) = \mathbb{E}\left[ \dot{\alpha}_t x_1 + \dot{\beta}_t Z \mid X_t = x \right],$$

Then, replacing $x_1$ with $X \sim p_1$ and taking the expectation gives the final flow-matching velocity field,

$$u_t(x) = \mathbb{E}[\dot{\alpha}_t X + \dot{\beta}_t Z \mid X_t = x].$$

Now, for any initial point $x_0 \in \mathbb{R}^d$, we solve the ODE below from $t = 0$ to $t = 1$:

$$\frac{dx_t}{dt} = u_t(x_t),$$

and denote the flow map by $\Psi_t(x_0) = x_t$. Then, under certain regularity conditions of $p_0$ (see for example (Wan et al., 2025)), we have that

$$p_t = (\Psi_t)_\# p_0, \quad t \in [0, 1].$$

It turns out that the velocity field $u_t$ can be also expressed in terms of another notion of denoiser (which is slightly different from the one we have used in the main text but still has the same structure as the posterior mean). Specifically, define the denoiser below for each $t \in [0, 1)$ and $x \in \mathbb{R}^d$:

$$m_t(x) := \mathbb{E}[X \mid X_t = x]. \tag{8}$$

Then, we can rewrite the flow-matching velocity field as (see (Wan et al., 2025, Eqn (8)) for details)

$$u_t(x) = \frac{\dot{\beta}_t}{\beta_t} x + \frac{\dot{\alpha}_t \beta_t - \alpha_t \dot{\beta}_t}{\beta_t} m_t(x).$$

**Relation to the $\sigma$ version.** For all $t \in (0, 1)$ and $x \in \mathbb{R}^d$, write $\sigma = \beta_t / \alpha_t$. Consequently, as also implicit in the proof of (Wan et al., 2025, Proposition 2.2),

$$m_t(\alpha_t x) = m_\sigma(x), \tag{9}$$

and hence, for all $t \in (0, 1)$,

$$m_t(X_t) = m_\sigma(X_\sigma). \tag{10}$$

Thus the later flow-matching statements are controlled by the same empirical posterior weights and posterior denoiser as the variance-exploding analysis in the main text.

**OT schedule.** One particular choice of schedule is to let $\alpha_t = t$ and $\beta_t = 1 - t$ which results in clean formulations for the conditional and unconditional flow-matching vector fields. In this case, the flow-matching path is given by

$$X_t = tX + (1 - t)Z, \qquad t \in (0, 1).$$

In this case,

$$u_t(x|x_1) = \frac{x_1 - x}{1 - t} \tag{11}$$

and

$$u_t(x) = \frac{m_t(x) - x}{1 - t}. \tag{12}$$

We will use this OT schedule in Appendix B.2 to study the cosine similarity concentration phenomenon observed by Bertrand et al. (2026).

## B. Posterior Concentration and Cosine Similarity Concentration

### B.1. Proof of Theorem 4.2

*Proof.* Fix $\delta \in (0, 1)$ and $q \in (\frac{1}{2}, 1)$. Recall that conditioning on $\boldsymbol{X} = x_1$ gives $\boldsymbol{X}_\sigma := x_1 + \sigma \boldsymbol{Z}$ where $\boldsymbol{Z} \sim N(0, I_d)$. By definition

$$w_j(\boldsymbol{X}_\sigma, \sigma) = \frac{\exp\left(-\frac{\|\boldsymbol{X}_\sigma - x_j\|^2}{2\sigma^2}\right)}{\sum_{k=1}^N \exp\left(-\frac{\|\boldsymbol{X}_\sigma - x_k\|^2}{2\sigma^2}\right)}.$$

For ease of notation write $w_j = w_j(\boldsymbol{X}_\sigma, \sigma)$ and notice that since $\sum_{j=1}^N w_j = 1$ we have

$$w_1 = \frac{1}{1 + \sum_{j \neq 1} \frac{w_j}{w_1}}. \tag{13}$$

Let $\hat{d}_j := x_j - x_1$ for all $x_j \in \mathcal{D}$. By direct calculation we have

$$\frac{w_j}{w_1} = \exp\left(\frac{1}{\sigma}\langle \boldsymbol{Z}, \hat{d}_j \rangle - \frac{1}{2\sigma^2}\|\hat{d}_j\|^2\right).$$

Define

$$\boldsymbol{V}_j := \frac{1}{\sigma}\langle \boldsymbol{Z}, \hat{d}_j \rangle - \frac{1}{2\sigma^2}\|\hat{d}_j\|^2$$

and without loss suppose that for all $K \in \{1, \ldots, N-1\}$, we have $\|x_{K+1} - x_1\| = d_K$ where $d_K := d_{(K+1)NN}(x_1)$ is the distance from $x_1$ to its $K^{\text{th}}$ nearest neighbor in $\mathcal{D}$. (Under this convention $d_{1NN}(x_1) = 0$). Fix a $K \in \{1, \ldots, N-1\}$ and let

$$c_K := \log\left(\frac{1-q}{qK}\right).$$

Note $c_K < 0$ since $q \in (\frac{1}{2}, 1)$ and therefore $(1-q)/q \in (0, 1)$. Define the event

$$\mathcal{E}_K := \bigcap_{j=2}^{K+1} \left\{\boldsymbol{V}_j \geq c_K\right\}$$

so that on $\mathcal{E}_K$,

$$\sum_{j \neq 1} \frac{w_j}{w_1} = \sum_{j \neq 1} e^{\boldsymbol{V}_j} \geq \sum_{j=2}^{K+1} e^{\boldsymbol{V}_j} \geq K e^{c_K} = \frac{1-q}{q}$$

which gives $w_1 \leq q$ via Eqn. (13). We now bound the complement event $\mathcal{E}_K^c$. By union bound we have

$$\mathbb{P}(\mathcal{E}_K^c) \leq \sum_{j=2}^{K+1} \mathbb{P}(\boldsymbol{V}_j < c_K). \tag{14}$$

For any fixed $j \neq 1$, the key idea is to observe that $\langle \boldsymbol{Z}, \hat{d}_j \rangle \sim \mathcal{N}\left(0, \|\hat{d}_j\|^2\right)$. Hence

$$\mathbb{P}(\boldsymbol{V}_j < c_K) = \mathbb{P}\left(\left\langle \boldsymbol{Z}, \frac{\hat{d}_j}{\|\hat{d}_j\|} \right\rangle < \frac{\|\hat{d}_j\|}{2\sigma} + \frac{\sigma c_K}{\|\hat{d}_j\|}\right) = F\left(\frac{\|\hat{d}_j\|}{2\sigma} + \frac{\sigma c_K}{\|\hat{d}_j\|}\right). \tag{15}$$

where $F$ is the CDF of the standard normal distribution. Put $z := F^{-1}(\delta/K)$ and note that since $F$ and $F^{-1}$ are increasing functions we may write

$$\frac{\|\hat{d}_j\|}{2\sigma} + \frac{\sigma c_K}{\|\hat{d}_j\|} \leq z \quad \Longleftrightarrow \quad F\left(\frac{\|\hat{d}_j\|}{2\sigma} + \frac{\sigma c_K}{\|\hat{d}_j\|}\right) \leq \frac{\delta}{K}. \tag{16}$$

Solving the equation

$$\frac{a}{2} + \frac{c_K}{a} = z$$

yields a positive solution

$$a_{K,\delta,q} := z + \sqrt{z^2 - 2c_K}$$

which is well defined since $c_K < 0$. Calculus shows that the function $g(a) = \frac{a}{2} + \frac{c_K}{a}$ is increasing for all $a > 0$, hence for all $j$,

$$\frac{\|\hat{d}_j\|}{\sigma} \leq a_{K,\delta,q} \quad \Longrightarrow \quad \frac{\|\hat{d}_j\|}{2\sigma} + \frac{\sigma c_K}{\|\hat{d}_j\|} \leq z. \tag{17}$$

Thus by Eqn. (16) and Eqn. (17) we have the following implication:
If

$$\sigma \geq \max_{2 \leq j \leq K+1} \frac{\|\hat{d}_j\|}{a_{K,\delta,q}} = \frac{d_K}{a_{K,\delta,q}}$$

then

$$F\left(\frac{\|\hat{d}_j\|}{2\sigma} + \frac{\sigma c_K}{\|\hat{d}_j\|}\right) \leq \frac{\delta}{K}$$

for all $j \in \{2, \ldots, K+1\}$. By Eqn. (14) and Eqn. (15) this further implies

$$\mathbb{P}(\mathcal{E}_K^c) \leq K \cdot \frac{\delta}{K} = \delta.$$

Since this construction holds for all $K$ we have

$$\sigma \geq \min_K \frac{d_K}{a_{K,\delta,q}} = \min_{K>1} \frac{d_{KNN}(x_1)}{a_{K-1,\delta,q}}$$

implies for $K_* := \arg\min_K (d_K/a_{K,\delta,q})$,

$$\mathbb{P}(\mathcal{E}_{K_*}) \geq 1 - \delta$$

and on $\mathcal{E}_{K_*}$,

$$w_1 \leq q.$$

Plugging in the respective definitions for $c_K$ and $z$ into $a_{K,\delta,q}$ completes the proof of the first implication.

To prove the second implication, let

$$c := \log\left(\frac{1-q}{q(N-1)}\right)$$

and notice that $c < 0$. Define the event

$$\mathcal{F} := \bigcap_{j \neq 1} \left\{V_j \leq c\right\}$$

so that on $\mathcal{F}$,

$$\sum_{j \neq 1} \frac{w_j}{w_1} = \sum_{j \neq 1} e^{V_j} \leq (N-1)e^c = \frac{1-q}{q}$$

which gives $w_1 \geq q$ via Eqn. (13). By union bound we have

$$\mathbb{P}(\mathcal{F}^c) \leq \sum_{j \neq 1} \mathbb{P}(V_j > c). \tag{18}$$

For any fixed $j \neq 1$, observe since $\langle \mathbf{Z}, \hat{d}_j \rangle \sim N(0, \|\hat{d}_j\|^2)$,

$$\mathbb{P}(\mathbf{V}_j > c) = \mathbb{P}\left( \left\langle \mathbf{Z}, \frac{\hat{d}_j}{\|\hat{d}_j\|} \right\rangle > \frac{\|\hat{d}_j\|}{2\sigma} + \frac{\sigma c}{\|\hat{d}_j\|} \right) = \tilde{F}\left( \frac{\|\hat{d}_j\|}{2\sigma} + \frac{\sigma c}{\|\hat{d}_j\|} \right) \tag{19}$$

where $\tilde{F} := 1 - F$. Put $\tilde{z} := \tilde{F}^{-1}(\delta/(N-1))$ and note that since both $\tilde{F}$ and $\tilde{F}^{-1}$ are decreasing functions we may write

$$\frac{\|\hat{d}_j\|}{2\sigma} + \frac{\sigma c}{\|\hat{d}_j\|} \geq \tilde{z} \iff \tilde{F}\left( \frac{\|\hat{d}_j\|}{2\sigma} + \frac{\sigma c}{\|\hat{d}_j\|} \right) \leq \frac{\delta}{N-1}. \tag{20}$$

Then solving the equation

$$\frac{b}{2} + \frac{c}{b} = \tilde{z}$$

yields a positive solution

$$b'_{\delta,q} := \tilde{z} + \sqrt{\tilde{z}^2 - 2c}$$

which is well defined since $c < 0$. By a similar argument to that used in the proof of the first implication we have for all $j \neq 1$,

$$\frac{\|\hat{d}_j\|}{\sigma} \geq b'_{\delta,q} \implies \frac{\|\hat{d}_j\|}{2\sigma} + \frac{\sigma c}{\|\hat{d}_j\|} \geq \tilde{z}. \tag{21}$$

Since $d_1 \leq \|\hat{d}_j\|$ for all $j$, by Eqn. (20) and Eqn. (21) we have the following implication:
If

$$\sigma \leq \frac{d_1}{b'_{\delta,q}}$$

then

$$\tilde{F}\left( \frac{\|\hat{d}_j\|}{2\sigma} + \frac{\sigma c}{\|\hat{d}_j\|} \right) \leq \frac{\delta}{N-1}$$

for all $j \neq 1$. By the above union bound in Eqn. (18) and Eqn. (19) this further implies

$$\mathbb{P}(\mathcal{F}^c) \leq \delta.$$

Lastly notice that since $\tilde{F}^{-1}$ is a decreasing function we have

$$b'_{\delta,q} = \tilde{F}^{-1}(\delta/(N-1)) + \sqrt{\left( \tilde{F}^{-1}(\delta/(N-1)) \right)^2 + 2\log\left( \frac{q(N-1)}{1-q} \right)}$$

$$\leq \tilde{F}^{-1}(\delta/N) + \sqrt{\left( \tilde{F}^{-1}(\delta/N) \right)^2 + 2\log\left( \frac{qN}{1-q} \right)} =: b_{\delta,q}$$

which implies that $\sigma \leq \frac{d_1}{b_{\delta,q}} = \frac{d_{2NN}(x_1)}{b_{\delta,q}}$ yields the same implication. This completes the proof. $\qquad\square$

## B.2. Cosine Similarity Concentration

The following lemma is similar to Theorem 4.2 with an additional bound on the distance from the denoiser $m_\sigma(\mathbf{X}_\sigma)$ to $x_1$ given that $\mathbf{X}_\sigma = x_1 + \sigma \mathbf{Z}$. This demonstrates that for small $\sigma$ the empirical optimal denoiser admits nearest neighbor behavior. Furthermore this result leads to Corollary B.2, where following inspiration from (Bertrand et al., 2026), we demonstrate a conditional bound on the cosine similarity between the conditional and final flow-matching vector fields with linear scheduling.

**Lemma B.1.** *Let $\mathcal{D} = \{x_1, \ldots, x_N\} \subset \mathbb{R}^d$ be a finite data set with uniform distribution $p_{\mathcal{D}}$. Define*

$$D := \max_{1 \leq i,j \leq N} \|x_i - x_j\|.$$

*let $\hat{d}_j := x_j - x_1$. Fix $\epsilon > 0$ and condition on $\boldsymbol{X} = x_1$, then with probability at least $1 - \delta$,*

$$w_1(\boldsymbol{X}_\sigma, \sigma) \geq 1 - \epsilon$$

*and consequently*

$$\|m_\sigma(\boldsymbol{X}_\sigma) - x_1\| \leq D\left(1 - w_1(\boldsymbol{X}_\sigma, \sigma)\right) \leq D\epsilon \tag{22}$$

*where*

$$\delta := \sum_{j \neq 1} \tilde{F}\left(\frac{\|\hat{d}_j\|}{2\sigma} + \frac{\sigma \kappa_\epsilon}{\|\hat{d}_j\|}\right)$$

*and*

$$\kappa_\epsilon := \log\left(\frac{\epsilon}{(N-1)(1-\epsilon)}\right)$$

*and $\tilde{F} := 1 - F$, where $F$ is the CDF of the standard normal distribution.*

*Proof.* Let $\epsilon > 0$ and recall that $\boldsymbol{X} = x_1$ gives $\boldsymbol{X}_\sigma = x_1 + \sigma \boldsymbol{Z}$. We have

$$w_j(\boldsymbol{X}_\sigma, \sigma) = \frac{\exp\left(-\frac{\|\boldsymbol{X}_\sigma - x_j\|^2}{2\sigma^2}\right)}{\sum_{k=1}^N \exp\left(-\frac{\|\boldsymbol{X}_\sigma - x_k\|^2}{2\sigma^2}\right)}.$$

For ease of notation write $w_j := w_j(\boldsymbol{X}_\sigma, \sigma)$. Since $\sum_{j=1}^N w_j = 1$ we have

$$w_1 = \frac{1}{1 + \sum_{j \neq 1} \frac{w_j}{w_1}}. \tag{23}$$

By direct calculation

$$\frac{w_j}{w_1} = \exp\left(\frac{1}{\sigma}\langle \boldsymbol{Z}, \hat{d}_j \rangle - \frac{1}{2\sigma^2}\|\hat{d}_j\|^2\right).$$

Define

$$\boldsymbol{V}_j := \frac{1}{\sigma}\langle \boldsymbol{Z}, \hat{d}_j \rangle - \frac{1}{2\sigma^2}\|\hat{d}_j\|^2$$

and put

$$\kappa_\epsilon := \log\left(\frac{\epsilon}{(N-1)(1-\epsilon)}\right).$$

Consider the event

$$\mathcal{E} := \bigcap_{j \neq 1} \{\boldsymbol{V}_j \leq \kappa_\epsilon\}$$

so that on $\mathcal{E}$,

$$\sum_{j \neq 1} \frac{w_j}{w_1} = \sum_{j \neq 1} e^{\boldsymbol{V}_j} \leq (N-1)e^{\kappa_\epsilon} = \frac{\epsilon}{1-\epsilon}$$

which gives $w_1 \geq 1 - \epsilon$ by Eqn. (23). Since $\left\langle \boldsymbol{Z}, \hat{d}_j \right\rangle \sim \mathcal{N}\left(0, \|\hat{d}_j\|^2\right)$ we have by union bound

$$\mathbb{P}(\mathcal{E}^c) \leq \sum_{j \neq 1} \mathbb{P}\left(\left\langle \boldsymbol{Z}, \frac{\hat{d}_j}{\|\hat{d}_j\|} \right\rangle > \frac{\|\hat{d}_j\|}{2\sigma} + \frac{\sigma \kappa_\epsilon}{\|\hat{d}_j\|}\right) = \sum_{j \neq 1} \tilde{F}\left(\frac{\|\hat{d}_j\|}{2\sigma} + \frac{\sigma \kappa_\epsilon}{\|\hat{d}_j\|}\right) =: \delta.$$

Recall

$$m_\sigma(\boldsymbol{X}_\sigma) = \sum_{j=1}^N w_j(\boldsymbol{X}_\sigma, \sigma)x_j$$

so

$$\|m_\sigma(\boldsymbol{X}_\sigma) - x_1\| = \left\|\sum_{j=1}^N w_j(x_j - x_1)\right\| \leq \sum_{j \neq 1} w_j \|x_j - x_1\| \leq D(1 - w_1).$$

Thus on $\mathcal{E}$,

$$\|m_\sigma(\boldsymbol{X}_\sigma) - x_1\| \leq D\epsilon$$

which completes the proof. $\qquad\square$

For the following corollary we use the flow-matching notation in Appendix A. This result is designed to help explain the empirical result found in Figure 1 of (Bertrand et al., 2026), where in high dimensions sharp concentration of the cosine similarity between the conditional and final flow-matching vector fields was observed when using linear scheduling.

**Corollary B.2.** *Assume the setting in Lemma B.1 for $\mathcal{D} = \{x_1, \ldots, x_N\} \subset \mathbb{R}^d$, and let $\sigma = (1-t)/t$ where $t \in (0,1)$. Condition on $\boldsymbol{X} = x_1$ and suppose $x_1 \neq 0$. Pick constants $a, c > 0$ and fix $\epsilon > 0$. Then with probability at least $1 - \delta_{\epsilon,a,c}$,*

$$\frac{\langle u_t(\boldsymbol{X}_t), u_t(\boldsymbol{X}_t \mid x_1) \rangle}{\|u_t(\boldsymbol{X}_t)\| \|u_t(\boldsymbol{X}_t \mid x_1)\|} \geq 1 - \frac{2D\epsilon}{(1-t)\sqrt{d - 2\sqrt{dc} + \|x_1\|^2 - a\|x_1\|} + D\epsilon}$$

*where*

$$\delta_{\epsilon,a,c} := e^{-c} + \tilde{F}\left(\frac{a}{2}\right) + \sum_{j \neq 1} \tilde{F}\left(\frac{\|\hat{d}_j\|}{2\sigma} + \frac{\sigma \kappa_\epsilon}{\|\hat{d}_j\|}\right)$$

*and*

$$\kappa_\epsilon := \log\left(\frac{\epsilon}{(N-1)(1-\epsilon)}\right)$$

*and $\tilde{F} := 1 - F$ where $F$ is the CDF of the standard normal distribution.*

*Remark* B.3. The strong dependence on the dimension $d$ should be noted in the bound. For empirical validation we use the CIFAR-10 dataset consisting of 50K images embedded into $[-1,1]^{3072}$ in the standard manner. At $t = 0.4$ (equivalently $\sigma = 1.5$) with $\epsilon = 0.01$, $a = 8$, and $c = 5$, averaging over a random sample of 500 images, Corollary B.2 yields an average lower bound for the cosine similarity of $0.940$ with an average probability of $0.941$. Though stronger concentration is seen in Figure 1 of (Bertrand et al., 2026) for smaller values of $t$, this result conveys the effect of high dimension and dataset diameter on alignment between $u_t(\boldsymbol{X}_t)$ and $u_t(\boldsymbol{X}_t \mid x_1)$.

*Proof.* Let $\epsilon, a, c > 0$. From $\boldsymbol{X} = x_1$ we have $\boldsymbol{X}_t = (1-t)\boldsymbol{Z} + tx_1$. Plugging $\boldsymbol{X}_t$ into Eqn. (11) and Eqn. (12) and then simplifying yields

$$u_t(\boldsymbol{X}_t \mid x_1) = x_1 - \boldsymbol{Z}.$$

and

$$u_t(\boldsymbol{X}_t) = \frac{m_t(\boldsymbol{X}_t) - x_1}{1 - t} + x_1 - \boldsymbol{Z}$$

Hence after expanding

$$\begin{aligned}
\frac{\langle u_t(\boldsymbol{X}_t), u_t(\boldsymbol{X}_t \mid x_1) \rangle}{\|u_t(\boldsymbol{X}_t)\| \|u_t(\boldsymbol{X}_t \mid x_1)\|} &= \frac{\|x_1 - \boldsymbol{Z}\|^2 + \frac{1}{1-t}\langle m_t(\boldsymbol{X}_t) - x_1, x_1 - \boldsymbol{Z}\rangle}{\|\frac{1}{1-t}(m_t(\boldsymbol{X}_t) - x_1) + x_1 - \boldsymbol{Z}\| \|x_1 - \boldsymbol{Z}\|} \\
&\geq \frac{\|x_1 - \boldsymbol{Z}\| - \frac{1}{1-t}\|m_t(\boldsymbol{X}_t) - x_1\|}{\|x_1 - \boldsymbol{Z}\| + \frac{1}{1-t}\|m_t(\boldsymbol{X}_t) - x_1\|} \\
&= 1 - \frac{2\|m_t(\boldsymbol{X}_t) - x_1\|}{(1-t)\|x_1 - \boldsymbol{Z}\| + \|m_t(\boldsymbol{X}_t) - x_1\|}
\end{aligned} \tag{24}$$

Where we used Cauchy–Schwartz and the triangle inequality. Recall $x_1 \neq 0$ and define the events

$$\mathcal{F}_a := \left\{ \left\langle \boldsymbol{Z}, \frac{x_1}{\|x_1\|} \right\rangle \leq \frac{a}{2} \right\}$$

and

$$\mathcal{G}_c := \left\{ \|\boldsymbol{Z}\| \geq \sqrt{d - 2\sqrt{dc}} \right\}.$$

So that on $\mathcal{F}_a \cap \mathcal{G}_c$,

$$\|x_1 - \boldsymbol{Z}\|^2 = \|\boldsymbol{Z}\|^2 + \|x_1\|^2 - 2\langle x_1, \boldsymbol{Z}\rangle \geq d - 2\sqrt{dc} + \|x_1\|^2 - a\|x_1\|. \tag{25}$$

Note since $\langle \boldsymbol{Z}, x_1 \rangle \sim \mathcal{N}(0, \|x_1\|^2)$ we have

$$\mathbb{P}(\mathcal{F}_a^c) = \tilde{F}\left(\frac{a}{2}\right)$$

where $\tilde{F} = 1 - F$ and $F$ is the CDF of the standard normal. Furthermore by the Laurent-Massart inequality (Laurent & Massart, 2000), we have with regrettable notation

$$\mathbb{P}(\mathcal{G}_c^c) \leq e^{-c}.$$

Let $\sigma := (1 - t)/t$ so that by Eqn. (10)

$$\|m_t(\boldsymbol{X}_t) - x_1\| = \|m_\sigma(\boldsymbol{X}_\sigma) - x_1\| \tag{26}$$

where $\boldsymbol{X}_\sigma = x_1 + \sigma\boldsymbol{Z}$. By the proof of Lemma B.1 we have

$$\|m_\sigma(\boldsymbol{X}_\sigma) - x_1\| \leq D\epsilon \tag{27}$$

on an event $\mathcal{E}$ with

$$\mathbb{P}(\mathcal{E}^c) \leq \sum_{j \neq 1} \tilde{F}\left(\frac{\|\hat{d}_j\|}{2\sigma} + \frac{\sigma\kappa_\epsilon}{\|\hat{d}_j\|}\right)$$

where

$$\kappa_\epsilon := \log\left(\frac{\epsilon}{(N-1)(1-\epsilon)}\right).$$

Define the event

$$\mathcal{H} := \mathcal{E} \cap \mathcal{F}_a \cap \mathcal{G}_c$$

so that on $\mathcal{H}$, using the equality in Eqn. (26) we have by Eqn. (24), Eqn. (25) and Eqn. (27),

$$\frac{\langle u_t(\boldsymbol{X}_t), u_t(\boldsymbol{X}_t \mid x_1)\rangle}{\|u_t(\boldsymbol{X}_t)\|\|u_t(\boldsymbol{X}_t \mid x_1)\|} \geq 1 - \frac{2D\epsilon}{(1-t)\sqrt{d - 2\sqrt{dc} + \|x_1\|^2 - a\|x_1\|} + D\epsilon}$$

while using union bound and previous complement bounds yields,

$$\mathbb{P}(\mathcal{H}^c) \leq e^{-c} + \tilde{F}\left(\frac{a}{2}\right) + \sum_{j \neq 1} \tilde{F}\left(\frac{\|\hat{d}_j\|}{2\sigma} + \frac{\sigma\kappa_\epsilon}{\|\hat{d}_j\|}\right) =: \delta_{\epsilon, a, c}$$

which completes the proof. $\qquad\square$

## C. Gaussian Shell and Coverage

*Proof of Lemma 4.5.* We first recall a standard concentration inequality for chi-square random variables due to Laurent and Massart (Laurent & Massart, 2000).

**Lemma C.1.** *Let $\boldsymbol{R} \sim \chi_d^2$. Then for every $c > 0$,*

$$\mathbb{P}\left(\boldsymbol{R} - d \geq 2\sqrt{dc} + 2c\right) \leq e^{-c}, \tag{28}$$

$$\mathbb{P}\left(d - \boldsymbol{R} \geq 2\sqrt{dc}\right) \leq e^{-c}. \tag{29}$$

Now we let $\boldsymbol{S} := \|\boldsymbol{Z}\|^2 \sim \chi_d^2$. By Lemma C.1 above, for every $c \geq 0$,

$$\mathbb{P}\left(\boldsymbol{S} \geq d + 2\sqrt{cd} + 2c\right) \leq e^{-c}, \qquad \mathbb{P}\left(\boldsymbol{S} \leq d - 2\sqrt{cd}\right) \leq e^{-c}.$$

Therefore,

$$\mathbb{P}\Big(d - 2\sqrt{cd} \leq S \leq d + 2\sqrt{cd} + 2c\Big) \geq 1 - 2e^{-c}.$$

Since $\big(r_{c,d}^{\text{in}}\big)^2 = d - 2\sqrt{cd}$ and $\big(r_{c,d}^{\text{out}}\big)^2 = d + 2\sqrt{cd} + 2c$, taking square roots yields

$$\mathbb{P}\big(\|\boldsymbol{Z}\| \in [r_{c,d}^{\text{in}}, r_{c,d}^{\text{out}}]\big) \geq 1 - 2e^{-c},$$

which proves the lemma. $\qquad\square$

### C.1. Proof of Theorem 4.8

We first introduce some terminology to assist with the proof. Let $(\Omega, \mathcal{F}, \mathbb{P})$ be a probability space supporting random variables

$$\boldsymbol{X}, \boldsymbol{X}', \boldsymbol{X}_1, \dots, \boldsymbol{X}_N : \Omega \to \mathbb{R}^d, \qquad \boldsymbol{Z} : \Omega \to \mathbb{R}^d,$$

such that $\boldsymbol{X}, \boldsymbol{X}', \boldsymbol{X}_1, \dots, \boldsymbol{X}_N$ are i.i.d. with law $p$, and $\boldsymbol{Z} \sim \mathcal{N}(0, I_d)$ is independent.

We first recall the following result:

**Lemma C.2** (Example 4.1.7 in (Durrett, 2019)). *Let $\boldsymbol{A}$ and $\boldsymbol{B}$ be independent random variables and let $\varphi$ be an integrable function such that $\mathbb{E}[|\varphi(\boldsymbol{A}, \boldsymbol{B})|] < \infty$. Define*

$$g(a) := \mathbb{E}[\varphi(a, \boldsymbol{B})].$$

*Then*

$$\mathbb{E}[\varphi(\boldsymbol{A}, \boldsymbol{B}) \mid \boldsymbol{A}] = g(\boldsymbol{A}) \qquad a.s.$$

**Lemma C.3.** *For each $i = 1, \dots, N$, define*

$$I_i := \mathbf{1}_{\{\boldsymbol{X}_\sigma \in S_\sigma(\boldsymbol{X}_i)\}}.$$

*Then:*

*(i)*

$$\mathbb{E}[I_i \mid \boldsymbol{X}_\sigma] = q_\sigma(\boldsymbol{X}_\sigma) \qquad a.s.,$$

*where*

$$q_\sigma(y) = \mathbb{P}\big(y \in S_\sigma(\boldsymbol{X}')\big),$$

*and $\boldsymbol{X}'$ is an independent copy of $\boldsymbol{X}_1$.*

*(ii) The family $(I_1, \dots, I_N)$ is conditionally independent given $\boldsymbol{X}_\sigma$.*

*Consequently, conditional on $\boldsymbol{X}_\sigma$, the random variables $I_1, \dots, I_N$ are i.i.d. Bernoulli with parameter $q_\sigma(\boldsymbol{X}_\sigma)$.*

*Proof.* Since $(\boldsymbol{X}_1, \dots, \boldsymbol{X}_N)$ is independent of $(\boldsymbol{X}, \boldsymbol{Z})$ and $\boldsymbol{X}_\sigma = \boldsymbol{X} + \sigma \boldsymbol{Z}$ is measurable with respect to $(\boldsymbol{X}, \boldsymbol{Z})$, it follows that $(\boldsymbol{X}_1, \dots, \boldsymbol{X}_N)$ is independent of $\boldsymbol{X}_\sigma$.

*(i)* Define $\varphi(y, x) := \mathbf{1}_{\{y \in S_\sigma(x)\}}$, $I_i = \varphi(\boldsymbol{X}_\sigma, \boldsymbol{X}_i)$.

Since $\boldsymbol{X}_\sigma$ and $\boldsymbol{X}_i$ are independent and $\varphi$ is bounded, Lemma C.2 yields

$$\mathbb{E}[I_i \mid \boldsymbol{X}_\sigma] = \mathbb{E}[\varphi(\boldsymbol{X}_\sigma, \boldsymbol{X}_i) \mid \boldsymbol{X}_\sigma] = g(\boldsymbol{X}_\sigma),$$

where $g(y) = \mathbb{E}[\varphi(y, \boldsymbol{X}_i)] = \mathbb{P}\big(y \in S_\sigma(\boldsymbol{X}_i)\big)$.

Since $\boldsymbol{X}_i \overset{d}{=} \boldsymbol{X}'$, we have $g(y) = q_\sigma(y)$ for all $y$. Hence

$$\mathbb{E}[I_i \mid \boldsymbol{X}_\sigma] = q_\sigma(\boldsymbol{X}_\sigma) \qquad \text{a.s.}$$

*(ii)* Let $J \subset \{1, \ldots, N\}$ be finite and nonempty, and fix $\varepsilon_j \in \{0, 1\}$ for each $j \in J$. Define the measurable function

$$\Phi : \mathbb{R}^d \times (\mathbb{R}^d)^J \to \{0, 1\}, \qquad \Phi\big(y, (x_j)_{j \in J}\big) := \prod_{j \in J} \mathbf{1}_{\{\mathbf{1}_{\{y \in S_\sigma(x_j)\}} = \varepsilon_j\}}.$$

Then

$$\mathbf{1}_{\cap_{j \in J} \{I_j = \varepsilon_j\}} = \Phi\big(\boldsymbol{X}_\sigma, (\boldsymbol{X}_j)_{j \in J}\big).$$

Since $\boldsymbol{X}_\sigma$ is independent of $(\boldsymbol{X}_j)_{j \in J}$ and $\Phi$ is bounded, Lemma C.2 yields

$$\mathbb{P}\left(\bigcap_{j \in J} \{I_j = \varepsilon_j\} \,\bigg|\, \boldsymbol{X}_\sigma\right) = \mathbb{E}\big[\Phi\big(\boldsymbol{X}_\sigma, (\boldsymbol{X}_j)_{j \in J}\big) \,\big|\, \boldsymbol{X}_\sigma\big] = G(\boldsymbol{X}_\sigma),$$

where

$$G(y) = \mathbb{E}\big[\Phi\big(y, (\boldsymbol{X}_j)_{j \in J}\big)\big].$$

Using independence of $(\boldsymbol{X}_j)_{j \in J}$, we factorize

$$G(y) = \mathbb{E}\left[\prod_{j \in J} \mathbf{1}_{\{\mathbf{1}_{\{y \in S_\sigma(\boldsymbol{X}_j)\}} = \varepsilon_j\}}\right] = \prod_{j \in J} \mathbb{E}\left[\mathbf{1}_{\{\mathbf{1}_{\{y \in S_\sigma(\boldsymbol{X}_j)\}} = \varepsilon_j\}}\right]$$

$$= \prod_{j \in J} \mathbb{P}\big(\mathbf{1}_{\{y \in S_\sigma(\boldsymbol{X}_j)\}} = \varepsilon_j\big) = \prod_{j \in J} \mathbb{P}(I_j = \varepsilon_j \mid \boldsymbol{X}_\sigma = y).$$

Hence

$$\mathbb{P}\left(\bigcap_{j \in J} \{I_j = \varepsilon_j\} \,\bigg|\, \boldsymbol{X}_\sigma\right) = \prod_{j \in J} \mathbb{P}(I_j = \varepsilon_j \mid \boldsymbol{X}_\sigma),$$

which proves that $(I_1, \ldots, I_N)$ is conditionally independent given $\boldsymbol{X}_\sigma$.

Finally, by part (i),

$$\mathbb{P}(I_i = 1 \mid \boldsymbol{X}_\sigma) = \mathbb{E}[I_i \mid \boldsymbol{X}_\sigma] = q_\sigma(\boldsymbol{X}_\sigma),$$

so $(I_1, \ldots, I_N)$ are conditionally i.i.d. Bernoulli with parameter $q_\sigma(\boldsymbol{X}_\sigma)$. $\qquad\square$

**Lemma C.4.** *Fix $\sigma > 0$. For any deterministic $x, x' \in \mathbb{R}^d$, define $\boldsymbol{Y}_x = x + \sigma \boldsymbol{Z}$ with $\boldsymbol{Z} \sim \mathcal{N}(0, I_d)$. Then*

$$\mathbb{P}\big(\boldsymbol{Y}_x \in S_\sigma(x) \cap S_\sigma(x')\big) = \Phi_{d,c}\left(\frac{\|x - x'\|}{\sigma}\right).$$

*Proof.* Let $\Delta := x - x'$ and $t := \|\Delta\|/\sigma$. Recall the definitions of $r_{c,d}^{\text{in}}$ and $r_{c,d}^{\text{out}}$ given in Lemma 4.5. Note that

$$\boldsymbol{Y}_x \in S_\sigma(x) \iff \|\boldsymbol{Y}_x - x\| \in [\sigma r_{c,d}^{\text{in}}, \sigma r_{c,d}^{\text{out}}] \iff \|\boldsymbol{Z}\| \in [r_{c,d}^{\text{in}}, r_{c,d}^{\text{out}}],$$

and

$$\boldsymbol{Y}_x \in S_\sigma(x') \iff \|\boldsymbol{Y}_x - x'\| \in [\sigma r_{c,d}^{\text{in}}, \sigma r_{c,d}^{\text{out}}] \iff \|\boldsymbol{Z} + \Delta/\sigma\| \in [r_{c,d}^{\text{in}}, r_{c,d}^{\text{out}}].$$

Choose an orthogonal matrix $Q \in O(d)$ such that $Q(\Delta/\sigma) = t e_1$. Since $\boldsymbol{Z} \sim \mathcal{N}(0, I_d)$ is rotation invariant, $Q\boldsymbol{Z} \stackrel{d}{=} \boldsymbol{Z}$. Therefore,

$$\mathbb{P}\Big(\|\boldsymbol{Z}\| \in [r_{c,d}^{\text{in}}, r_{c,d}^{\text{out}}], \, \|\boldsymbol{Z} + \Delta/\sigma\| \in [r_{c,d}^{\text{in}}, r_{c,d}^{\text{out}}]\Big)$$

$$= \mathbb{P}\Big(\|Q\boldsymbol{Z}\| \in [r_{c,d}^{\text{in}}, r_{c,d}^{\text{out}}], \, \|Q\boldsymbol{Z} + Q(\Delta/\sigma)\| \in [r_{c,d}^{\text{in}}, r_{c,d}^{\text{out}}]\Big)$$

$$= \mathbb{P}\Big(\|\boldsymbol{Z}\| \in [r_{c,d}^{\text{in}}, r_{c,d}^{\text{out}}], \, \|\boldsymbol{Z} + t e_1\| \in [r_{c,d}^{\text{in}}, r_{c,d}^{\text{out}}]\Big) = \Phi_{d,c}(t).$$

$\qquad\square$

**Lemma C.5.** *Let $(\Omega, \mathcal{F}, \mathbb{P})$ be a probability space supporting random vectors $\boldsymbol{X}, \boldsymbol{X}' : \Omega \to \mathbb{R}^d$ and $\boldsymbol{Z} : \Omega \to \mathbb{R}^d$. Assume $\boldsymbol{Z} \sim \mathcal{N}(0, I_d)$ and that $\boldsymbol{Z}$ is independent of $(\boldsymbol{X}, \boldsymbol{X}')$. Define the noisy point $\boldsymbol{X}_\sigma := \boldsymbol{X} + \sigma \boldsymbol{Z}$ and the indicator*

$$U := \mathbf{1}_{\{\boldsymbol{X}_\sigma \in S_\sigma(\boldsymbol{X}) \cap S_\sigma(\boldsymbol{X}')\}} = \mathbf{1}_{\{\boldsymbol{X} + \sigma \boldsymbol{Z} \in S_\sigma(\boldsymbol{X}) \cap S_\sigma(\boldsymbol{X}')\}}.$$

*Then*

$$\mathbb{E}\big[U \mid \boldsymbol{X}, \boldsymbol{X}'\big] = \Phi_{d,c}\left(\frac{\|\boldsymbol{X} - \boldsymbol{X}'\|}{\sigma}\right) \quad a.s.$$

*Proof.* Define

$$\Psi(x, x', z) := \mathbf{1}_{\{x + \sigma z \in S_\sigma(x) \cap S_\sigma(x')\}}, \qquad (x, x', z) \in \mathbb{R}^d \times \mathbb{R}^d \times \mathbb{R}^d.$$

Then $U = \Psi(\boldsymbol{X}, \boldsymbol{X}', \boldsymbol{Z})$.

For fixed $(x, x')$, by independence and the definition of $S_\sigma(x)$,

$$\mathbb{E}_Z[\Psi(x, x', \boldsymbol{Z})] = \mathbb{P}\left(\|\boldsymbol{Z}\| \in [r_{c,d}^{\text{in}}, r_{c,d}^{\text{out}}], \left\|\boldsymbol{Z} + \tfrac{x - x'}{\sigma}\right\| \in [r_{c,d}^{\text{in}}, r_{c,d}^{\text{out}}]\right).$$

By Lemma C.4, this probability depends only on $\|x - x'\|/\sigma$, and equals

$$\Phi_{d,c}\left(\frac{\|x - x'\|}{\sigma}\right).$$

Since $\boldsymbol{Z}$ is independent of $(\boldsymbol{X}, \boldsymbol{X}')$, we conclude by Lemma C.2 again that

$$\mathbb{E}[U \mid \boldsymbol{X}, \boldsymbol{X}'] = \Phi_{d,c}\left(\frac{\|\boldsymbol{X} - \boldsymbol{X}'\|}{\sigma}\right) \quad \text{a.s.}$$

$\square$

Now we are ready to prove our theorem.

*Proof of Theorem 4.8.* By definition,

$$\mathbf{1}_{\{\boldsymbol{X}_\sigma \notin \mathcal{U}_\sigma\}} = \prod_{i=1}^{N} \mathbf{1}_{\{\boldsymbol{X}_\sigma \notin S_\sigma(\boldsymbol{X}_i)\}} = \prod_{i=1}^{N} (1 - I_i), \qquad I_i := \mathbf{1}_{\{\boldsymbol{X}_\sigma \in S_\sigma(\boldsymbol{X}_i)\}}.$$

Using Lemma C.3, we obtain almost surely

$$\mathbb{E}\big[\mathbf{1}_{\{\boldsymbol{X}_\sigma \notin \mathcal{U}_\sigma\}} \mid \boldsymbol{X}_\sigma\big] = \prod_{i=1}^{N} \mathbb{E}[1 - I_i \mid \boldsymbol{X}_\sigma] = \prod_{i=1}^{N} (1 - q_\sigma(\boldsymbol{X}_\sigma)) = (1 - q_\sigma(\boldsymbol{X}_\sigma))^N.$$

Therefore,

$$\mathbb{E}\big[\mathbf{1}_{\{\boldsymbol{X}_\sigma \in \mathcal{U}_\sigma\}} \mid \boldsymbol{X}_\sigma\big] = 1 - (1 - q_\sigma(\boldsymbol{X}_\sigma))^N.$$

Applying the tower property of conditional expectation yields

$$\mathbb{P}(\boldsymbol{X}_\sigma \in \mathcal{U}_\sigma) = \mathbb{E}\big[\mathbf{1}_{\{\boldsymbol{X}_\sigma \in \mathcal{U}_\sigma\}}\big] = \mathbb{E}\big[\mathbb{E}\big[\mathbf{1}_{\{\boldsymbol{X}_\sigma \in \mathcal{U}_\sigma\}} \mid \boldsymbol{X}_\sigma\big]\big] = \mathbb{E}\big[1 - (1 - q_\sigma(\boldsymbol{X}_\sigma))^N\big].$$

Now, we define the nearest-neighbor index

$$i^\star(\omega) := \min\left\{i : \|\boldsymbol{X}(\omega) - \boldsymbol{X}_i(\omega)\| = d_{1\text{NN}}(\boldsymbol{X}(\omega))\right\},$$

which is measurable by construction, and note that $S_\sigma(\boldsymbol{X}_{i^\star}) \subseteq \mathcal{U}_\sigma$ pointwise. Hence,

$$\mathbf{1}_{\{\boldsymbol{X}_\sigma \in \mathcal{U}_\sigma\}} \geq \mathbf{1}_{\{\boldsymbol{X}_\sigma \in S_\sigma(\boldsymbol{X}) \cap S_\sigma(\boldsymbol{X}_{i^\star})\}}.$$

Taking expectations and using the definition of probability as expectation, we obtain

$$\mathbb{P}(\boldsymbol{X}_\sigma \in \mathcal{U}_\sigma) \geq \mathbb{E}\big[\mathbf{1}_{\{\boldsymbol{X}_\sigma \in S_\sigma(\boldsymbol{X}) \cap S_\sigma(\boldsymbol{X}_{i^\star})\}}\big].$$

Now consider the conditional expectation with respect to $\boldsymbol{X}, \boldsymbol{X}_{i^\star}$. By Lemma C.5,

$$\mathbb{E}\big[\mathbf{1}_{\{\boldsymbol{X}_\sigma \in S_\sigma(\boldsymbol{X}) \cap S_\sigma(\boldsymbol{X}_{i^\star})\}} \mid \boldsymbol{X}, \boldsymbol{X}_{i^\star}\big] = \Phi_{d,c}\left(\frac{\|\boldsymbol{X} - \boldsymbol{X}_{i^\star}\|}{\sigma}\right) = \Phi_{d,c}\left(\frac{d_{1\mathrm{NN}}(\boldsymbol{X})}{\sigma}\right).$$

Applying the tower property once more gives

$$\mathbb{P}(\boldsymbol{X}_\sigma \in \mathcal{U}_\sigma) \geq \mathbb{E}\left[\Phi_{d,c}\left(\frac{d_{1\mathrm{NN}}(\boldsymbol{X})}{\sigma}\right)\right],$$

which establishes the lower bound.

Write

$$\mathbf{1}_{\{\boldsymbol{X}_\sigma \in \mathcal{U}_\sigma\}} \leq \mathbf{1}_{E^c} + \mathbf{1}_E \mathbf{1}_{\{\boldsymbol{X}_\sigma \in \mathcal{U}_\sigma\}}, \qquad E := \{\boldsymbol{X}_\sigma \in S_\sigma(\boldsymbol{X})\}.$$

Taking expectations yields

$$\mathbb{P}(\boldsymbol{X}_\sigma \in \mathcal{U}_\sigma) \leq \mathbb{P}(E^c) + \mathbb{P}(E \cap \{\boldsymbol{X}_\sigma \in \mathcal{U}_\sigma\}).$$

By Lemma 4.5, $\mathbb{P}(E^c) = \mathbb{P}(\|\boldsymbol{Z}\| \notin [r_{c,d}^{\mathrm{in}}, r_{c,d}^{\mathrm{out}}]) \leq 2e^{-c}$.

On the event $E$, if $\boldsymbol{X}_\sigma \in \mathcal{U}_\sigma$ then $\boldsymbol{X}_\sigma \in S_\sigma(\boldsymbol{X}_i)$ for at least one $i$. Thus,

$$\mathbf{1}_E \mathbf{1}_{\{\boldsymbol{X}_\sigma \in \mathcal{U}_\sigma\}} \leq \sum_{i=1}^N \mathbf{1}_E \mathbf{1}_{\{\boldsymbol{X}_\sigma \in S_\sigma(\boldsymbol{X}_i)\}}.$$

Taking expectations and using linearity, we have that

$$\mathbb{P}(E \cap \{\boldsymbol{X}_\sigma \in \mathcal{U}_\sigma\}) \leq \sum_{i=1}^N \mathbb{P}(E \cap \{\boldsymbol{X}_\sigma \in S_\sigma(\boldsymbol{X}_i)\}).$$

Fix $i$. By Lemma C.5 and the tower property,

$$\mathbb{P}(E \cap \{\boldsymbol{X}_\sigma \in S_\sigma(\boldsymbol{X}_i)\}) = \mathbb{E}\big[\mathbb{E}\big[\mathbf{1}_{\{\boldsymbol{X}_\sigma \in S_\sigma(\boldsymbol{X}) \cap S_\sigma(\boldsymbol{X}_i)\}} \mid \boldsymbol{X}, \boldsymbol{X}_i\big]\big] = \mathbb{E}\left[\Phi_{d,c}\left(\frac{\|\boldsymbol{X} - \boldsymbol{X}_i\|}{\sigma}\right)\right].$$

Since $\boldsymbol{X}_i$ is independent of $\boldsymbol{X}$ and $\boldsymbol{X}_i \overset{d}{=} \boldsymbol{X}'$, this equals $\mathbb{E}[\Phi_{d,c}(\|\boldsymbol{X} - \boldsymbol{X}'\|/\sigma)]$. Summing over $i$ and adding the bound on $\mathbb{P}(E^c)$ yields

$$\mathbb{P}(\boldsymbol{X}_\sigma \in \mathcal{U}_\sigma) \leq 2e^{-c} + N\,\mathbb{E}\left[\Phi_{d,c}\left(\frac{\|\boldsymbol{X} - \boldsymbol{X}'\|}{\sigma}\right)\right],$$

which completes the proof. $\qquad\square$

### C.2. Proof of Theorem 4.9

We formalize that, when training is effectively restricted to disjoint Gaussian shells, the denoising objective is highly non-identifiable and admits infinitely many global optima. To isolate what the objective does (and does not) constrain on each shell, we idealize the Gaussian corruption by replacing the radially concentrated law of $\boldsymbol{Z} \sim \mathcal{N}(0, I)$ with a *uniform distribution* on the corresponding shell $\mathcal{S} := \{z \in \mathbb{R}^d : r_{c,d}^{\mathrm{in}} \leq \|z\| \leq r_{c,d}^{\mathrm{out}}\}$.

**Theorem C.6.** *Fix $\sigma > 0$ and a finite dataset $\mathcal{D} = \{x_1, \ldots, x_N\} \subset \mathbb{R}^d$. Let $S_\sigma(x_i)$ be the Gaussian shells from Eqn. (6), and assume they are pairwise disjoint:*

$$S_\sigma(x_i) \cap S_\sigma(x_j) = \emptyset \qquad \text{for all } i \neq j.$$

*Let $\boldsymbol{Z} \sim \mathrm{Unif}(\mathcal{S})$, where $\mathcal{S} := \{z \in \mathbb{R}^d : r_{c,d}^{\mathrm{in}} \le \|z\| \le r_{c,d}^{\mathrm{out}}\}$, and consider the shell-only objective*

$$\mathcal{L}_\sigma(m) := \mathbb{E}_{\substack{i \sim \mathrm{Unif}([N]) \\ \boldsymbol{Z} \sim \mathrm{Unif}(\mathcal{S})}}\left[\left\|m(x_i + \sigma \boldsymbol{Z}) - x_i\right\|_2^2\right]$$

*over measurable $m : \mathbb{R}^d \to \mathbb{R}^d$. Then, there are infinitely many global minimizers: any $m$ such that*

$$m(y) = x_i \quad \text{for all } y \in S_\sigma(x_i), \ \ i = 1, \dots, N$$

*is a minimizer, and $m$ can be defined arbitrarily on $\mathbb{R}^d \setminus \bigcup_{i=1}^N S_\sigma(x_i)$.*

*Proof.* We first note that $\mathcal{L}_\sigma(m) \ge 0$ for every measurable $m$.

Let $m$ be any measurable map satisfying

$$m(y) = x_i \quad \text{for all } y \in S_\sigma(x_i), \ \ i = 1, \dots, N.$$

Then for every $i \in [N]$ and every $\boldsymbol{Z}$ with $\|\boldsymbol{Z}\| \in [r_{c,d}^{\mathrm{in}}, r_{c,d}^{\mathrm{out}}]$, we again have $x_i + \sigma \boldsymbol{Z} \in S_\sigma(x_i)$ and therefore

$$m(x_i + \sigma \boldsymbol{Z}) = x_i, \qquad \|m(x_i + \sigma \boldsymbol{Z}) - x_i\|_2^2 = 0.$$

Taking expectation shows $\mathcal{L}_\sigma(m) = 0$, hence every such $m$ is a global minimizer. $\qquad\square$

## D. Denoiser Behavior at Large Noise Levels

In this section, we prove Theorem 4.11. Instead of working with the $\sigma$ parameter directly, we first work with a parameter $t \in [0, 1]$ from flow matching as already described in Appendix A.

We use $\boldsymbol{X} \sim p$ to denote a random variable. Since we have assumed $p$ to have bounded support, we further assume that there is $R > 0$ such that $\mathrm{supp}(p) \subset B_R(0)$. We next rewrite the denoiser $m_t$ (cf. Eqn. (8)) as follows.

For $x \in \mathbb{R}^d$ and $t \in (-1, 1)$, define

$$W_t(y; x) := \exp\left(-\frac{\|x - ty\|^2}{2(1-t)^2}\right), \qquad D_t(x) := \mathbb{E}[W_t(\boldsymbol{X}; x)], \qquad N_t(x) := \mathbb{E}[\boldsymbol{X}\, W_t(\boldsymbol{X}; x)]. \tag{30}$$

Then, we have that for any $t \in [0, 1)$

$$m_t(x) = \mathbb{E}[\boldsymbol{X} \mid \boldsymbol{X}_t = x] = \frac{N_t(x)}{D_t(x)}. \tag{31}$$

Note that although $\mathbb{E}[\boldsymbol{X} \mid \boldsymbol{X}_t = x]$ is only defined for $t \in [0, 1)$, the formula $N_t(x)/D_t(x)$ is well-defined for all $t \in (-1, 1)$, and we will work with this extension of $m_t$ in the proof. Note that $0 < W_t(y; x) \le 1$ for all $t \in (-1, 1)$, so $D_t(x) \in (0, 1]$.

We begin by establishing a collection of lemmas that will be used in the proof of Theorem 4.11.

**Lemma D.1.** *Fix $\delta \in (0, 1)$. Then for every $x \in \mathbb{R}^d$ the functions*

$$t \longmapsto D_t(x), \qquad t \longmapsto N_t(x)$$

*are $C^2$ on $[-\delta, \delta]$. Moreover, for $k = 0, 1, 2$,*

$$(x, t) \longmapsto \partial_t^k D_t(x), \qquad (x, t) \longmapsto \partial_t^k N_t(x)$$

*are continuous on $\mathbb{R}^d \times [-\delta, \delta]$. Moreover,*

$$(x, t) \longmapsto \partial_t^2 m_t(x)$$

*is continuous on $\mathbb{R}^d \times [-\delta, \delta]$.*

*Proof.* Write

$$W_t(y; x) = e^{F(t,x,y)}, \qquad F(t, x, y) = -\frac{\|x - ty\|^2}{2(1-t)^2}.$$

The function $F$ is smooth on $(-1, 1) \times \mathbb{R}^d \times \mathbb{R}^d$. For $k = 0, 1, 2$,

$$\partial_t^k W_t(y; x) = P_k(x, y, t) W_t(y; x),$$

where $P_k$ is rational in $t$ with the denominators of the form $(1-t)^{-m}$, and polynomial in $(x, y)$.

Fix $a > 0$. Since $|t| \leq \delta < 1$, the factors $(1-t)^{-m}$ are uniformly bounded. Hence there exists $C > 0$ such that

$$\sup_{\substack{\|x\| \leq a \\ |t| \leq \delta \\ \|y\| \leq R}} |\partial_t^k W_t(y; x)| \leq C.$$

Because $\|\boldsymbol{X}\| \leq R$ a.s., we obtain

$$|\partial_t^k W_t(\boldsymbol{X}; x)| \leq C, \qquad \|\boldsymbol{X}\, \partial_t^k W_t(\boldsymbol{X}; x)\| \leq RC \quad \text{a.s.}$$

By dominated convergence,

$$\partial_t^k D_t(x) = \mathbb{E}[\partial_t^k W_t(\boldsymbol{X}; x)], \qquad \partial_t^k N_t(x) = \mathbb{E}[\boldsymbol{X}\, \partial_t^k W_t(\boldsymbol{X}; x)],$$

and these depend continuously on $(x, t)$. We have $m_t(x) = N_t(x)/D_t(x)$ and $D_t(x) > 0$. Since $N_t$ and $D_t$ are $C^2$ in $t$ with continuous derivatives. The quotient rule gives the claim that $\partial_t^2 m_t(x)$ is continuous on $\mathbb{R}^d \times [-\delta, \delta]$. $\qquad\square$

Now let $\mu := \mathbb{E}[\boldsymbol{X}]$ and $\Sigma := \mathrm{Cov}(\boldsymbol{X})$. Let $G = \mathcal{N}(\mu, \Sigma)$ and define $m_t^G(x)$ by the same formula Eqn. (30)–Eqn. (31) but with expectation taken under $G$.

Recall from Eqn. (7) that in the Gaussian case $Y \sim \mathcal{N}(\mu, \Sigma)$ with $\Sigma \succeq 0$, we have a closed-form expression for $m_\sigma^G(x)$. Using Eqn. (9), this yields the following closed form for $m_t^G$ for every $t \in (-1, 1)$:

$$m_t^G(x) = \mu + t\,\Sigma\big(t^2\Sigma + (1-t)^2 I\big)^{-1}(x - t\,\mu). \tag{32}$$

This formula follows from the conditional mean formula for multivariate Gaussian distributions.

**Lemma D.2.** *For every fixed $x \in \mathbb{R}^d$,*

$$m_0(x) = \mu, \qquad \frac{d}{dt}m_t(x)\bigg|_{t=0} = \Sigma x.$$

*The same identities hold for $m_t^G(x)$ (with the same $\mu, \Sigma$).*

*Proof.* At $t = 0$, the weight is constant in $y$:

$$W_0(y; x) = \exp(-\|x\|^2/2) =: W_0(x).$$

Therefore $N_0(x) = W_0(x)\mathbb{E}[\boldsymbol{X}] = W_0(x)\mu$ and $D_0(x) = W_0(x)$, giving $m_0(x) = \mu$.

For the first derivative, differentiate $m_t = N_t/D_t$

$$m_t' = \frac{N_t'}{D_t} - \frac{N_t}{D_t}\frac{D_t'}{D_t}. \tag{33}$$

It suffices to compute $N_0', D_0'$. A direct calculation yields

$$\partial_t \log W_t(y; x)|_{t=0} = -\|x\|^2 + \langle x, y\rangle, \qquad \partial_t W_t(y; x)|_{t=0} = W_0(x)\left(-\|x\|^2 + \langle x, y\rangle\right).$$

Hence

$$D_0'(x) = W_0(x)\mathbb{E}[-\|x\|^2 + \langle x, \boldsymbol{X}\rangle], \qquad N_0'(x) = W_0(x)\mathbb{E}\big[\boldsymbol{X}(-\|x\|^2 + \langle x, \boldsymbol{X}\rangle)\big].$$

Substituting into the Eqn. (33) at $t = 0$ and using $m_0(x) = \mu$ gives

$$m_0'(x) = \mathbb{E}\big[\boldsymbol{X}\langle x, \boldsymbol{X}\rangle\big] - \mu\,\mathbb{E}[\langle x, \boldsymbol{X}\rangle] = \mathbb{E}\big[(\boldsymbol{X} - \mu)\langle x, (\boldsymbol{X} - \mu)\rangle\big] = \Sigma x.$$

The same computation applies to $G = \mathcal{N}(\mu, \Sigma)$, so the identities hold for $m_t^G$ as well. $\qquad\square$

**Theorem D.3.** *Assume that* $\mathrm{supp}(p) \subset B_R(0)$. *Then for all* $\delta \in (0, 1)$, *there exists a bounded function*

$$\tilde{H} : \overline{B_R(0)} \times [-\delta, \delta] \to \mathbb{R}^d$$

*such that for all* $x \in \overline{B_R(0)}$ *and all* $|t| \leq \delta$,

$$m_t(x) = m_t^G(x) + t^2 \tilde{H}(x, t).$$

*Proof.* For $x \in \overline{B_R(0)}$, define

$$f_x(t) := m_t(x) - m_t^G(x).$$

By Lemma D.1, the function $f_x$ is of class $C^2$ in $t$. Moreover, Lemma D.2 implies that

$$f_x(0) = f_x'(0) = 0.$$

Again by Lemma D.1, the map $(x, t) \mapsto f_x''(t)$ is continuous on $\overline{B_R(0)} \times [-\delta, \delta]$.

Define

$$\tilde{H}(x, t) := \int_0^1 (1 - s) f_x''(st) \, ds.$$

Then the map $(x, t, s) \mapsto (1 - s) f_x''(st)$ is continuous on the compact set

$$\overline{B_R(0)} \times [-\delta, \delta] \times [0, 1],$$

and therefore $\tilde{H}$ is continuous on $\overline{B_R(0)} \times [-\delta, \delta]$. By compactness, $\tilde{H}$ is uniformly bounded on this set.

Applying Taylor's theorem with integral remainder, we obtain for all $|t| \leq \delta$,

$$f_x(t) = t^2 \int_0^1 (1 - s) f_x''(st) \, ds = t^2 \tilde{H}(x, t).$$

Consequently,

$$m_t(x) = m_t^G(x) + t^2 \tilde{H}(x, t),$$

which completes the proof. $\qquad\square$

Now, we are going to translate the second-order expansion in the small parameter $t$ into an asymptotic expansion in terms of the scale parameter $\sigma$.

*Proof of Theorem 4.11.* Consider the transformation $\sigma(t) = \frac{1-t}{t}$ (then $t(\sigma) = \frac{1}{1+\sigma}$) and define $\delta = \frac{1}{1+\sigma_0}$. In this way, for any $\sigma \geq \sigma_0$, we have $|t| \leq \delta$.

Consider the function $\tilde{H}$ from Theorem D.3. For $x \in \overline{B_R(0)}$ and $\sigma \geq \sigma_0$, define

$$H(x, \sigma) := \tilde{H}(t(\sigma)x, t(\sigma)).$$

This is well-defined because $t(\sigma) \in [-\delta, \delta]$ for $\sigma \geq \sigma_0$ and $t(\sigma)x \in \overline{B_R(0)}$ since $\|t(\sigma)x\| \leq \|x\| \leq R$. Since $\tilde{H}$ is bounded, $H$ is bounded as well on $\overline{B_R(0)} \times [\sigma_0, \infty)$.

Now, by Eqn. (9) we have that

$$m_\sigma(x) = m_t(tx), \qquad m_\sigma^G(x) = m_t^G(tx).$$

By Theorem D.3, for any $|t| \leq \delta$ and $x \in \overline{B_R(0)}$,

$$m_t(x) = m_t^G(x) + t^2 \tilde{H}(x, t).$$

Replacing $x$ by $tx$, we obtain

$$m_t(tx) = m_t^G(tx) + t^2 \tilde{H}(tx, t).$$

By the relationship between $m_t$ and $m_\sigma$ (Eqn. (9)) as well as the definition of $H$, we have

$$m_\sigma(x) = m_\sigma^G(x) + \frac{1}{(1 + \sigma)^2} H(x, \sigma),$$

This proves the theorem. $\qquad\square$

# E. Sensitivity Localization

We analyze the case when the data distribution $p$ is a Gaussian distribution $p = \mathcal{N}(\mu, \Sigma)$ on $\mathbb{R}^d$ to establish a spatial sensitivity decay control. Recall from Eqn. (7) that the optimal denoiser for $p$ is given by the closed-form

$$m_\sigma(x) = \mu + \Sigma(\Sigma + \sigma^2 I)^{-1}(x - \mu).$$

In the following we only consider the case of a single channel. The multivariate case can be handled by the same arguments by treating each channel separately. We think of $d = r \times r$ where $r$ is the resolution and then each dimension represents a single-valued pixel. To quantify how much the denoiser at pixel $i$ depends on observation $x_j$, we study the Jacobian $\nabla_x m_\sigma(x)$. By Tweedie's formula (Efron, 2011), we have that

$$\nabla_x m_\sigma(x) = \frac{1}{\sigma^2} \operatorname{Cov}(\boldsymbol{X} \mid \boldsymbol{X}_\sigma = x) = \Sigma(\Sigma + \sigma^2 I)^{-1}.$$

The expression for $\nabla_x m_\sigma(y)$ was derived in (Lukoianov et al., 2025), who also observed that $\nabla_x m_\sigma(y) \to I$ as $\sigma \to 0$. However, convergence to the identity does not by itself characterize how the off-diagonal entries decay with spatial distance.

We now make the spatial decay claim precise under cyclic stationary.

**Lemma E.1.** *For any $i \neq j$, we have that*

$$\left| \frac{\partial [m_\sigma]_i}{\partial x_j} \right| = \sigma^2 \left| [(\Sigma + \sigma^2 I)^{-1}]_{ij} \right|, \tag{34}$$

*where $[m_\sigma]_i$ denotes the $i$-th component of the vector $m_\sigma$.*

*Proof.* The identity (34) follows from the simple identity $\Sigma(\Sigma + \sigma^2 I)^{-1} = I - \sigma^2(\Sigma + \sigma^2 I)^{-1}$ and the fact that $I_{ij} = 0$ for $i \neq j$. $\qquad \square$

**Definition E.2** (Circulant covariance). *A matrix $\Sigma \in \mathbb{R}^{d \times d}$ is called *circulant* if there exists a vector $c = [c_0, \ldots, c_{d-1}]^T$ such that*

$$\Sigma_{ij} = c_{(i-j) \bmod d}, \qquad i, j \in \{0, \ldots, d-1\}.$$

Equivalently, each row of $\Sigma$ is a cyclic shift of its first row.

**Lemma E.3.** *Let $\Sigma \in \mathbb{R}^{d \times d}$ be circulant and let $\omega = e^{2\pi i/d}$. Then $\Sigma$ is diagonalized by the discrete Fourier basis $(\omega^{jk})_{j,k=0}^{d-1}$, with eigenvalues $(\lambda_k)_{k=0}^{d-1}$. In particular, for any function $f$ applied spectrally to $\Sigma$, the matrix $f(\Sigma)$ is circulant and its entries depend only on the wrap-around distance $\operatorname{dist}(i, j) := \min\{|i - j|, d - |i - j|\}$.*

*Proof sketch.* This is a classical property of circulant matrices: they are simultaneously diagonalized by the discrete Fourier transform, and functional calculus preserves circulant structure. See (Gray, 2006) for a detailed treatment of the spectral theory of circulant and Toeplitz matrices. $\qquad \square$

**Theorem E.4.** *Assume $\Sigma$ is circulant. Let $\omega = e^{2\pi i/d}$ and let $(\lambda_k)_{k=0}^{d-1}$ be the eigenvalues of $\Sigma$ in the discrete Fourier basis (Lemma E.3). Define*

$$h_k := \frac{\sigma^2}{\lambda_k + \sigma^2}, \qquad q_r := \frac{1}{d} \sum_{k=0}^{d-1} h_k \, \omega^{rk}, \qquad r \in \{0, \ldots, d-1\}.$$

*For all $i \neq j$, let $n := \min\{|i - j|, d - |i - j|\}$ be the wrap-around distance. Then,*

$$\left| \frac{\partial [m_\sigma]_i}{\partial x_j} \right| = |q_n|.$$

*Moreover, for all $n \in \{1, \ldots, \lfloor d/2 \rfloor\}$,*

$$\left| \frac{\partial [m_\sigma]_i}{\partial x_j} \right| \leq \frac{\operatorname{TV}(h)}{4n}, \qquad \operatorname{TV}(h) := \sum_{k=0}^{d-1} |h_{k+1} - h_k| \quad (\text{indices mod } d). \tag{35}$$

*Proof.* Define the matrix

$$Q_\sigma := \sigma^2 (\Sigma + \sigma^2 I)^{-1}.$$

Since $\Sigma$ is circulant, so is $\Sigma + \sigma^2 I$, and hence $(\Sigma + \sigma^2 I)^{-1}$ and $Q_\sigma$ are circulant as well (Lemma E.3 applied to the spectral function $f(t) = 1/(t + \sigma^2)$). In the DFT basis, the eigenvalues of $Q_\sigma$ are

$$h_k = \frac{\sigma^2}{\lambda_k + \sigma^2}, \qquad k = 0, \dots, d-1.$$

Therefore $Q_\sigma$ has circulant kernel $(q_r)_{r=0}^{d-1}$ given by the inverse DFT:

$$[Q_\sigma]_{ij} = q_{(i-j) \bmod d}, \qquad q_r = \frac{1}{d} \sum_{k=0}^{d-1} h_k \, \omega^{rk}.$$

Now fix $i \neq j$ and let $n := \min\{|i - j|, \, d - |i - j|\}$. By Lemma E.1,

$$\left| \frac{\partial [m_\sigma]_i}{\partial x_j} \right| = \sigma^2 \left| [(\Sigma + \sigma^2 I)^{-1}]_{ij} \right| = |[Q_\sigma]_{ij}| = |q_n|,$$

which proves the first claim.

For the decay bound, let $(\Delta h)_k := h_{k+1} - h_k$ (indices modulo $d$). For $n \not\equiv 0 \pmod{d}$, a cyclic summation-by-parts identity (discrete integration by parts) yields

$$q_n = \frac{\omega^n}{d(1 - \omega^n)} \sum_{k=0}^{d-1} (\Delta h)_k \, \omega^{nk}.$$

Taking absolute values and using the triangle inequality gives

$$|q_n| \leq \frac{\sum_{k=0}^{d-1} |\Delta h_k|}{d \, |1 - \omega^n|} = \frac{\mathrm{TV}(h)}{d \, |1 - \omega^n|}.$$

Finally, for $1 \leq n \leq d/2$ we have

$$|1 - \omega^n| = 2|\sin(\pi n/d)| \geq \frac{4n}{d},$$

and substituting into the previous display yields (35). $\qquad\square$

## F. Experiment Details

### F.1. Sampling Schedule

Unless otherwise specified, the sampling schedule used in our experiments is the polynomial sampling schedule from (Karras et al., 2022):

$$\sigma_i = \left( \sigma_{\max}^{1/\rho} + \frac{i}{N-1} \left( \sigma_{\min}^{1/\rho} - \sigma_{\max}^{1/\rho} \right) \right)^\rho, \tag{36}$$

for $i = 0, \dots, N-1$, where $\sigma_{\max} = 80$, $\sigma_{\min} = 0.002$ and $N = 18$ is a popular choice for sampling on CIFAR-10.

### F.2. One-Step Memorization Showcase

*Table 2.* **Time Parameterization Conventions for EDM Sampling.** EDM uses 18 sampling steps for CIFAR-10 with noise scales $\sigma$. Red denotes initial (high noise) steps, blue denotes middle steps, green denotes end (low noise) steps.

| $\sigma$ | 80.00 | 57.59 | 40.79 | 28.37 | 19.35 | 12.91 | 8.40 | 5.32 | 3.26 | 1.92 | 1.09 | 0.59 | 0.30 | 0.14 | 0.06 | 0.02 | 0.01 | 0.00 |
|---|---|---|---|---|---|---|---|---|---|---|---|---|---|---|---|---|---|---|

We use the standard 18 steps Table 2 of EDM for sampling with Euler method, the noise level $\sigma$ and step translation can be seen in Section F.2. Figures 6–11 visualize one-step denoising output at six different noise levels spanning the full diffusion schedule. Each figure shows, from top to bottom: the initial clean test image, noisy input, **EDM-1K** output followed by its 1-NN in the training set, **EDM-50K** output followed by its 1-NN, and **EMP-1K** output followed by its 1-NN.

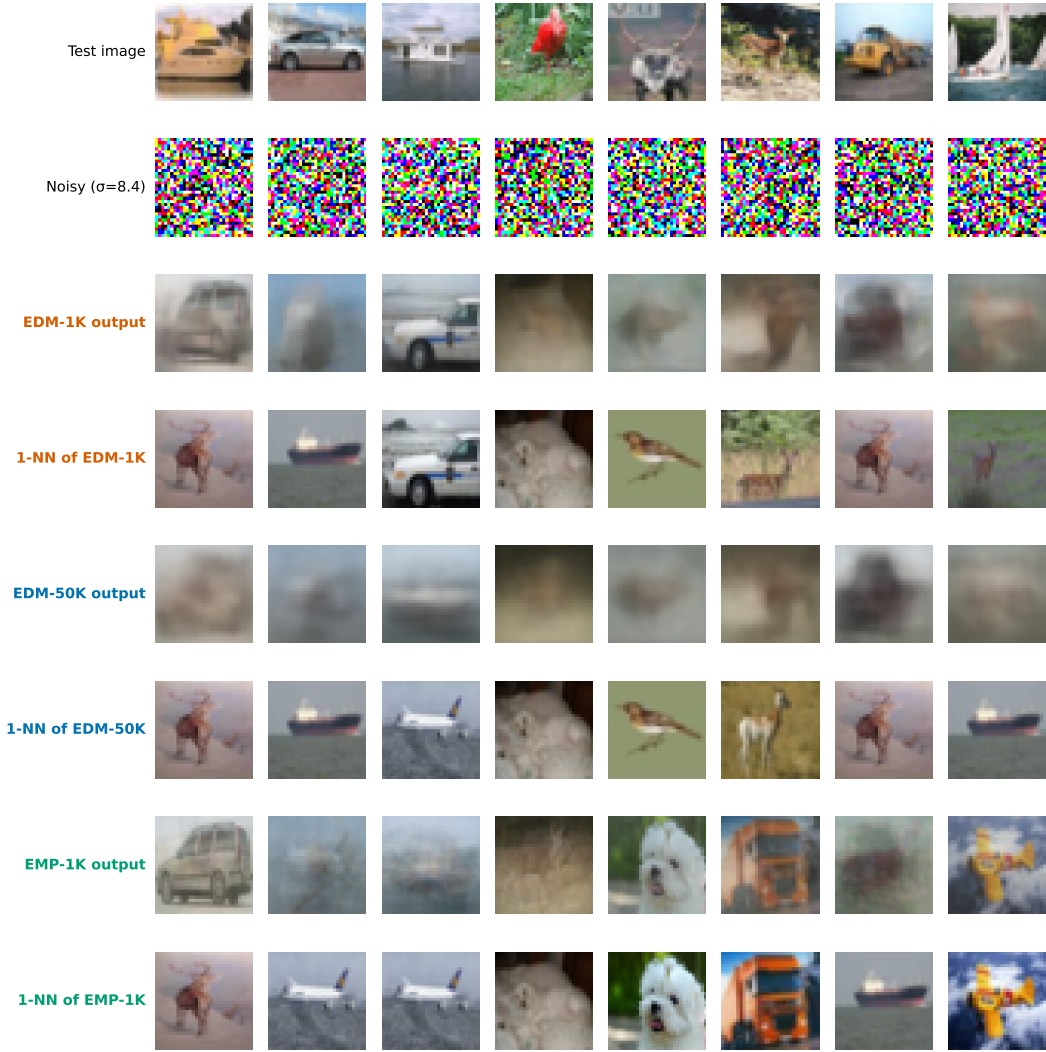

*Figure 6.* **One-Step Denoising at** $\sigma = 8.4$**.** From top: initial test image, noisy input, **EDM-1K** output, 1-NN of **EDM-1K**, **EDM-50K** output, 1-NN of **EDM-50K**, **EMP-1K** output, 1-NN of **EMP-1K**. At this high noise level, all denoisers perform similarly with no memorization.

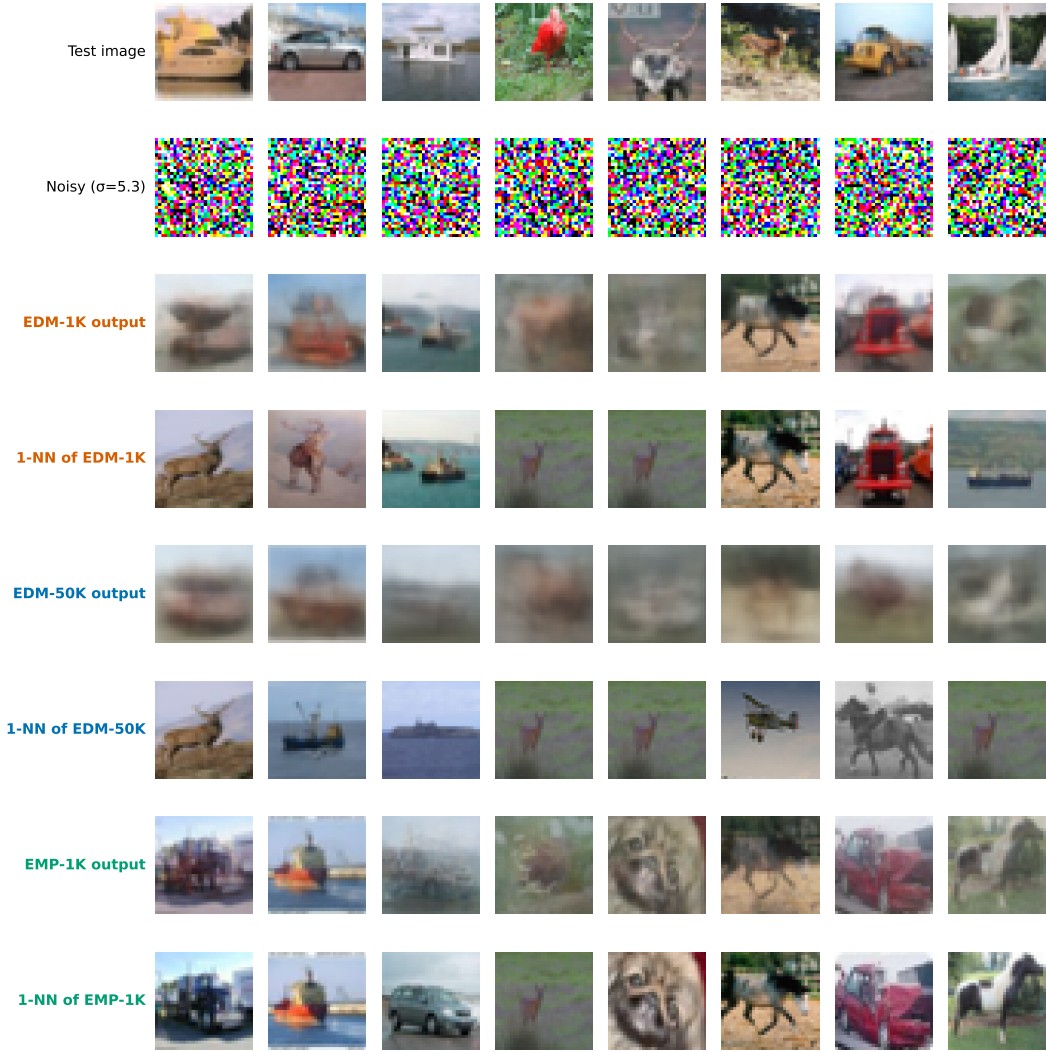

*Figure 7.* **One-Step Denoising at** $\sigma = 5.3$**.** From top: initial test image, noisy input, **EDM-1K** output, 1-NN of **EDM-1K**, **EDM-50K** output, 1-NN of **EDM-50K**, **EMP-1K** output, 1-NN of **EMP-1K**. Denoising begins to show model differences as **EDM-1K** starts exhibiting some preference for training images.

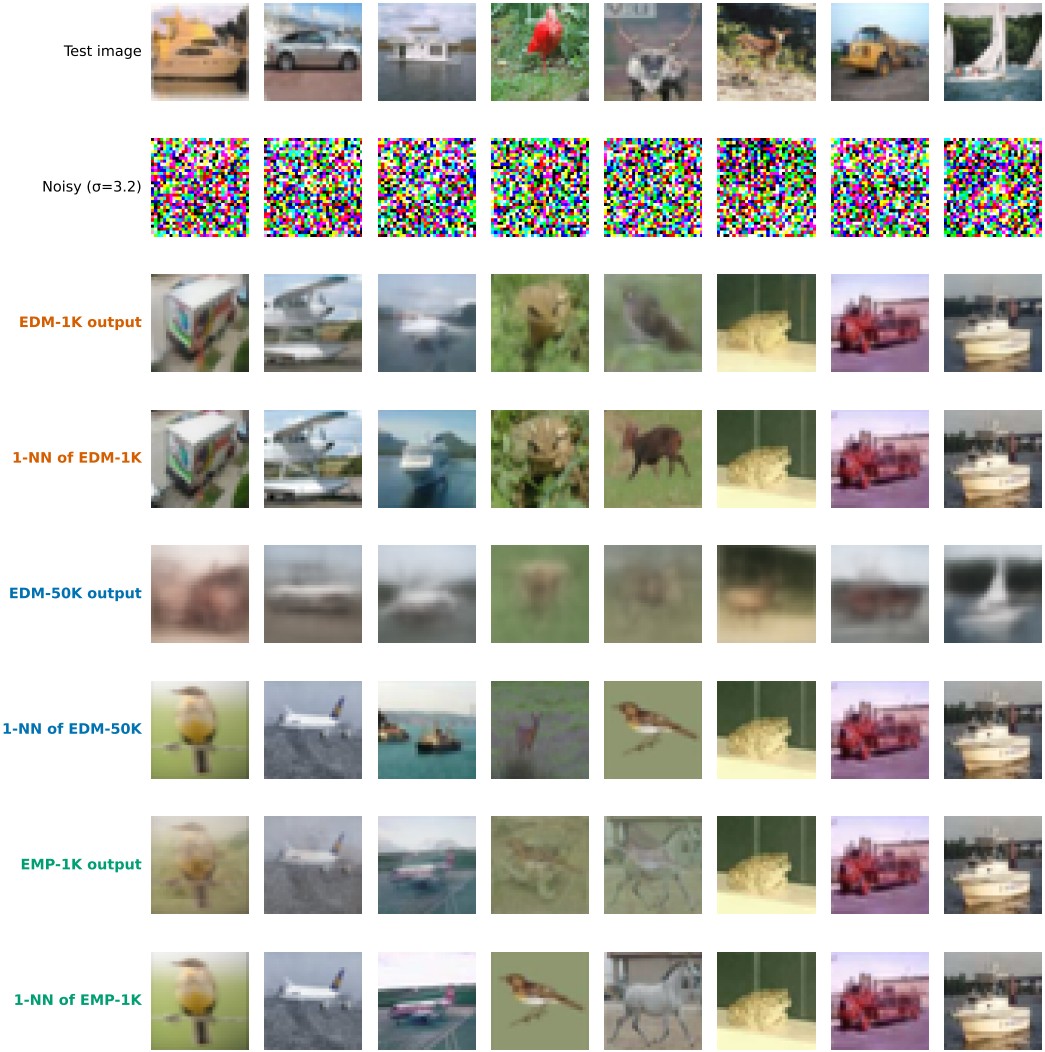

*Figure 8.* **One-Step Denoising at** $\sigma = 3.2$**.** From top: initial test image, noisy input, **EDM-1K** output, 1-NN of **EDM-1K**, **EDM-50K** output, 1-NN of **EDM-50K**, **EMP-1K** output, 1-NN of **EMP-1K**. **EDM-1K** memorization becomes pronounced, with outputs matching training nearest neighbors. This is near the peak memorization region.

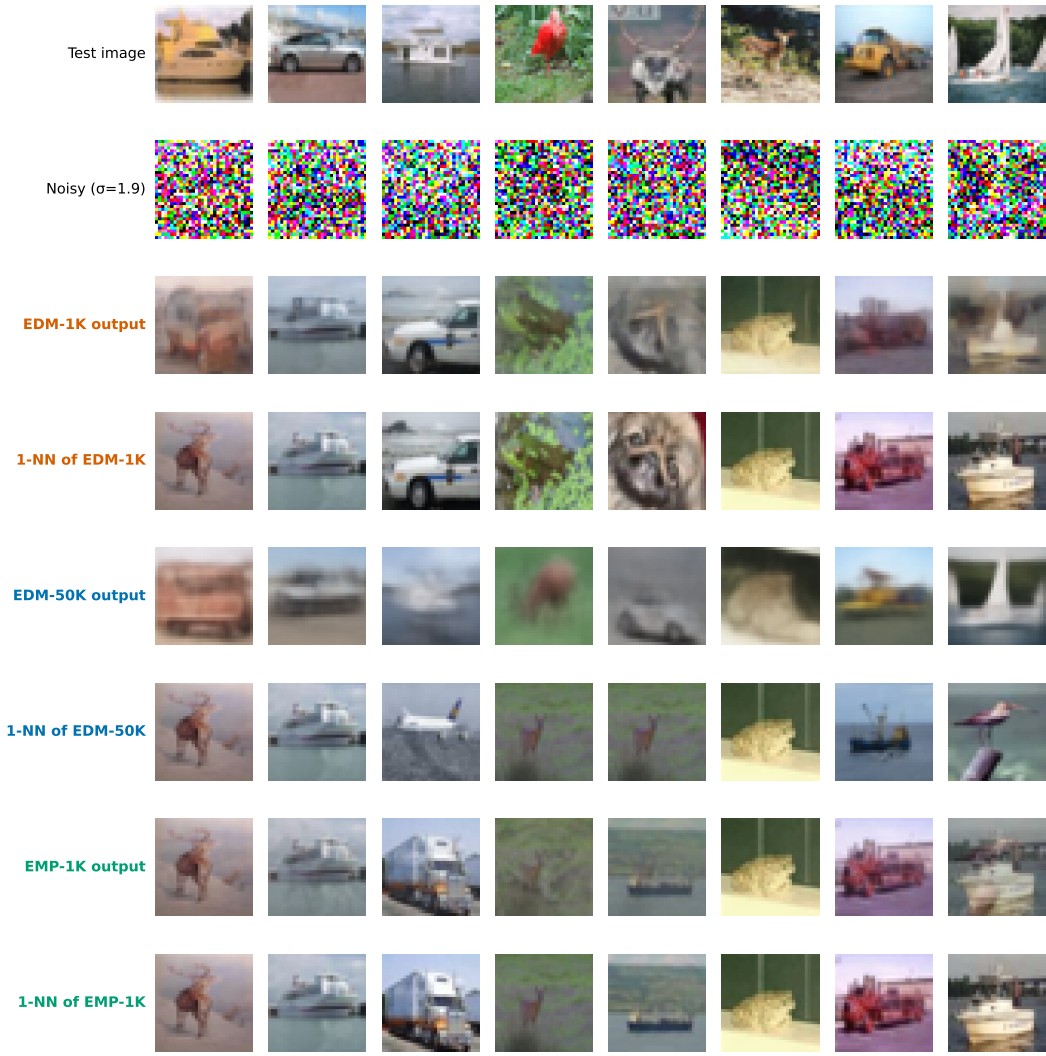

*Figure 9.* **One-Step Denoising at** $\sigma = 1.9$**.** From top: initial test image, noisy input, **EDM-1K** output, 1-NN of **EDM-1K**, **EDM-50K** output, 1-NN of **EDM-50K**, **EMP-1K** output, 1-NN of **EMP-1K**. **EDM-1K** outputs are much closer to training images than **EDM-50K** at this critical mid-$\sigma$ level.

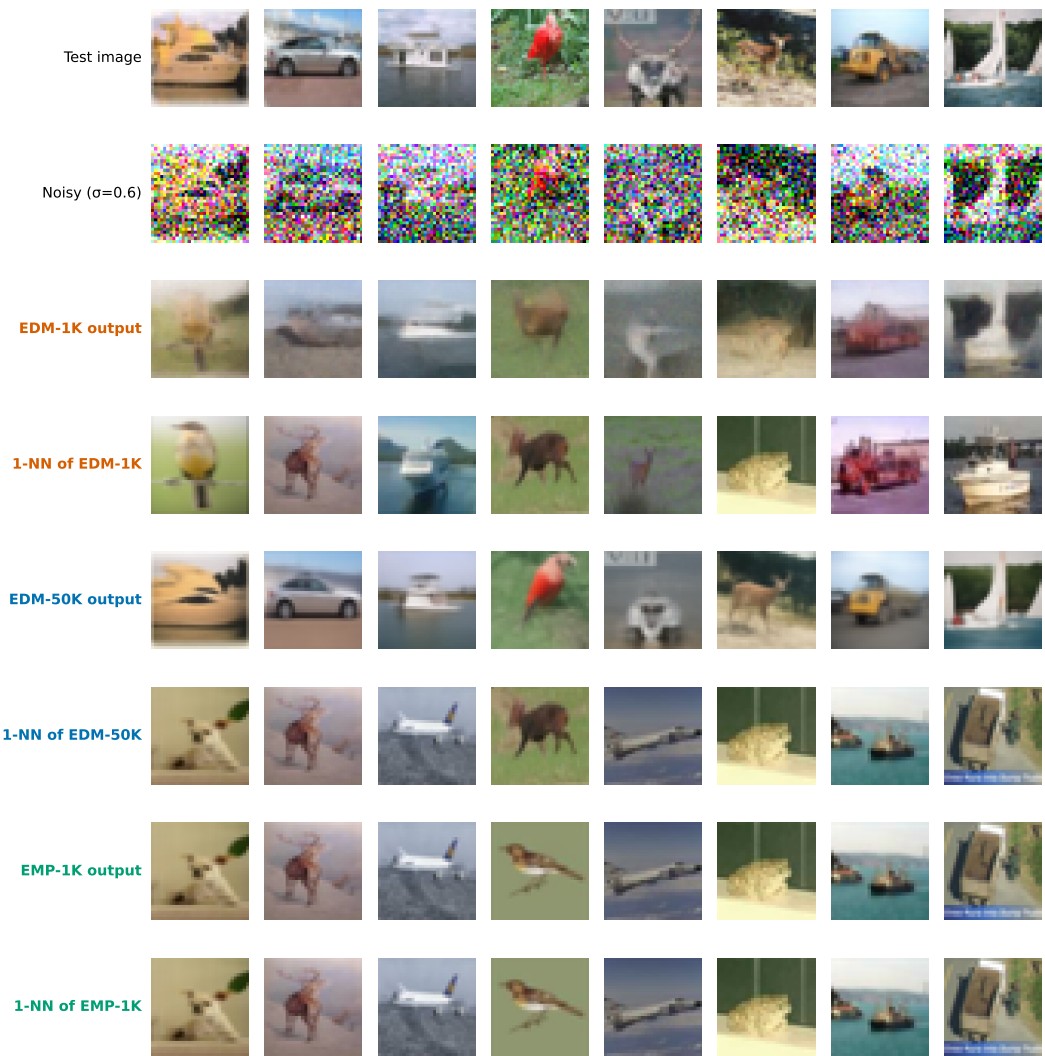

*Figure 10.* **One-Step Denoising at** $\sigma = 0.6$**.** From top: initial test image, noisy input, **EDM-1K** output, 1-NN of **EDM-1K**, **EDM-50K** output, 1-NN of **EDM-50K**, **EMP-1K** output, 1-NN of **EMP-1K**. At this lower noise level, **EDM-1K** memorization weakens while **EMP-1K** shows stronger nearest-neighbor retrieval.

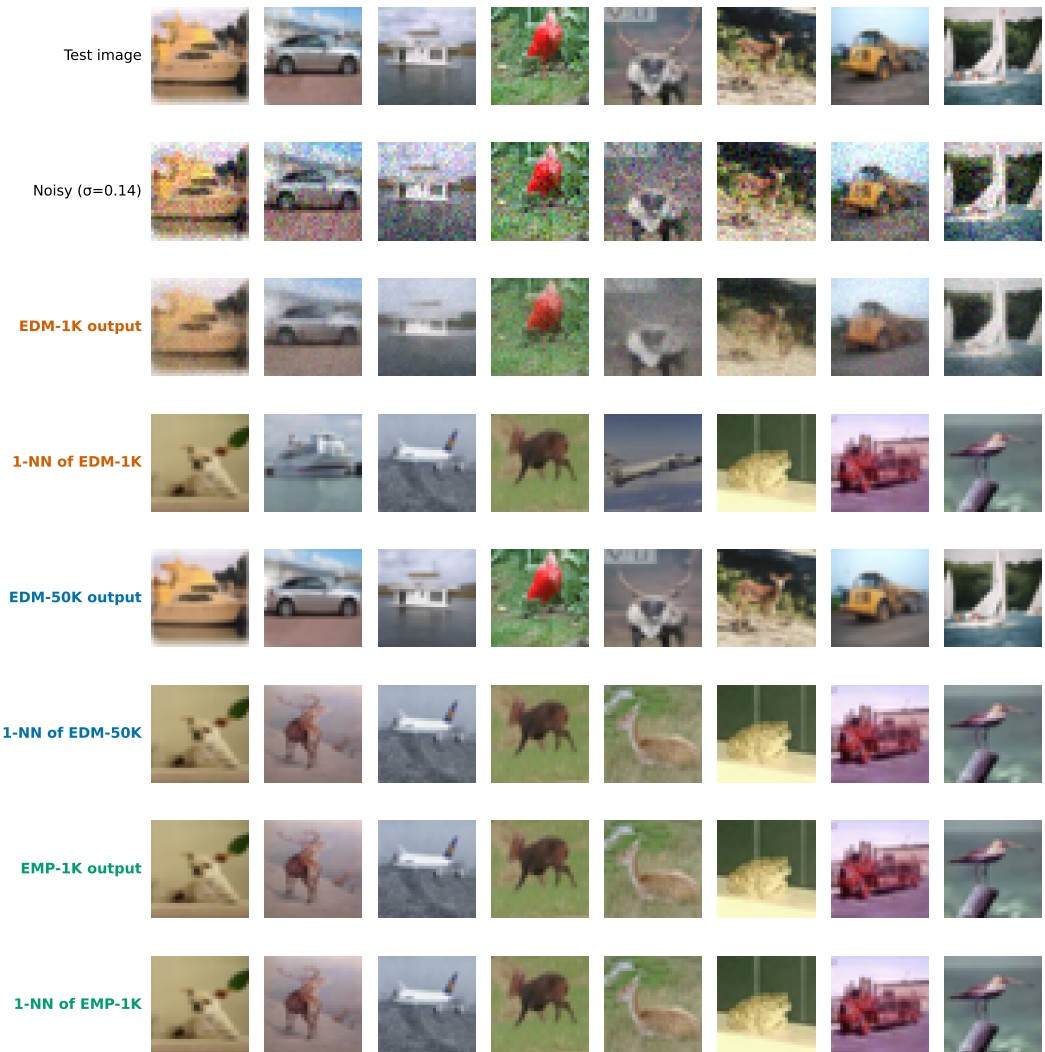

*Figure 11.* **One-Step Denoising at** $\sigma = 0.14$**.** From top: initial test image, noisy input, **EDM-1K** output, 1-NN of **EDM-1K**, **EDM-50K** output, 1-NN of **EDM-50K**, **EMP-1K** output, 1-NN of **EMP-1K**. At this low noise level, **EDM-1K** shows minimal memorization while **EMP-1K** exhibits nearly perfect nearest-neighbor behavior.

## F.3. Denoiser Swapping Experiments

To understand which noise levels are critical for sample quality, we perform denoiser swapping experiments. We generate samples using hybrid schedules that switch between **EDM-1K** and **EDM-50K** at different $\sigma$ regions (large: $\sigma > 8.4$, medium: $\sigma \in [0.14, 8.4]$, small: $\sigma < 0.14$).

Figure 12 shows representative samples from each condition. Key findings:

- Swapping to **EDM-50K** in the **medium region** (steps 6–13) produces the largest quality improvement when starting from **EDM-1K**.

- Swapping to **EDM-1K** in the **medium region** causes the largest quality degradation when starting from **EDM-50K**.

- large and small region swaps have minimal impact on final sample quality.

This confirms that the mid-$\sigma$ region is where memorization most impacts generation quality, aligning with our one-step memorization analysis. Table 1 in the main text quantifies the memorization rates across all conditions.

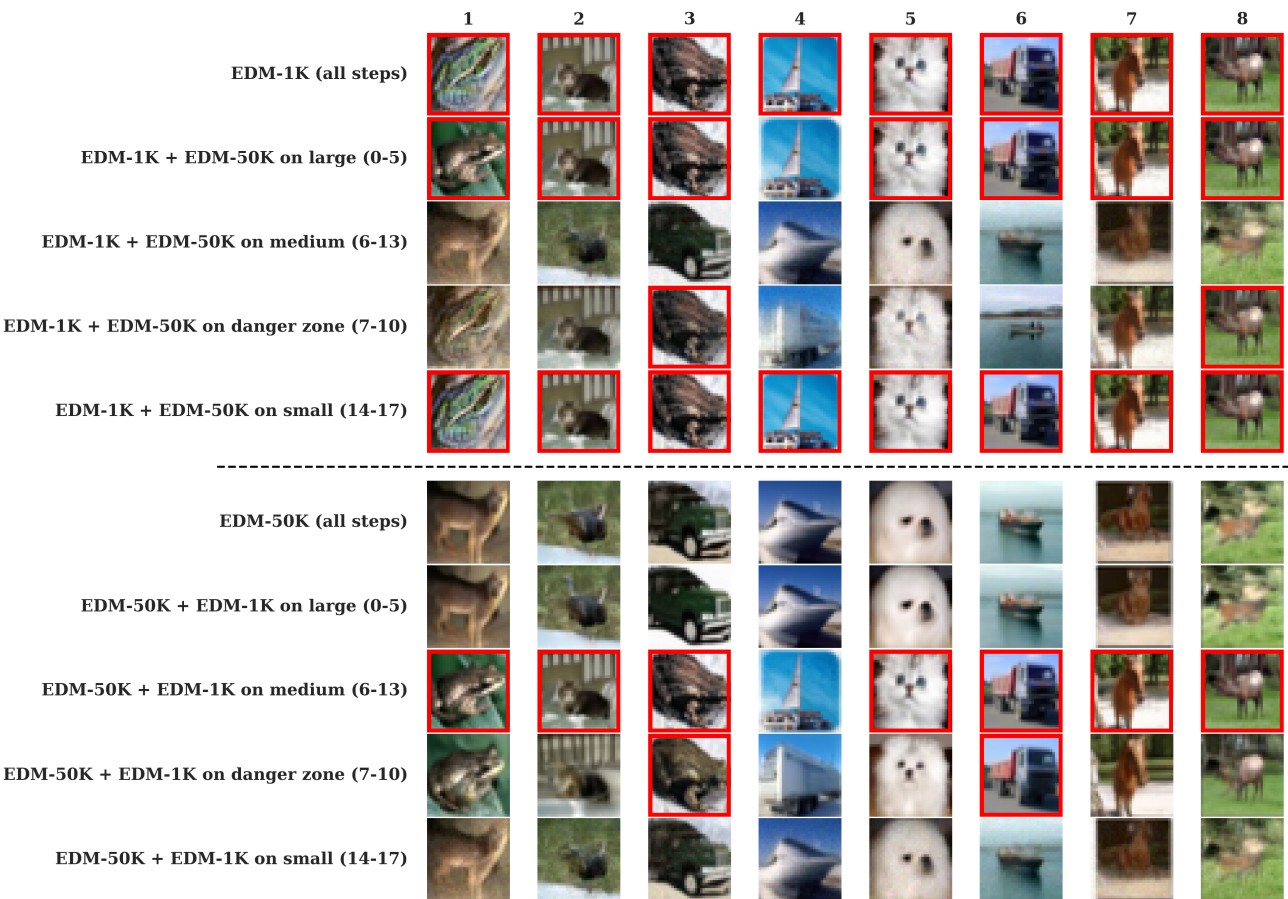

*Figure 12.* **Denoiser Swapping Samples.** Top half: **EDM-1K** base with **EDM-50K** swapped in different regions. Bottom half: **EDM-50K** base with **EDM-1K** swapped in. Red boxes indicate memorized samples (where $d_{1\text{NN}} < d_{2\text{NN}}/3$). Swapping in the medium region (steps 6–13) completely flips memorization behavior, while swapping only in the danger zone (steps 7–10) shows partial effects.

## F.4. Continuous Trajectory Ratio Heatmap

We consider $\sigma_0 > \cdots > \sigma_{17}$ from Table 2. For each CIFAR-10 test image $x_{\text{test}}$ and each $i = 0, \ldots, 17$, initialize $x_i^i = x_{\text{test}} + \sigma_i \mathbf{Z}$. We then follow the Euler discretization of (4) starting from $\sigma = \sigma_i$, obtaining successive iterates $x_{k+1}^i = \text{ODE}_k^\theta(x_k^i)$ for $k = i, \ldots, 16$.

For each intermediate step $k$, we record the nearest-neighbor ratio $r_{i:k} = d_{1\text{NN}}(m^\theta_{\sigma_k}(x^i_k))/d_{2\text{NN}}(m^\theta_{\sigma_k}(x^i_k))$ and define the memorization rate at that step as the fraction of samples with $r_{i:k} < 1/3$; see Figure 13 for the resulting heatmap.

We find that the accumulation effect is present and most significant in the medium noise regime: even if the one-step memorization rate at the noise level $\sigma_7 = 3.3$ is only 14.1%, after four ODE steps, the memorization rate jumps to 91%.

Because the threshold $1/3$ is somewhat arbitrary, we also report the mean ratio $r_{i:k}$ in Figure 14. The same pattern appears without thresholding: as the trajectory passes through the medium-noise region, denoiser outputs become progressively closer to a single nearest training image relative to the second nearest neighbor.

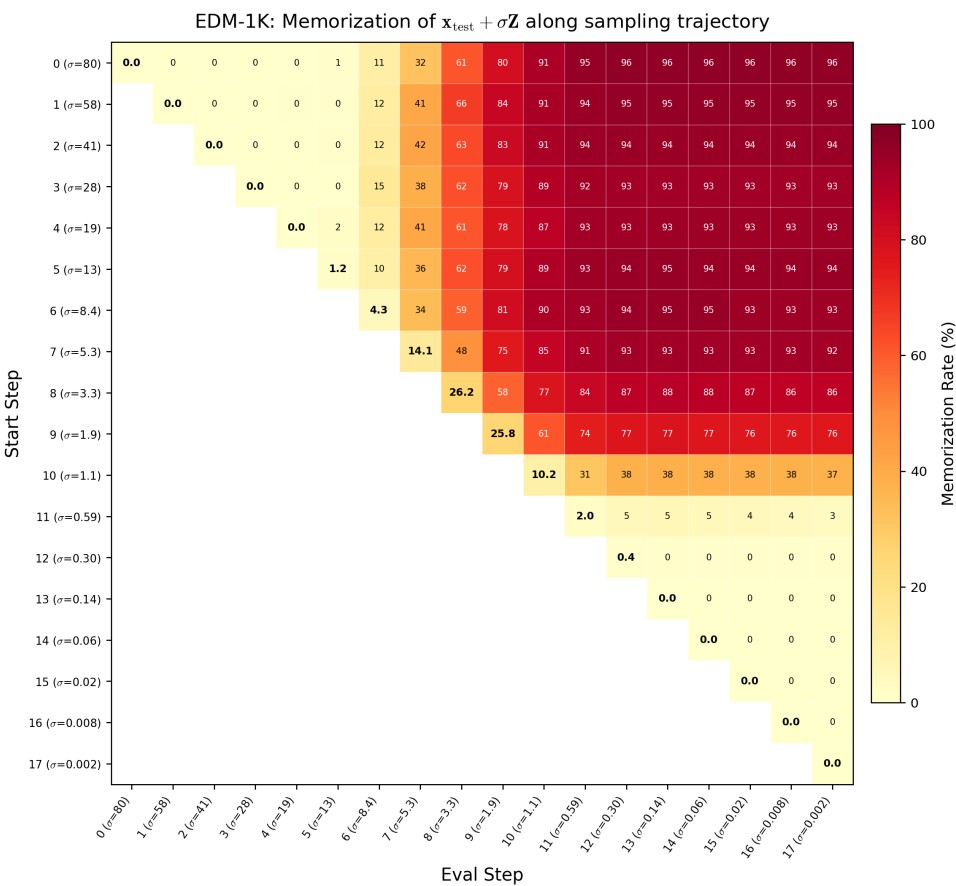

*Figure 13.* **Euler ODE trajectory memorization heatmap (EDM-1K).** Each cell $(i, k)$ reports the fraction of samples with $d_{1\text{NN}}(m^\theta_{\sigma_k}(x^i_k))/d_{2\text{NN}}(m^\theta_{\sigma_k}(x^i_k)) < 1/3$ after applying the Euler discretization of (4) from $\sigma_i$ to $\sigma_k$. Diagonal entries agree with the per-noise-level memorization rates, while off-diagonal entries show accumulation along the denoising trajectory, strongest in the medium-noise regime.

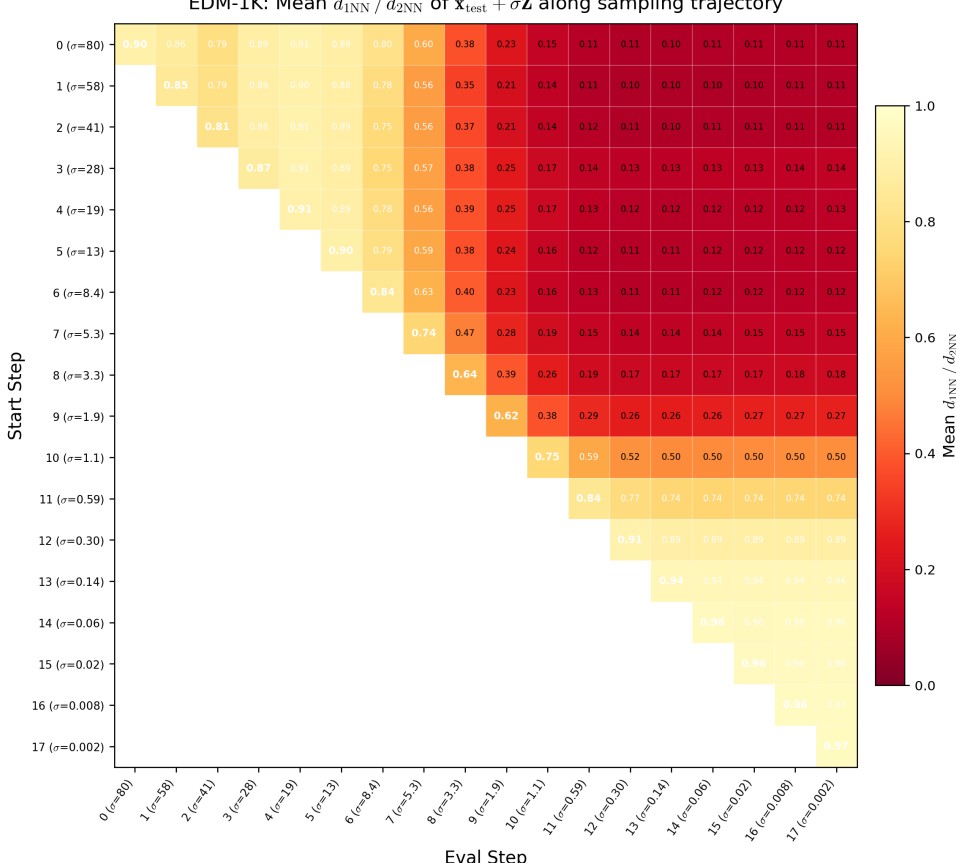

*Figure 14.* **Mean nearest-neighbor ratio heatmap (EDM-1K).** Same layout as Figure 13, but each cell reports the mean $d_{1\text{NN}}/d_{2\text{NN}}$ ratio rather than a binary memorization indicator. Lower ratios indicate that the denoiser output is much closer to one training image than to its next nearest alternative. The dark band through the medium-noise steps mirrors the thresholded heatmap, confirming that trajectory-level memorization reflects a continuous concentration process rather than an artifact of the chosen threshold.

## F.5. Empirical Shell Coverage on CIFAR-10

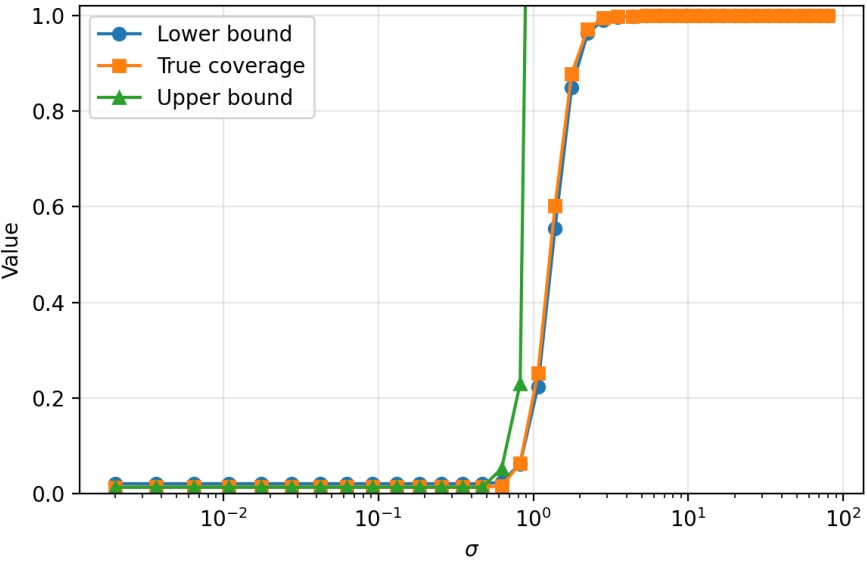

*Figure 15.* **Empirical shell coverage on CIFAR-10.** We plot the lower bound, coverage, and upper bound from Theorem 4.8. The lower bound curve (blue) is nearly indistinguishable from the coverage curve across the entire range of $\sigma$, as it almost perfectly overlaps with it.

We empirically evaluate the lower and upper bounds in Theorem 4.8 and compare them with the Gaussian shell coverage probability on the CIFAR-10 dataset; see Figure 15. All images are flattened to vectors in $\mathbb{R}^d$ with $d = 3072$ and linearly rescaled to lie in $[-1, 1]^d$. For a fixed shell width parameter $c = 5$, we estimate the function

$$\Phi_{d,c}(t) = \mathbb{P}(\|\boldsymbol{Z}\| \in [a_d, b_d], \ \|\boldsymbol{Z} + t e_1\| \in [a_d, b_d]), \qquad \boldsymbol{Z} \sim \mathcal{N}(0, I_d),$$

using Monte Carlo with 5000 i.i.d. Gaussian samples. We reuse the same Gaussian samples to evaluate $\Phi_{d,c}(t)$ for all values of $t$ encountered in the experiment.

**Noise schedule.** We use the EDM polynomial noise schedule (Eqn. (36)) with $\sigma_{\max} = 80$, $\sigma_{\min} = 0.002$, and a 40-step discretization ($N = 40$).

**Upper bound.** To estimate the upper bound term $N \, \mathbb{E}[\Phi_{d,c}(\|\boldsymbol{X} - \boldsymbol{X}'\|/\sigma)]$ with $N = 1000$, we independently sample 12,000 pairs $(\boldsymbol{X}, \boldsymbol{X}')$ from the CIFAR-10 training set and compute the empirical average of $\Phi_{d,c}(\|\boldsymbol{X} - \boldsymbol{X}'\|/\sigma)$ for each noise level $\sigma$.

**Lower bound.** To estimate the lower bound term $\mathbb{E}[\Phi_{d,c}(d_{1\mathrm{NN}}(\boldsymbol{X})/\sigma)]$, we perform 250 independent trials. In each trial, we sample a fresh subset $\{\boldsymbol{X}_1, \ldots, \boldsymbol{X}_N\}$ of $N = 1000$ training images and independently sample 8 test images $\boldsymbol{X}$. For each test image, we compute the nearest-neighbor distance $d_{1\mathrm{NN}}(\boldsymbol{X}) = \min_{1 \leq i \leq N} \|\boldsymbol{X} - \boldsymbol{X}_i\|$ and evaluate $\Phi_{d,c}(d_{1\mathrm{NN}}(\boldsymbol{X})/\sigma)$. The results are averaged over test images and trials.

**Gaussian shell coverage.** We estimate the true coverage probability $\mathbb{P}(\boldsymbol{X}_\sigma \in \mathcal{U}_\sigma)$ using 350 independent trials. In each trial, we sample a fresh subset $\{\boldsymbol{X}_1, \ldots, \boldsymbol{X}_N\}$ of size $N = 1000$ and an independent test image $\boldsymbol{X}$. For each noise level $\sigma$, we draw 10 independent Gaussian noises $\boldsymbol{Z}$ and check whether $\boldsymbol{X} + \sigma \boldsymbol{Z}$ lies in at least one shell centered at the subset points. The coverage probability is estimated by averaging over noises and trials.

## F.6. Dataset-Size and Latent-Space Scaling

To check whether the small-noise sparsity and medium-noise transition are artifacts of using a small CIFAR-10 subset, we repeat the same distance and coverage diagnostics across larger collections. Figures 16 and 17 report the nearest-neighbor separation diagnostics, and Figure 18 reports the corresponding coverage–weight curves across CIFAR-10 dataset sizes.

The nearest-neighbor distance distributions remain sharply concentrated as the CIFAR-10 subset size grows and also in high-dimensional ImageNet SD-VAE latent space. Thus the local separation responsible for low small-noise coverage does not disappear simply by adding more samples. At the same time, the CIFAR-10 coverage–weight curves retain a clear overlap region as $N$ increases, indicating that the danger-zone geometry is not specific to the 1K setting.

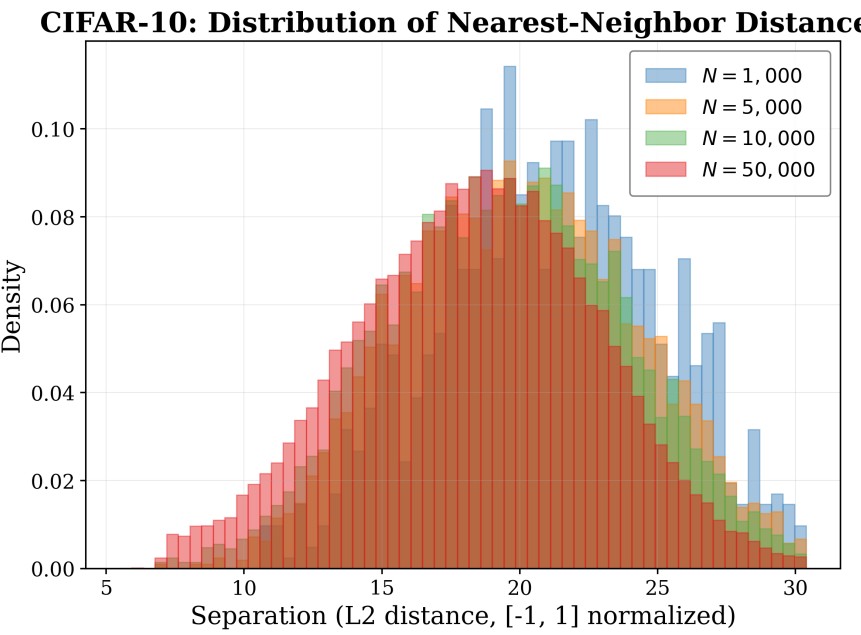

*Figure 16.* **CIFAR-10 nearest-neighbor separation across dataset sizes.** The 1-NN distance distributions largely overlap as the subset size increases, showing that local separation remains stable rather than collapsing in the larger CIFAR-10 samples. This supports the small-noise picture: Gaussian shells around individual training points remain sparse until the noise level is large enough to bridge typical nearest-neighbor gaps.

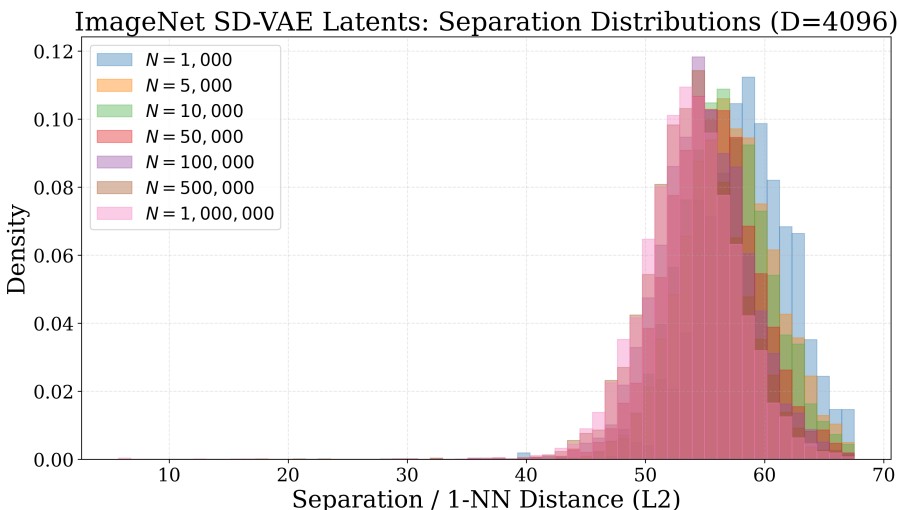

*Figure 17.* **ImageNet SD-VAE latent nearest-neighbor separation.** We repeat the same 1-NN distance diagnostic after encoding ImageNet images into SD-VAE latent space ($D = 4096$). The distributions remain concentrated and stable up to $N = 500K$, suggesting that the local sparsity underlying shell coverage is also visible in a representation used by modern latent diffusion models.

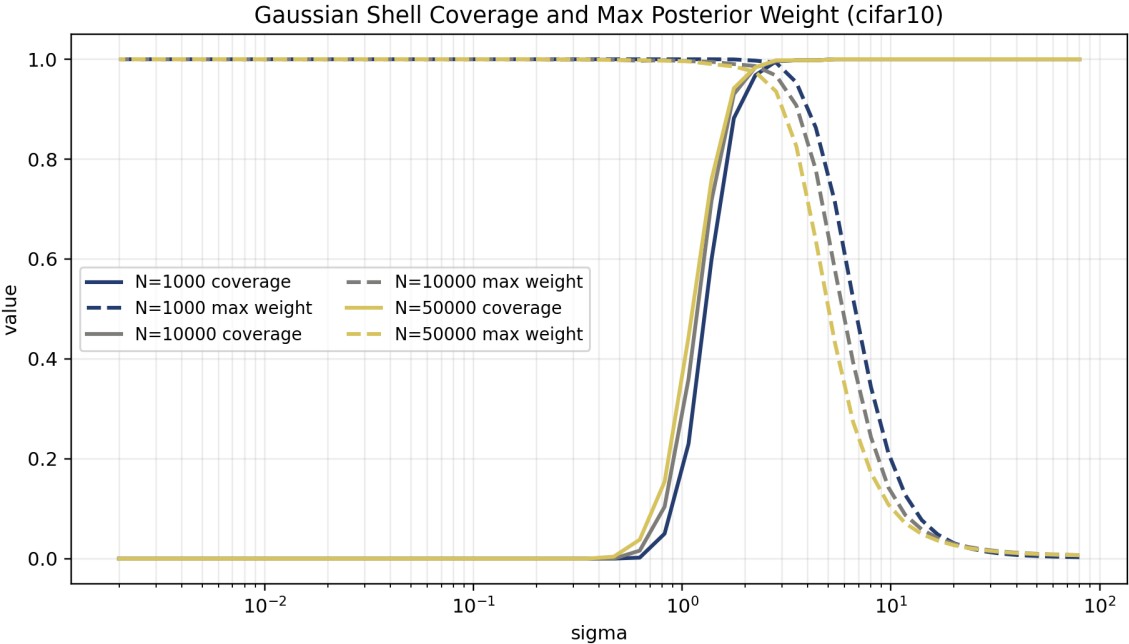

*Figure 18.* **Gaussian shell coverage and posterior weight across dataset sizes.** We plot coverage $C_\sigma(p, \mathcal{D})$ and maximum posterior weight $W_\sigma(\mathcal{D})$ for CIFAR-10 subsets with $N = 1K, 10K, 50K$. As $N$ grows, the curves continue to show a sharp transition and an intermediate overlap region where coverage is already substantial while posterior weight remains concentrated. This overlap is the predicted memorization danger zone.

## F.7. Max vs Self Posterior Weight

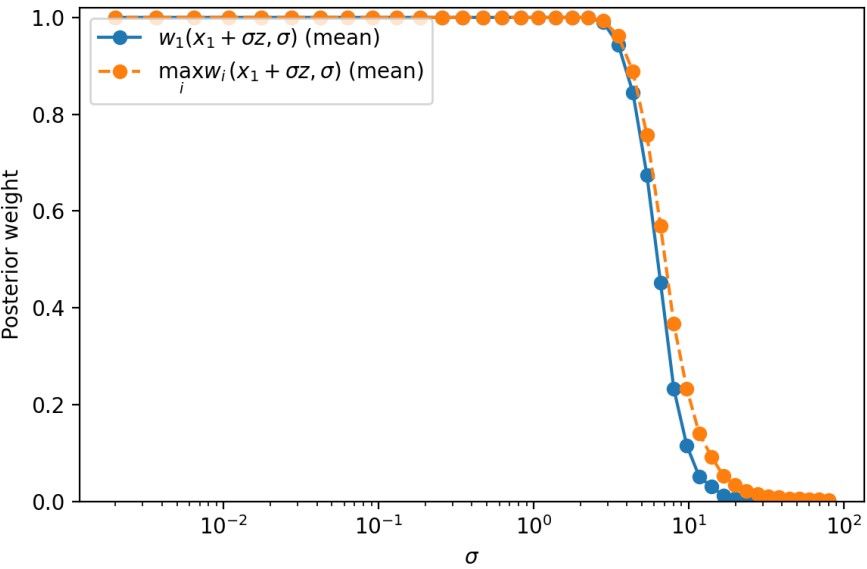

*Figure 19.* **Max vs. Sample Weight on 1K CIFAR-10.** We plot average values of $\max_{1 \le i \le N} w_i(x_1 + \sigma \mathbf{Z}, \sigma)$ and $w_1(x_1 + \sigma \mathbf{Z}, \sigma)$ across a range of $\sigma$. The two curves show that the max posterior weight can usually be well estimated by the sample weight, particularly at smaller noise levels where they coincide.

For each noise level $\sigma$, we fix a subset of $N = 1000$ images sampled uniformly without replacement from the CIFAR-10 dataset. We then draw 100 base points without replacement from the subset and for each base point $x_1$ we compute for

each $i \in \{1, \ldots, N\}$, $w_i(x_1 + \sigma z, \sigma)$ over 400 independent Gaussian noises $z$. We then compute the averages for both $\max_i w_i(x_1 + \sigma z, \sigma)$ and $w_1(x_1 + \sigma z, \sigma)$ over all base points $x_1$ and subsequent samples $z$. The resulting curves show the averages for each value of $\sigma$.

### F.8. Danger Zone Identification: 2k Car Subset

In Figure 20, we overlay the coverage and max posterior weight curves from Figure 3 for the 2k car training set and 1k car test set. The danger zone is identified as the region where both metrics are high, which in the case of $2k$-car is a narrow interval containing the intersection point $\sigma \approx 1.9$.

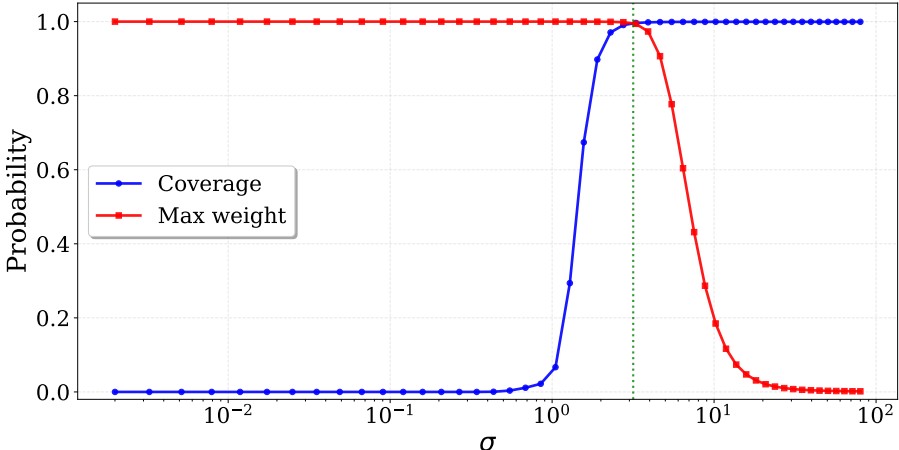

*Figure 20.* Coverage and max posterior weight vs noise level for 2k car training set and 1k car test set. The danger zone is where both metrics are high, spanning $\sigma \in [1, 5]$ with intersection near $\sigma \approx 1.9$.

### F.9. ImageNet Goldfish Gap Training

We repeat the danger-zone identification and gap-training intervention on the 1,300 ImageNet goldfish images resized to $64 \times 64$. The coverage curve rises in the same range where the max posterior weight remains high, identifying an overlap region around $\sigma \in [3, 7]$. We then train EDM models with and without excluding this interval from the training noise distribution.

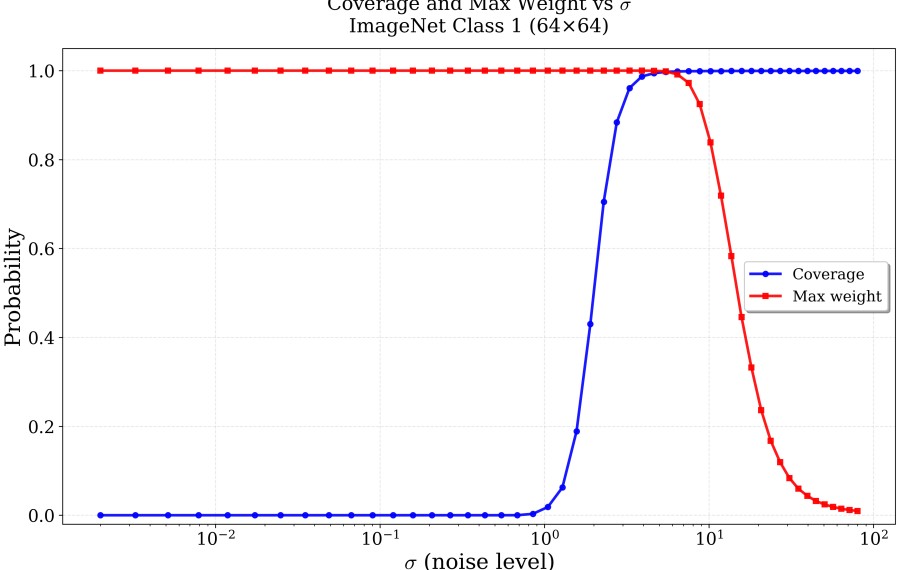

*Figure 21.* **ImageNet goldfish danger-zone identification.** Coverage and max posterior weight as a function of noise level $\sigma$ for ImageNet goldfish (1,300 images, $64 \times 64$). Coverage transitions upward while the max posterior weight remains concentrated over the range $\sigma \approx 3$–7, producing the same kind of simultaneous high-coverage/high-weight region observed for CIFAR-10 cars. We use this interval as the predicted danger zone for the gap-training experiment.

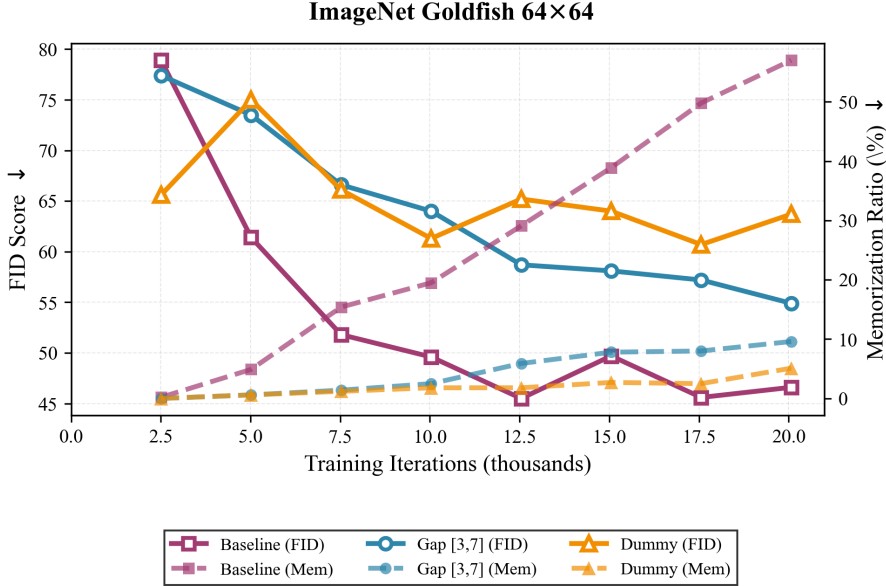

*Figure 22.* **ImageNet goldfish gap training.** FID and memorization ratio over training for ImageNet goldfish ($64 \times 64$, 1,300 images). The baseline increasingly memorizes as training proceeds, reaching $57.0\%$ memorization at the final checkpoint. Excluding the predicted danger zone, Gap $[3, 7]$, keeps memorization to $9.6\%$ at the same checkpoint with a moderate FID increase ($54.9$ vs. $46.6$). The noise-dummy baseline reduces memorization further ($5.1\%$) but incurs a larger quality cost (FID $63.7$).

## F.10. Stable Diffusion v1.4 Finetuning

We also test the coverage–weight picture in a text-to-image finetuning setting, where prompt conditioning can reduce the effective training set to very few images. In our finetuning set, each prompt $y$ is paired with a single training image. Within this empirical conditional distribution, the conditioned posterior over training images is therefore a point mass on the given image, so the max posterior weight is always 1 for every $\sigma$. Thus, unlike in the unconditional multi-image setting, there is no large-$\sigma$ protection from posterior mass spreading across many examples. The relevant transition is instead a localized coverage transition: when the Gaussian shell around the training latent begins to overlap with the population of plausible images compatible with the prompt.

We estimate this transition using a localized coverage curve. For each training latent $x_i$ and noise level $\sigma$, we measure how much the Gaussian shell centered at $x_i$ overlaps a reference population of natural-image latents. Averaging over finetuning examples gives a curve that is near zero when the noisy shell remains isolated around the memorized example and rises when the shell begins to intersect plausible alternative images. This can be viewed as a localized version of the coverage studied in the main text: because it studies when the single-example noisy neighborhood gains population-level coverage. Since the true prompt-conditioned population is unobserved, we use a broad LAION data reference proxy: we encode 50K LAION-Aesthetic images using the SD VAE and compute the shell-overlap curve around each memorized training latent. This conservative proxy places the onset of nontrivial coverage near $\sigma \approx 3$. We then finetune SD v1.4 on 200 memorized image–prompt pairs, comparing the unmodified schedule with progressively stricter caps on the sampled training noise.

Figures 23–27 summarize the finetuning experiment. Figure 23 first shows that, for a single prompt–image pair, the posterior weight remains fixed at one while the localized LAION-reference coverage begins to rise around $\sigma \approx 3$. Figure 24 then tests the resulting prediction by tracking memorization during SD v1.4 finetuning: the uncapped schedule produces the strongest increase in SSCD similarity, while progressively lower noise caps reduce memorization. Figures 25–27 provide the corresponding qualitative comparison, showing that the baseline generations increasingly collapse toward the memorized training images, whereas stricter caps preserve more diversity across epochs and seeds.

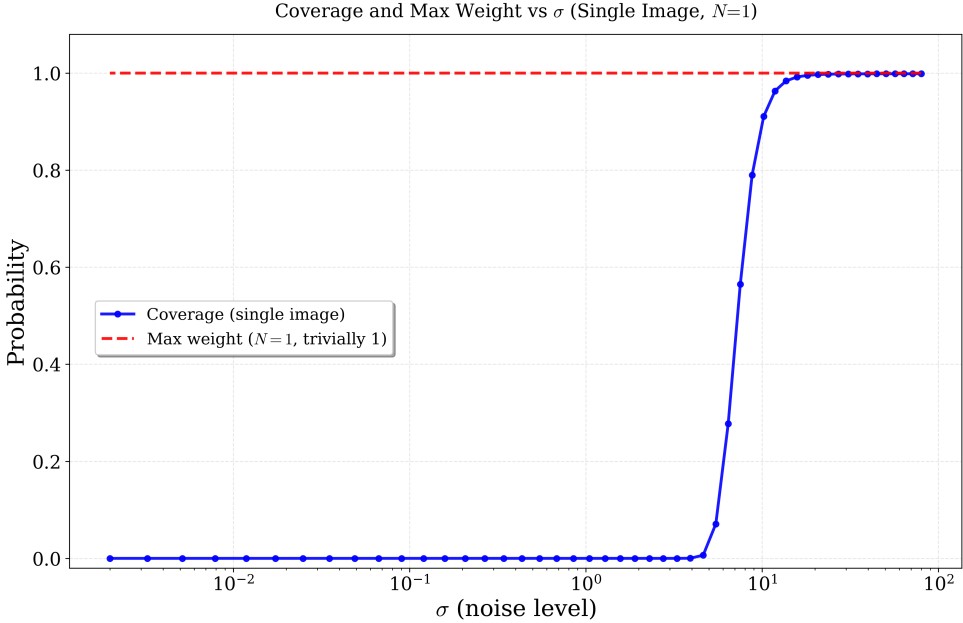

*Figure 23.* **Single-image coverage in SD v1.4 latent space.** Coverage and max posterior weight as a function of noise level $\sigma$ for a single image-prompt pair. The max posterior weight is one because the prompt identifies a single training image. The (localized) coverage curve uses a LAION data reference estimate: we encode 50K LAION-Aesthetic images with the SD VAE and measure when their noisy latents fall in the Gaussian shell around the memorized training latent. This gives nontrivial coverage around $\sigma \approx 3$. Above this point, noisy latents from the training image overlap the broader latent population while the prompt-conditioned posterior remains concentrated, giving the predicted danger zone $[\sigma_{\mathrm{cov}}, \infty)$.

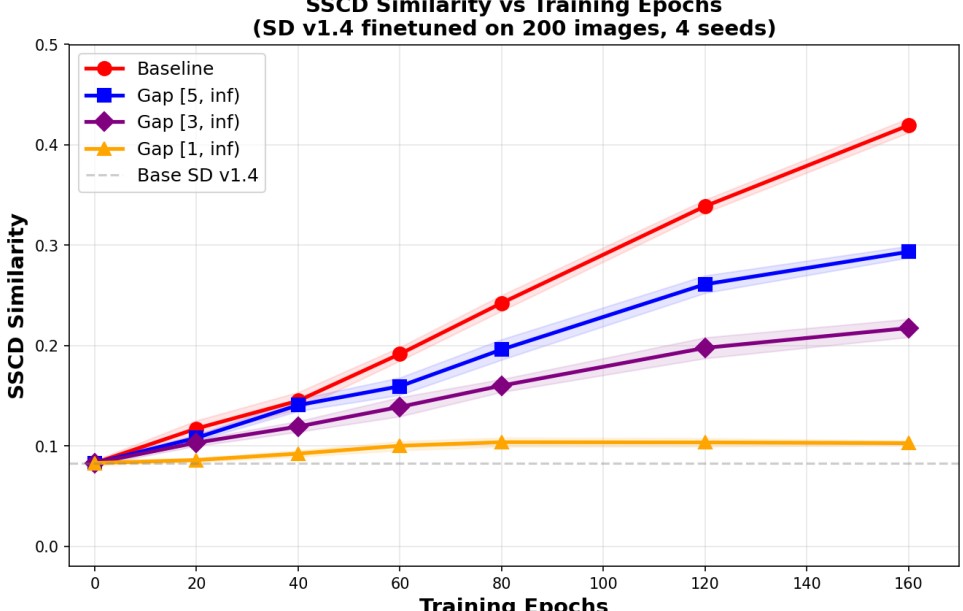

*Figure 24.* **Stable Diffusion finetuning SSCD over epochs.** Mean SSCD similarity between generated and training images during SD v1.4 finetuning on 200 memorized image-prompt pairs, averaged over four random seeds. The unmodified schedule produces a steady rise in copying, reaching mean SSCD $0.42$ at 160 epochs. Removing high-coverage noise levels lowers the curve in a dose-dependent way: caps at $\sigma < 5$, $\sigma < 3$, and $\sigma < 1$ reduce the final mean SSCD to $0.29$, $0.22$, and $0.10$, respectively. The pretrained SD v1.4 model before finetuning is shown as a reference.

*Figure 25.* **Stable Diffusion qualitative finetuning samples: ballet prompt.** Qualitative samples from the SD v1.4 finetuning experiment for the NYC Ballet prompt. Rows show the baseline and caps $\sigma < 5$, $\sigma < 3$, and $\sigma < 1$, while columns show the pretrained model and later finetuning epochs, with four random seeds per cell.

**Prompt: 3D Metal Cornish Harbour Painting**

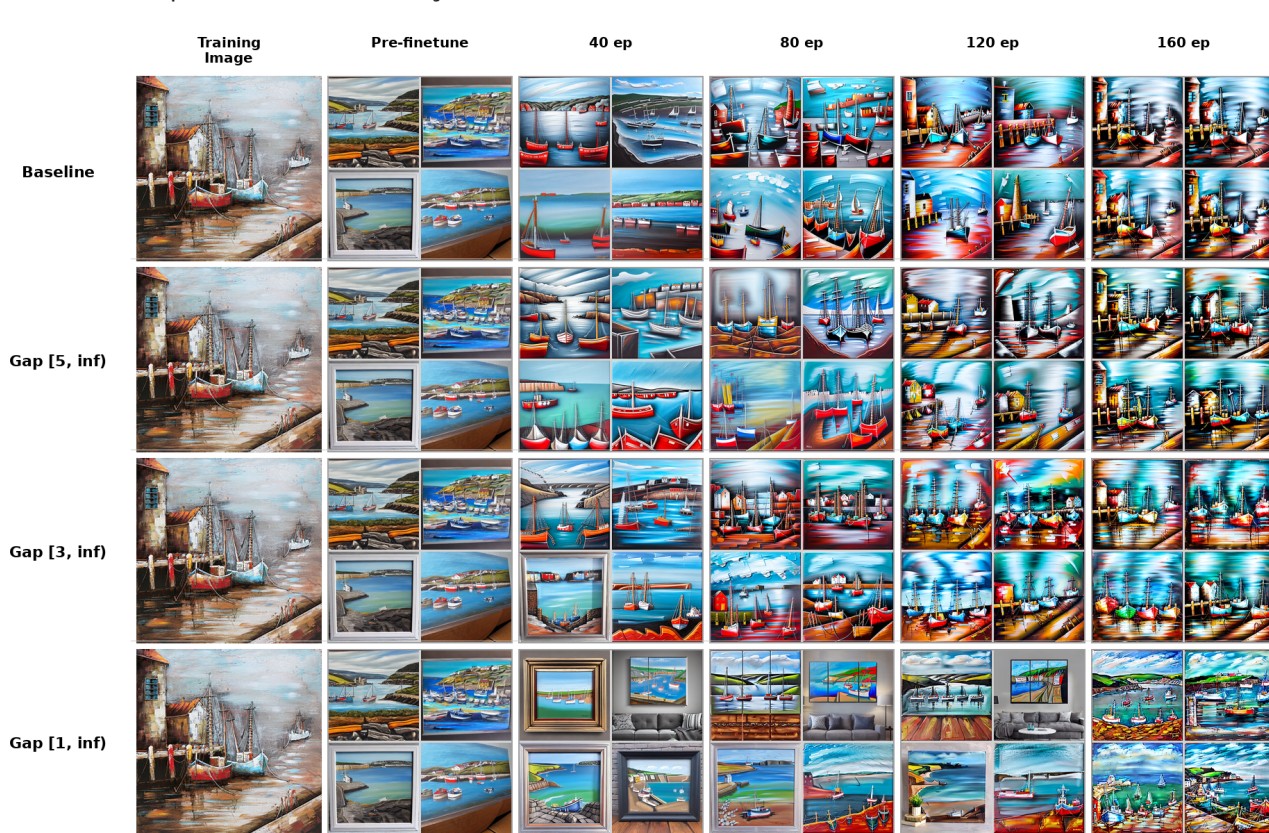

*Figure 26.* **Stable Diffusion qualitative finetuning samples: harbor-painting prompt.** Same comparison for the 3D Metal Cornish Harbour Painting prompt. Baseline generations become increasingly similar to the associated training image, whereas stricter caps slow or prevent this collapse and preserve more variation across seeds.

Prompt: THE STANDARD, LONDON - INTERIOR: The central London hotel, opposite St Pancras Station, features colourful interiors

*Figure 27.* **Stable Diffusion qualitative finetuning samples: hotel-interior prompt.** Same comparison for the Standard London hotel-interior prompt. Across prompts, progressively stricter noise caps reduce the visible collapse toward the memorized training image.

## F.11. Pareto Frontier of Training Configurations on 1024 Grayscale CelebA Images

We compute the FID score of 10k samples against 10k held out test images. We visualize the Pareto frontier of different timestep gap configurations on the 1024 grayscale CelebA training set in Figure 28. Each point represents a different gap configuration, plotting FID score against memorization rate (FMEM%). The dashed line connects non-dominated solutions forming the Pareto frontier.

*Figure 28.* Pareto frontier showing the trade-off between image quality (FID) and memorization (FMEM%). Each point represents a different timestep gap configuration. The dashed line connects non-dominated solutions forming the Pareto frontier.

## F.12. Detailed Memorization Visualizations for Gap Training

We provide detailed visualizations comparing the memorization behavior of the three anti-memorization methods evaluated in Section 5.

Each image pair is arranged as [Generated sample — 1-NN from training set], where the left image shows a generated sample and the right image shows its nearest neighbor from the 2,000-image training set. A yellow border around both images in a pair indicates that the sample is classified as memorized according to the criterion: $d_{1NN} < d_{2NN}/3$. Figures 29, 30, and 31 show complete 8×8 grids with 64 generated samples for each method, enabling detailed assessment of memorization patterns.

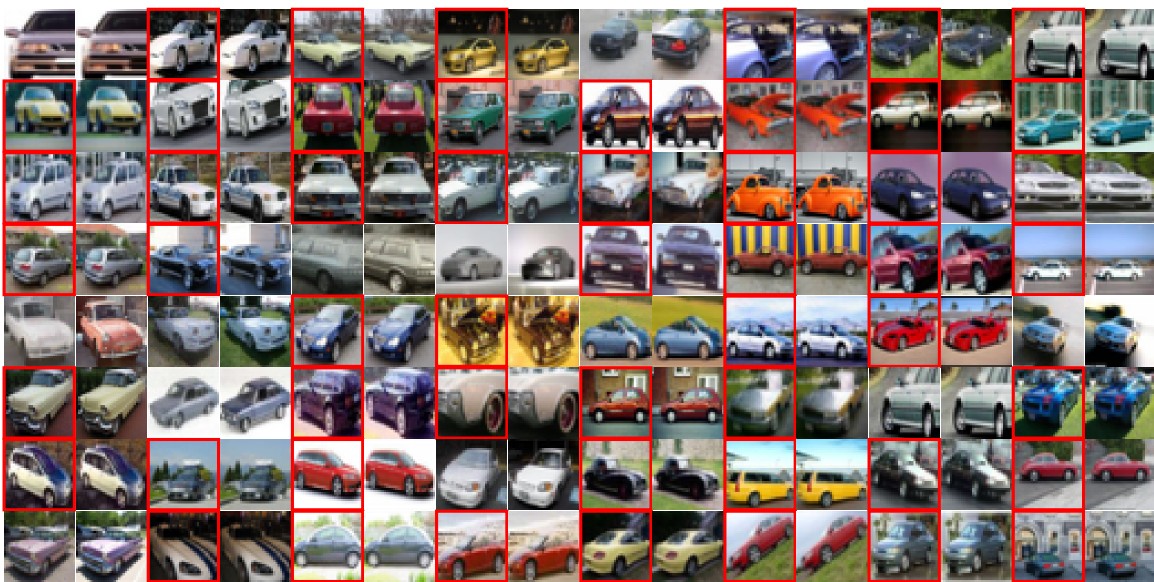

*Figure 29.* **2k-Baseline detailed visualization.** This 8×8 grid shows 64 image pairs, each arranged as [Generated — 1-NN from training]. In each pair, the generated sample appears on the *left* and its nearest neighbor from the 2,000-image training set appears on the *right*. Red borders indicate memorized pairs. The high prevalence of red borders confirms severe memorization in the baseline case.

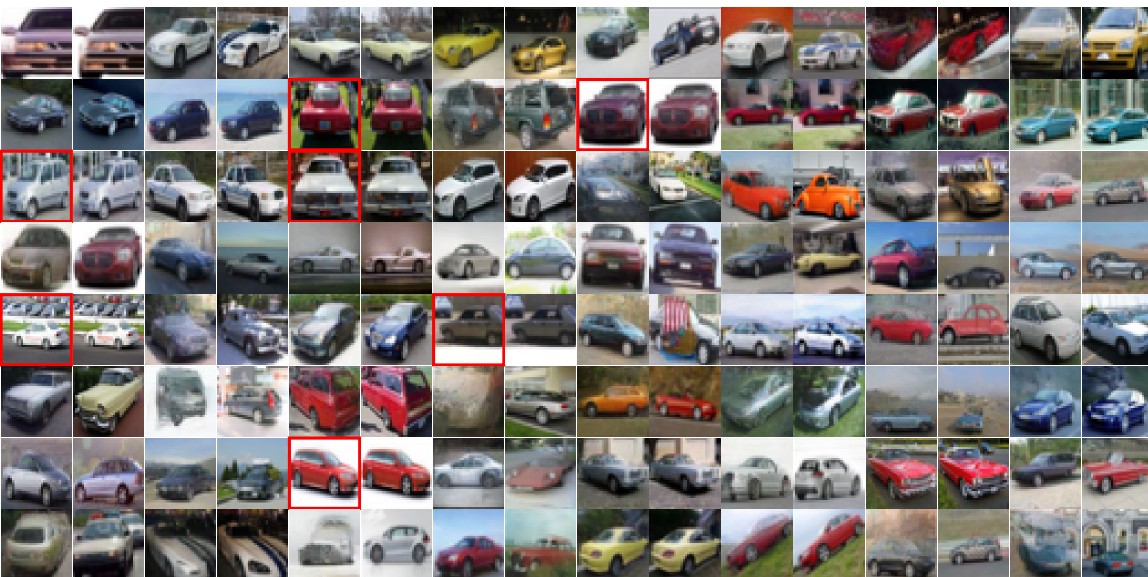

*Figure 30.* **2k-Dummy detailed visualization.** Same layout as Figure 29: each of the 64 pairs shows [Generated — 1-NN]. Red borders mark memorized pairs. Conditional training with an auxiliary noise class substantially reduces memorization. Most generated samples now deviate meaningfully from their nearest neighbors, though several memorized samples remain (red borders).

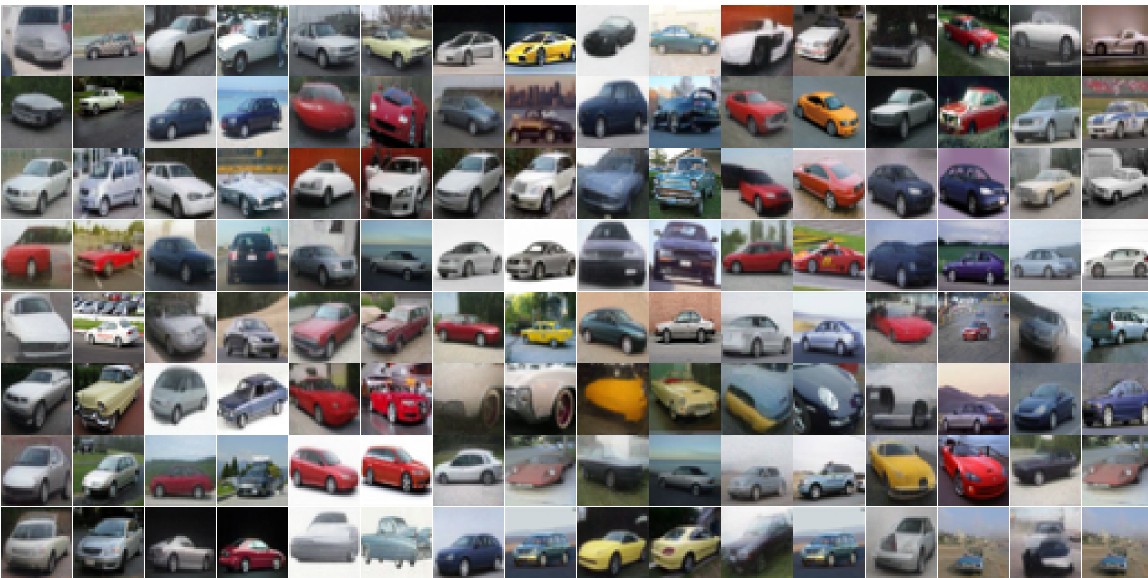

*Figure 31.* **2k-Gap detailed visualization.** Same layout: 64 pairs of [Generated — 1-NN]. Gap training (excluding $\sigma \in [1.0, 5.0]$ from training) achieves minimal memorization. Generated samples maintain semantic similarity to cars while introducing novel variations in color, viewpoint, and detail. This demonstrates that targeted undertraining in the medium-$\sigma$ danger zone enables robust generalization.

### F.13. Timestep Gap Configurations: Visual Comparisons

We provide visual comparison of memorization behavior for different timestep gap configurations on 1024 grayscale CelebA samples in Figure 32, Figure 33, Figure 34 and Figure 35. Each 4×4 grid shows 16 image pairs arranged as [Generated — 1-NN from training]. Red borders indicate memorized samples according to the criterion: $d_{1NN} < d_{2-NN}/3$.

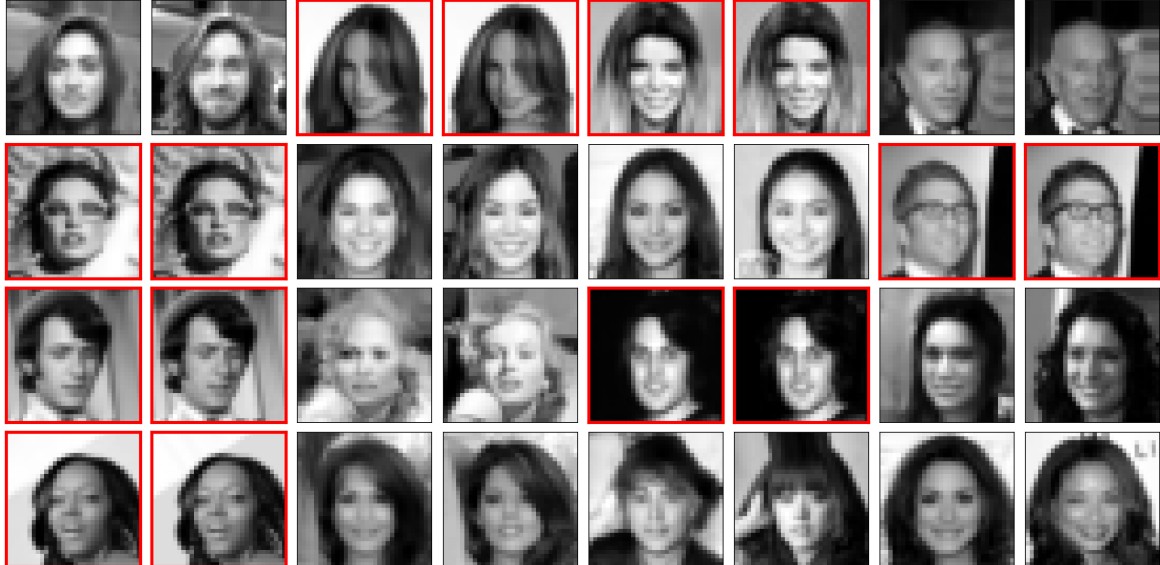

*Figure 32.* **Baseline (no gap).** 74.97% memorization. The high prevalence of red borders confirms severe memorization in this low-data regime.

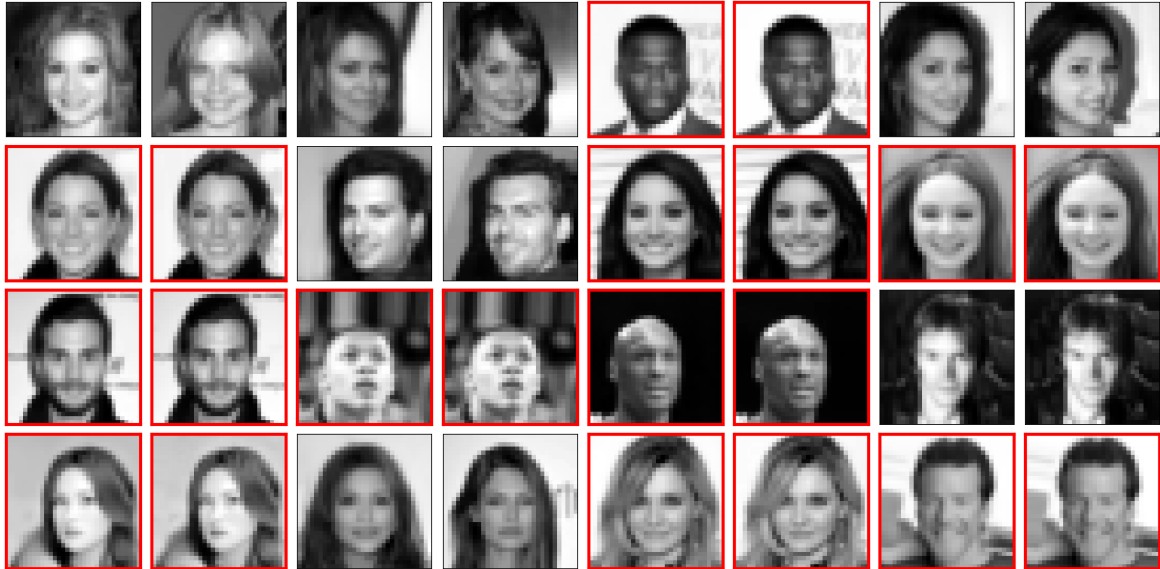

*Figure 33.* **Gap [100–200].** 42.93% memorization. Skipping 100 timesteps from 100 to 200 in the early danger zone reduces baseline's 74.97% memorization effectively.

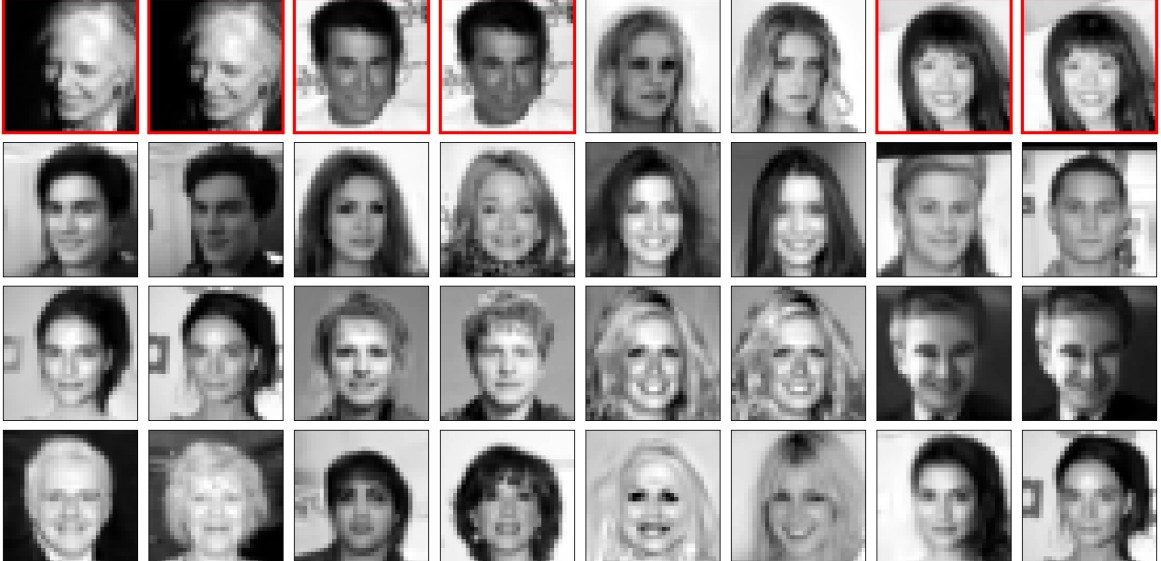

*Figure 34.* **Gap [100–275].** 13.45% memorization. Extending the gap further into the danger zone dramatically reduces memorization.

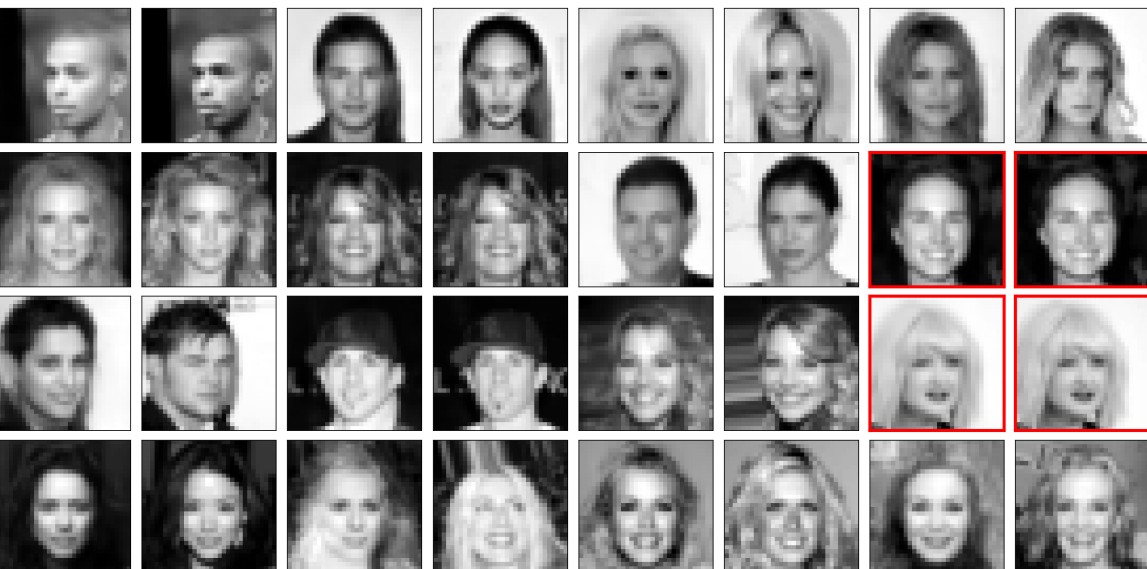

*Figure 35.* **Gap [100–350].** 9.24% memorization. The widest gap achieves the lowest memorization rate, though at the cost of increased FID (26.74 vs 13.09 baseline).

