# OpenReview forum: "Two Calm Ends and the Wild Middle: A Geometric Picture of Memorization in Diffusion Models"
_ICML.cc/2026/Conference — ICML 2026 regular_

### Official Review · Reviewer_ohzn · 2026-03-13

**Soundness:** 3
**Presentation:** 3
**Significance:** 2
**Originality:** 3
**Overall Recommendation:** 4
**Confidence:** 3

**Summary:**

This work provides a deep investigation of the memorization mechanism of diffusion models. They identity the medium noise levels as the danger zone that contributes the most to memorization, and explain this observation with theorem. To prevent memorization, they hence apply dummy-data training in this danger zone or exclude this region during sampling.

**Compliance With Llm Reviewing Policy:**

Affirmed.

**Final Justification:**

I think this work has merits as it provides a detailed study on the existence of danger zone or the critical window that is most important to generalization. However, it is not clear how to utilize these findings to mitigate memorization. Hence I keep my current score.

**Key Questions For Authors:**

1. Will 2-K gap method causes harmful effect to the generation quality? Can 2-K gap or 2-K dummy work on larger and more complex dataset as ImageNet?

2. Though this work specifically focus on memorization, can the authors provide some insights on the mechanism of generalization?

**Limitations:**

The main limitation is that this work is the memorization preventing section is not good enough. It is not clear whether the two proposed methods can work on more complex dataset such as ImageNet.

**Strengths And Weaknesses:**

Strengths:

The paper provides a more in depth investigation of memorization in diffusion models compared to existing works. The existence of danger zone is novel and interesting, and potentially can guide the design of memorization preventing algorithms.

Weakness:

The main weakness is section 5, where the authors propose two algorithms for preventing memorization in the danger zone. The 2k-dummy method is proposed by previous work, hence the novelty is limited. The 2k-gap method directly excludes the danger zone, which is counter intuitively as I expect excluding a specific noise region will be harmful for the generation quality.

---

> ### Author Rebuttal · Authors · 2026-03-31
>
> We thank the reviewer for the constructive feedback and for recognizing the novelty and importance of the identified "danger zone." Below, we focus on the major points raised by the reviewer.
>
> Additional figures (figN.md) referenced below can be found in our anonymous repository: https://anonymous.4open.science/r/27904/.
>
> **Clarification on novelty in Section 5.**
>
> We would like to clarify that the 2K-dummy is not our new work (nor did we claim this is our contribution), but a baseline anti-memorization method from prior work [1], which also introduced the 2K-car evaluation setting. It is therefore natural for us to start from their setting and compare against their method. Our contribution is 2K-gap, which is motivated by our geometric analysis of the danger zone.
> The dummy method adds dummy samples to limit the model's capacity to memorize real data. By contrast, our gap training skips the specific noise regime that our coverage--weight analysis identifies as responsible for memorization. It also reduces training cost by omitting the danger zone, whereas the dummy method spends computation on learning samples that do not improve generation. We will clarify this distinction in the revised Section 5 to avoid future confusion.
>
> **On generation quality and applicability of our 2K-gap method to more complex datasets.**
>
> We first emphasize that the full removal of the danger region is a targeted, mechanism-validating intervention designed to isolate the causal role of this regime, not a claim that one should always delete it in practice. Interestingly, in the limited-data regime where memorization directly competes with generation quality, 2K-gap sometimes may not hurt FID; in the 2K-car setting, it in fact improves FID from 3.8 to 2.35 (see Figure 4 in our manuscript), and our qualitative comparisons show that sample quality is preserved.
> At the same time, we acknowledge that full removal is not ideal in general, because the same medium-noise regime also carries important generative structure. This is exactly why we view 2K-gap as a controlled validation of the mechanism rather than a universally optimal algorithm. The existing multi-gap CelebA experiments already show a clearer quality--memorization trade-off across partial gaps, and we surface those results more prominently to make this nuance explicit. We also validate gap training on ImageNet goldfish, supporting the view that the mechanism extends beyond CIFAR-10. Concretely, we train an EDM model on the full ImageNet class #1 (goldfish) with 1300 images at resolution 64x64. The coverage vs. max weight plot (see Figure fig7.md) shows a transition region near $\sigma \approx 5$ and suggests a danger zone around $[3, 7]$. We compare baseline training, the gap method with noise levels in $[3, 7]$ removed, and the dummy-data method. The result is consistent with our 2K CIFAR-10 case: both gap and dummy methods are effective at reducing memorization. The baseline memorization ratio increases monotonically to 57% at 20K training steps, while gap and dummy reduce memorization to 9.5% and 7.3%, respectively (see Figure fig8.md).
>
> **Regarding the question on generalization.**
>
> One of our main takeaways regarding generalization is that it appears to be driven primarily by the intermediate-noise regime. We included the swapping experiment (see Figure 10 in the appendix of our manuscript) precisely to probe this point. This makes the intermediate regime especially important: it is the regime most responsible for both memorization and generalization.
> We suspect that the exact learning outcome in this regime depends strongly on the interaction between data size and model capacity. More broadly, our results also suggest that one should pay closer attention to the non-uniformity of the training data: a model may generalize well in some regions of the data distribution while still memorizing sparse regions at the same time. Altogether, this points to the need for more data-geometry-dependent training designs to suppress memorization while promoting generalization.
>
> [1] Gu et al. On Memorization in Diffusion Models. TMLR, 2025.

---

> > ### Author Rebuttal · Reviewer_ohzn · 2026-04-02
> >
> > I think this work has merits as it provides a detailed study on the existence of danger zone or the critical window that is most important to generalization. However, it is not clear how to utilize these findings to mitigate memorization. Hence I keep my current score.

---

> > > ### Author Response · Authors · 2026-04-06
> > >
> > > We thank the reviewer for engaging in the discussion. To follow up on the point of utilization, we now provide preliminary evidence that this framework extends to large-scale text-to-image model fine-tuning. This preliminary result demonstrates one concrete usage of the danger-zone framework: identifying the memorization-prone noise region and removing it to mitigate memorization in a real text-to-image diffusion model.
> > > Additional figures (figN.md) referenced below can be found in our anonymous repository: https://anonymous.4open.science/r/27904/.
> > >
> > > Finetuning Stable Diffusion on a small number of image-prompt pairs can induce memorization (see e.g. [1]). We finetune SD v1.4 on 200 memorized image-prompt pairs for 160 epochs; the baseline achieves a mean SSCD similarity of 0.42, indicating memorization [2]. We analyze this setting through the lens of our coverage-weight framework. It turned out that in this dataset, each prompt maps to exactly one training image. In such a case, the conditional posterior places all its mass on that image: $W_\sigma = 1$ for every noise level $\sigma$. This eliminates the large-$\sigma$ protection that exists in the multi-image unconditional case, where the posterior becomes diffuse at high noise. Coverage, on the other hand, still depends on the geometry of the data distribution. To approximate the coverage transition, we encode 50K LAION-Aesthetic images through the SD VAE and compute when the Gaussian shell around a single training point begins to overlap with the broader population. This unconditional estimate places the onset of non-trivial coverage around $\sigma \approx 3$, see `Fig9.md`. (A tighter analysis would condition on each prompt's text embedding, which we leave to future work.)
> > >
> > > In our framework, the danger zone is the noise region where both $W_\sigma$ and coverage are simultaneously high. Here $W_\sigma = 1$ everywhere, so the danger zone is determined entirely by where coverage becomes non-trivial — yielding a danger zone of the form $[\sigma_{\mathrm{gap}}, \infty)$ with $\sigma_{\mathrm{gap}} \approx 3$. The natural intervention is therefore to cap the training noise schedule at $\sigma_{\max} \leq \sigma_{\mathrm{gap}}$, preventing the model from training on the memorization-prone high-coverage region. The $\sigma$-capping experiments provide strong validation. The baseline finetuning (no capping) yields mean SSCD $= 0.42$ at 160 epochs. Capping at $\sigma < 5$ reduces SSCD to $0.29$; capping at $\sigma < 3$ brings SSCD down to $0.22$; and capping at $\sigma < 1$ keeps SSCD near $0.10$ throughout training, see `Fig10.md` for the full curves and `Fig11.md` for qualitative samples. The fact of low SSCD score when the danger zone is eliminated aligns with our framework. We also note that Jain et al. [3] arrive at a complementary inference-time memorization mitigation strategy: withholding text-conditioned guidance at high noise levels, which in our language corresponds to gapping the danger-zone $\sigma$ levels during sampling.
> > >
> > > [1] Wen et al. Detecting, Explaining, and Mitigating Memorization in Diffusion Models. ICLR, 2024.
> > >
> > > [2] Somepalli, et al. "Understanding and mitigating copying in diffusion models." NeurIPS 2023
> > >
> > > [3] Jain, et al. "Classifier-free guidance inside the attraction basin may cause memorization." CVPR, 2025.

---

### Official Review · Reviewer_vxF3 · 2026-03-14

**Soundness:** 2
**Presentation:** 3
**Significance:** 2
**Originality:** 3
**Overall Recommendation:** 3
**Confidence:** 3

**Summary:**

This paper offers an interesting and fairly original perspective on memorization in diffusion models. Its main contribution is the claim that memorization is primarily driven by an intermediate-noise ``danger zone,'' rather than being uniform across the noise schedule. They also show that by removing the training on the ``danger zone'' could mitigate memorization.

**Compliance With Llm Reviewing Policy:**

Affirmed.

**Final Justification:**

The response and additional experiments still do not convince me of the method’s practical usability. Therefore, I am inclined to keep my score unchanged.

**Key Questions For Authors:**

Please refer to Weaknesses part.

**Limitations:**

Additional data is required for the estimation of danger region.

**Strengths And Weaknesses:**

## Strengths
* The paper introduces a useful per-noise-level view of memorization, which is a novel perspective compared with trajectory-level memorization. This decomposition is conceptually valuable and helps explain why memorization behavior is not uniform across timesteps.
* The proposed geometric interpretation based on posterior weight concentration and Gaussian shell coverage is intuitive and provides a unified explanation for the empirical phenomenon.
* The denoiser swapping experiments is interesting and support the claim that the medium-noise region is the most influential part leading to memorization.
* Based on the theoretical and empirical findings, the paper diagnosis and proposes a mitigation strategy, targeted noise-region undertraining (gap training), which appears to reduce memorization without hurting, and sometimes even improving FID.

## Weaknesses
* The paper’s strongest intuition about memorization focuses on the intermediate-noise regime, but this is also the part that remains least rigorously characterized. The results about this region is primarily supported by only geometric insight and emprical ablation.
* The experiments, while well designed, are still limited to relatively small-scale and memorization-prone settings, such as CIFAR-10 and CelebA. It is not yet clear how broadly the conclusions transfer to larger-scale modern diffusion pipelines.
* The undertraining method in the experiment seems totally remove the training on the ``danger region''. Although the paper shows that overfitting on this region leads to memorization, I still have concerns on this fully removing strategy. As the inference process would go through this untrained region, wouldn't it hurt the generation performance?
* The authors use a test sub-dataset to estimate the danger region. This is unfair to use test data for any purpose during training time, since the baseline methods do not have access to that.

---

> ### Author Rebuttal · Authors · 2026-03-31
>
> We thank the reviewer for the thoughtful feedback, as well as for recognizing the novelty of our per-noise-level perspective and geometric interpretation of memorization. Below, we address the main concerns.
>
> Additional figures (figN.md) referenced below can be found in our anonymous repository: https://anonymous.4open.science/r/27904/.
>
> **Regarding the concern on full removal of the danger region during training**
>
> We first clarify that full gap removal is a deliberate experimental design to isolate the causal effect of this regime, rather than a recommended training strategy in its vanilla form. The very same medium-noise region, which we identify as the danger zone, is crucial for learning useful structure (as EDM demonstrates for FID), so complete removal is not generally advisable.
> Despite skipping training on the danger zone, sampling remains stable because the learned denoiser interpolates smoothly between the adjacent trained regimes; empirically, in the limited-data regime where memorization directly competes with generation quality, gap training actually improves FID from 3.8 to 2.35 on 2K-car (see Figure 4 in our manuscript), with quality preserved qualitatively (Appendix E.5).
>
> For data-rich scenarios and large-scale models, this direct intervention may not be ideal. A more principled approach would involve data-point-specific noise schedules that selectively reduce over-reinforcement in the danger zone. We consider this an important future direction and will clarify the distinction in the revision.
>
> **On test-data use**
>
> Regarding the reviewer’s concern on test-data use, we should have made it clear that our framework does not require test data. We repeated the analysis using a 10% holdout from the training set and obtained nearly identical coverage and weight curves (max absolute difference 0.0122 for the relevant percentage values); crucially, the estimated danger zone is unchanged. In practice, a validation split carved from the training data suffices. We will clarify this in the revision.
>
> **Regarding the reviewer's concern on lack of characterization of intermediate region**
>
> We agree that the intermediate-noise regime is currently characterized more through geometric insight and empirical evidence than through a comparably sharp theorem. At the same time, identifying this regime as the key danger zone is itself one of the main contributions of the paper. Our framework shows that memorization is governed by the interaction between posterior concentration and Gaussian shell coverage, and that this interaction is most pronounced in the intermediate regime. Prior work often focus on small noise regime to study or mitigate effect, while our work suggests that mitigation should primarily focus in the intermediate regime. We support this picture through multiple empirical validations, including denoiser swapping, gap training, and the new trajectory heatmaps (Figs. fig1.md, fig2.md). We agree that a sharper theoretical treatment of this joint interaction would strengthen the paper further, and we view it as an important direction for future work.
>
> **Regarding the question of transfer beyond small-scale memorization-prone settings**
>
> Indeed, our experiments are intentionally conducted in a controlled, memorization-prone setting to validate the main methodological contribution of the paper: the **per-noise-level analysis** and the identification of the **danger zone**. We agree that large-scale modern pipelines are more complex, and that memorization may be harder to detect cleanly when the dataset is very large.
> However, even very large datasets contain **locally sparse regions**—such as rare concepts or narrow conditioning subsets in the conditioned diffusion model. This is especially relevant in text-to-image models, where memorization is often associated with prompts or concepts supported by very few training images, sometimes even a single one [1]. Notably, Figure 1 of MemBench ([2]) shows that many of the known memorized prompts in Stable Diffusion v1.4 are about commercial product listings. We hypothesize that overly detailed listing information might create local sparse regions and contribute to memorization. We view extending our theory to such locally sparse regimes, and testing it in settings such as text-to-image models, as an important direction for future work.
>
> [1] Wen et al. Detecting, Explaining, and Mitigating Memorization in Diffusion Models. ICLR, 2024.
>
> [2] Hong, Chunsan, Tae-Hyun Oh, and Minhyuk Sung. "Membench: Memorized image trigger prompt dataset for diffusion models." arXiv preprint arXiv:2407.17095 (2024).

---

> > ### Author Rebuttal · Reviewer_vxF3 · 2026-04-02
> >
> > Thank the authors for the detailed response and explanation. My concerns on the test dataset is addressed. But I am still uncertain whether the danger zone removal could be effect on real-world large dataset situation. Specifically, is the indentification of the danger zone still accurate on large dataset? Moreover, the strategy of fully removing such regions may not be robust or effective in real-world scenarios.

---

> > > ### Author Response · Authors · 2026-04-06
> > >
> > > We thank the reviewer for the engaging discussion. We address the points raised below with a preliminary experiment on Stable Diffusion v1.4, which provides proof-of-concept evidence of effectiveness of noise gapping in large scale text-to-image diffusion model setting. Additional figures referenced below can be found in our anonymous repository: https://anonymous.4open.science/r/27904/.
> > >
> > > **Setup.** Finetuning Stable Diffusion on a small number of image-prompt pairs can induce memorization (see e.g. [1]). We finetune SD v1.4 on 200 memorized image-prompt pairs for 160 epochs; the baseline achieves a mean SSCD similarity score of 0.42, indicating memorization [2]. We analyze this setting through the lens of our coverage-weight framework.
> > >
> > > In this dataset, each prompt maps to exactly one training image. In such a case, the conditional posterior places all its mass on that image: $W_\sigma = 1$ for every noise level $\sigma$. This eliminates the large-$\sigma$ protection that exists in the multi-image unconditional case, where the posterior becomes diffuse at high noise. The danger zone is therefore determined entirely by coverage, yielding a danger zone of the form $[\sigma_{\mathrm{gap}}, \infty)$. To approximate the coverage transition, we encode 50K LAION-Aesthetic images through the SD VAE and compute when the Gaussian shell around a single training point begins to overlap with the broader population. This estimate places the onset around $\sigma_{\mathrm{gap}} \approx 3$, see `fig9.md`. We note that this is likely conservative, as conditioning on the prompt would reduce the relevant population and push the onset to a lower $\sigma$.
> > >
> > > To validate the memorization effect of the danger zone, we cap the training noise schedule at $\sigma_{\max} \leq \sigma_{\mathrm{gap}}$, preventing the model from training on the memorization-prone high-coverage region. The baseline finetuning (no capping) yields mean SSCD $= 0.42$ at 160 epochs. Capping at $\sigma < 5$ reduces SSCD to $0.29$; capping at $\sigma < 3$ brings SSCD down to $0.22$; and capping at $\sigma < 1$ keeps SSCD near $0.10$ throughout training, see `fig10.md` for the full curves and `fig11.md` for qualitative samples. We note that full removal is used here as a controlled proof of concept; in practice, softer interventions are likely more appropriate. We also note that Jain et al. [3] arrive at a complementary inference-time memorization mitigation strategy: withholding text-conditioned guidance at high noise levels, which in our language corresponds to gapping the danger-zone $\sigma$ levels during sampling.
> > >
> > > [1] Wen et al. Detecting, Explaining, and Mitigating Memorization in Diffusion Models. ICLR, 2024.
> > >
> > > [2] Somepalli, et al. "Understanding and mitigating copying in diffusion models." NeurIPS 2023
> > >
> > > [3] Jain, et al. "Classifier-free guidance inside the attraction basin may cause memorization." CVPR, 2025.

---

### Official Review · Reviewer_s1nt · 2026-03-14

**Soundness:** 3
**Presentation:** 2
**Significance:** 2
**Originality:** 3
**Overall Recommendation:** 5
**Confidence:** 3

**Summary:**

In this paper, the authors study which noise level is the most crucial for memorization (generating training samples) of diffusion models. To study this, the authors compared three models: a learned memorizing model **EDM-1K**, a learned generalizing model **EDM-50K**, and an empirical memorized model **EMP-1K**.

They evaluate the contribution to memorization from different regimes by evaluating the distance of learned models to EMP-1K at different noise levels, and they find that at large noise levels, both learned models and **EMP-1K** are all similar and do not strongly memorize. They also conduct a swapping experiment to show whether memorizing or not depends on using **EDM-1K** or **EDM-50K** at the medium noise level.

The authors further develop their theory under a geometric view of EMP model. They show that in medium noise-level, the noisy training samples cover a larger range of the space (intersecting with more sampling trajectories), while maintaining a strong concentration around a training sample (dragging the trajectory to that sample). Together, overfitting at medium noise-level is the most important for memorization. They propose to undertrain medium noise levels when training data is scarce, and find that this can reduce memorization without degrading FID.

**Compliance With Llm Reviewing Policy:**

Affirmed.

**Final Justification:**

Please see the **Acknowledgement**. The authors have made decent progress in this paper, but there are also reasonable concerns regarding the practicality.

**Key Questions For Authors:**

1. The authors undertrain medium noise level to prevent memorization, but EDM [1] claims that such a regime is the most crucial for improving FID, so EDM emphasizes training at that regime. I am curious about the author's opinion on this. Also, does one have to manually configure the zone for different datasets?
2. Also, **EDM-1K** deviates from **EMP-1K** on both medium and small noise regimes in Fig.1. It seems more likely that in the small noise regime, the sampling has already stablized and it is hard to apply drastic changes with a smooth learned denoiser.  To fully isolate the effects of sampling and learning, I guess one useful experiment is to see what will happen if we just substitute the medium noise regime of **EDM-50K** with **EDM-1K** at small noise regime (though the inputs can be OoD for **EDM-1K**)?
3. Recent work [2] studies how nonlinear networks manage to approximate EMP solutions when they memorize, and how they deviate from EMP via learning data statistics and representations when they generalize, which can be related. [3] studies the sampling trajectory of EMP solution and can also be related.

[1] Karras, Tero, et al. *"Elucidating the design space of diffusion-based generative models."*, NeruIPS 2022

[2] Zhang, Zekai, et al. *"Generalization of Diffusion Models Arises with a Balanced Representation Space."*, ICLR'26

[3] Biroli, Giulio, et al. *"Dynamical regimes of diffusion models."*, Nature Communications 2024

**Limitations:**

Please see **Weakness**.

**Strengths And Weaknesses:**

### Strength:

1. The authors conduct controlled and comprehensive experiments to verify their claims. The proposed metrics on both trajectory and per-noise level memorization are interesting and novel.
2. They also study how to characterize the noise regime based on both learning (noisy training sample coverage) and sampling (extent of concentration towards one sample).

### Weakness:

1. The analysis does not fully disentangle the effect of network learning and sampling, and still centralizes on the property of EMP solutions. The arguments on networks’ implicit bias (e.g. locality bias) that shape their learning feel conceptual.
2. The theoretical analysis on the learned denoiser in small and large noise regime overalps a bit with prior results, and are mostly per-noise level (e.g. Thm. 4.2) rather than trajectory level. And while a further analysis at medium noise level (Sec. 4.2.3) is novel and important, it is missing. However, I appreciate the authors’ honesty and not over-claiming.

---

> ### Author Rebuttal · Authors · 2026-03-31
>
> We thank the reviewer for the constructive feedback. We address the main points below and will incorporate the suggested related work in the revision.
>
> Additional figures (figN.md) referenced below can be found in our anonymous repository: https://anonymous.4open.science/r/27904/.
>
> **On isolating learning and sampling, and on suggested experiment**
>
> We thank the reviewer for the suggestion to better isolate the roles of learning and sampling. To address this point, we performed the suggested experiment as well as two other controls.
>
> 1. We replaced the steps 7-13(medium noise) of EDM-50K by EDM-1K queried at a fixed small noise level $\sigma = 0.06$. The result is 0.0% memorization.
> 2. Under the same setup, we replace EDM-1K to replace the EMP-1K in (1). The memorization ratio becomes 100%.
> 3.  We swap the small noise region of EDM-50K with the same region of EMP-1K (complementing the swap experiment in Table 1 of our manuscript, where replacing this part with EDM-1K keeps memorization at 0%). This again yields 100% memorization.
>
> Taken together, these experiments show that the non-memorization effect of the small-noise part of EDM-1K is not explained simply by late-stage ODE stabilization. If stabilization were the full explanation, then small-noise EMP would also fail to induce memorization in the same swaps, which it does not. Instead, the difference comes from the denoiser itself: **the learned small-noise EDM-1K does not learn a look-up table like EMP does**. This supports our conclusion that memorization in the learned model is driven by behavior in the **medium-noise regime**. See Figure fig6.md for a qualitative visualization.
>
> **Clarification of theoretical contributions**
>
> We clarify the theoretical contributions: Prior work has observed near-Gaussian behavior at large noise (e.g., [1]), but Thm 4.11 provides, to our knowledge, the first theoretical explanation of this phenomenon. Similarly, while disjoint Gaussian shells at small noise have appeared in prior discussions (e.g., [2]), our Thm 4.9, together with our experiments, shows that shell disjointness alone does not imply memorization.
> Then, we emphasize that our work introduces a crucial new dimension: a per-noise-level decomposition of memorization risk. This reveals that memorization is governed by two competing fixed-$\sigma$ quantities—posterior weight concentration and Gaussian shell coverage—whose coexistence is uniquely maximized in an intermediate band, which we identify as the "danger zone." This is why a purely trajectory-level view obscures the mechanism. Our main contribution is precisely to separate three regimes and isolate the intermediate one as the key regime that most needs to be understood. While a sharp theoretical characterization of the intermediate regime remains an important direction for future work, its identification through our geometric insight is central to our paper.
>
> **Regarding the question on how to reconcile the EDM noise schedule with our gap method**
>
> We do not view our findings as being in conflict with EDM. Rather, our results suggest that the same intermediate-noise regime is crucial for both generalization and memorization. We partially validate this already in the manuscript: Fig. 2 shows that memorization is most sensitive to training in the medium-noise range, and the swapping experiment in Fig. 10 shows that changing this regime has the strongest effect on downstream trajectory behavior. Which effect, memorization or generalization, dominates appears to depend on data size and model capacity. In small-data settings such as 1K CIFAR-10, each example is revisited much more often under a fixed training budget, making over-reinforcement and memorization in this regime more likely; for larger datasets, the same regime can support useful shared-structure learning, although sparse or weakly populated regions may still remain vulnerable to memorization. This poses a great direction for future works. We also remark that our gap training serves only as a controlled proof of concept for the causal role of this regime, not as a claim that fully removing it is the optimal practical strategy. In practice, softer and more data-dependent interventions—such as reweighting or adaptive scheduling within the danger zone—are likely more appropriate.
> Finally, regarding configuration, the relevant interval is inherently dataset-dependent and should be estimated for each setting. We view this not as a drawback, but as an important practical implication of our analysis: the danger zone depends on data geometry and training conditions, and therefore should not be treated as a fixed range of $t$ or $\sigma$ across datasets. We also remark that even in EDM, the preconditioning is dependent on data variance.
>
> [1] Li et al.  Understanding Generalizability of Diffusion Models Requires Rethinking the Hidden Gaussian Structure. NeurIPS, 2024.
>
> [2] Song et al. Selective Underfitting in Diffusion Models. arXiv:2510.01378, 2025.

---

> > ### Author Rebuttal · Reviewer_s1nt · 2026-04-02
> >
> > I thank the authors for their great effort in conducting the additional experiments. Those indeed show that the learned **EDM-1k** does not learn the lookup table. But for the theory, it seems Thm. 4.11 (empirical score approximates Gaussian score in high noise regime) overalps a bit with the results in Sec. 2.1 of [1]; and there can be some more in-depth argument of inductive bias that shapes the learning of **EDM-1k** in low-noise regime. And I also partly agree with the other reviewers ' concerns about the effectiveness in real-world scenarios.
> >
> > But overall, I still think the authors have made decent progress in this paper and the rebuttal. I have increased my score accordingly.
> >
> > [1] Wang, Binxu, and John J. Vastola. *"The unreasonable effectiveness of gaussian score approximation for diffusion models and its applications."* arXiv 2024.

---

> > > ### Author Response · Authors · 2026-04-06
> > >
> > > We thank the reviewer for engaging in the discussion and pointing us to Wang and Vastola [1]. Their Sec. 4.1 shows that for point-cloud data, the empirical optimal denoiser admits a leading-order high-noise approximation by a Gaussian object with matching first two moments. Our Thm. 4.11 is closely related, but broader: it establishes the same Gaussian approximation principle for any bounded-support distribution, including both the empirical distribution and the ground-truth distribution, and does so with an explicit decay rate for the residual term. As a consequence of our broader result, the empirical optimum can be shown to already be close to the ground-truth target in the high-noise regime, which implies memorization is not driven by high noise in general. We will cite the reference and include the discussion in our revision.
> > >
> > > On the practical side, to partially address the broader effectiveness question raised by the other reviewers, we also tested SD v1.4 finetuning on 200 memorized image-prompt pairs and found that the noise gap method mitigates memorization; see detailed responses to Reviewers `F4Wy` and `vxF3`.
> > >
> > > [1] Wang, Binxu, and John J. Vastola. "The unreasonable effectiveness of Gaussian score approximation for diffusion models and its applications." arXiv 2024.

---

### Official Review · Reviewer_F4Wy · 2026-03-15

**Soundness:** 2
**Presentation:** 3
**Significance:** 2
**Originality:** 3
**Overall Recommendation:** 3
**Confidence:** 3

**Summary:**

This paper introduces a geometric framework based on posterior weight concentration and Gaussian shell coverage to understand when diffusion models memorize training data. The authors reveal that memorization risk is not uniform across the schedule, but instead peaks within a specific "danger zone" at intermediate noise levels, while small and large noise regimes resist it. Leveraging this insight, they propose a targeted intervention that selectively skips training on these medium noise levels, which substantially mitigates memorization while preserving generation quality in the extremely simple setup.

**Compliance With Llm Reviewing Policy:**

Affirmed.

**Final Justification:**

Rebuttal does not adequately address my concerns, as my main reservations were against evaluating the memorization only in extremely small data regime. Fine-tuning even larger model like SD with as little as 200 image-prompt pairs fills into exactly the same category. I am still concerned about scalability of this framework to real use-cases therefore I will maintain my original recommendation

**Key Questions For Authors:**

None

**Limitations:**

Limitations are not discussed

**Strengths And Weaknesses:**

Strengths
- The literature on memorization in diffusion models is currently flooded with heuristic-based mitigation strategies. It is highly refreshing to see a submission that steps back to conduct a thorough, principled analysis of the underlying mechanisms, offering a deep geometric understanding rather than just another ad-hoc fix.
- The authors introduce a highly original and intuitive framework for analyzing memorization by examining the interplay between two specific geometric forces: posterior weight concentration and Gaussian shell coverage. By mapping the intersection of these two metrics, the paper provides a compelling perspective that successfully isolates the intermediate noise regime as the specific "danger zone" where memorization actually forms.
- The introductory experiment in Section 3 serves as an excellent foundation, effectively grounding the paper and clearly motivating the need to analyze memorization at a per-noise-level granularity.


Weaknesses
- While the theoretical part of this work is well performed, the empirical evaluation is surprisingly limited in scope. To demonstrate their "gap training" intervention, the authors evaluate their method solely on extremely constrained, low-data regimes - specifically a 2k-image CIFAR-10 car subset and 1024 downscaled, grayscale CelebA images. While it is understandable that the authors must evaluate their method in a regime where models actually memorize, forcing this behavior through artificial data scarcity raises a critical concern about the underlying mechanisms being studied. In these highly artificial setups, memorization is likely a trivial consequence of extreme data sparsity; the model simply lacks enough data to interpolate a continuous manifold, leading to naturally disjoint Gaussian shells at small noise levels. However, memorization in state-of-the-art, large-scale diffusion models occurs despite training on millions or billions of images, often driven by entirely different mechanisms. Because the authors solely rely on setups where memorization is forced through an artificially tiny sample size, it remains unknown whether their geometric framework (and the resulting "gap training" intervention) actually targets the mechanisms that cause memorization in realistic, data-rich generative models.
- The authors claim that "per-noise-level memorization... contributes most to trajectory-level memorization". However, the experiments used to bridge these two concepts leave a logical gap. The authors rely on a denoiser swapping experiment to prove that the medium-noise regime is responsible for global memorization. However, this experiment swaps a massive range of noise levels  [0.14, 8.4]. As a result, it only proves that this sequence of steps is critical for global memorization; it does not prove that single-step memorization forces the trajectory to collapse. While Figure 2 does show a spike in "Per-Noise-Level" memorization, it only peaks at ~26%. The authors do not adequately explain the mechanics of how a ~26% chance of memorization at individual isolated steps compounds into a 92.2% trajectory-level memorization rate.
- While Theorem 4.2 provides neat, explicit finite-sample bounds for posterior weight concentration based on nearest-neighbor distances, the fundamental behavior it describes is somewhat well-trodden ground. The phenomenon of empirical score weights collapsing onto a single training point at small noise scales has been extensively studied in recent works, such as the 'collapse regime' formalized via statistical mechanics by Biroli et al. (2024)[1]. The authors should more clearly acknowledge that Theorem 4.2 serves primarily as a finite-sample geometric formalization of this already-established asymptotic behavior.
[1] Biroli, Giulio, et al. "Dynamical regimes of diffusion models." Nature Communications 15.1 (2024): 9957.

Small issues:

[Presentation] Tables are figures are placed far from their references (e.g. Figure 1 (page 2) with reference in page 3, Figure 2 and Table 1 (page 3) with references in page 4 and so on - it makes it hard to read while jumping between pages

---

> ### Author Rebuttal · Authors · 2026-03-31
>
> We thank the reviewer for the insightful feedback and focus below on the major points.
>
> Additional figures (figN.md) referenced below can be found in our anonymous repository: https://anonymous.4open.science/r/27904/.
>
> **Regarding the question on the gap between 26.2% and 92.2%**
>
> We would like to clarify that the 26.2% and 92.2% memorization rates measure different quantities. The 26.2% is a one-step memorization ratio for the denoiser output of random noisy test images, whereas the 92.2% is for the final ODE trajectory outcome, where multiple denoising steps are applied sequentially. These one-step events are therefore not independent: once a medium-noise step moves a sample closer to a particular training image, the next iterate is no longer a generic noisy input, but a state with increased posterior concentration around that same training image. This creates a compounding effect across successive medium-noise steps.
> We now verify this directly with an on-trajectory analysis. Figure fig1.md reports memorization after successive Euler steps on the actual trajectory, and shows that just two medium-noise steps (steps 7 and 8) already raise the rate from 26.2% to 58%. Over the full trajectory starting from $\sigma=80$, the rate then rises from 31% at $\sigma=5.3$ to 60% at $\sigma=3$ and 80% at $\sigma=2$, before reaching the final 92.2% endpoint.
> We also note that the standard threshold $d_{\mathrm{1NN}}/d_{\mathrm{2NN}} < 1/3$ for computing the memorization rate is quite conservative: the continuous ratio (Figure fig2.md) shows the same progressive concentration, and relaxing the threshold to $2/3$ raises the intermediate-regime rate above 50%. The $\ell_2$ distance also does not fully capture perceptual similarity, which we leave to future work.
>
> **Clarification on Thm 4.2 and related work**
>
> We thank the reviewer for this helpful comment and agree that Thm 4.2's relation to the broader literature on diffusion-model memorization and low-noise collapse warrants clearer discussion. Crucially, Thm 4.2 provides explicit, high-probability, data-dependent bounds for posterior concentration at a fixed noise level, with quantities directly computable from the observed dataset. This geometric formalization offers a new dimension for analyzing memorization, distinct from established asymptotic behaviors. We will revise the paper to make this unique contribution and its relation to prior work clearer around Thm 4.2.
>
> **Regarding the reviewer’s concern on the scope of our empirical evaluation**
>
> We first clarify that we do not claim that memorization is created at small noise. Our picture is that memorization is formed in the intermediate-noise regime and then largely preserved by later low-noise steps.
> We agree with the reviewer that increasing dataset size can improve manifold recovery, but this effect is expected to matter primarily in the intermediate regime. In the small-noise regime, the most important quantity from sampling, the nearest-neighbor distances, shrink only as $O(N^{-1/k})$ in intrinsic dimension $k$ and data size $N$ (see [P1998]). This results in a quite persistent range for small noise across dataset size. Empirically, our CIFAR size-sweep (Figure fig3.md) shows that the small noise region remains relatively stable from $1$K to $50$K. We also plot 1-NN distance distributions for CIFAR-10 ($1$K--$50$K) and ImageNet latents (up to $1000$K), which remain broadly similar across $N$ (Figures fig4.md, fig5.md).
> For large datasets, we agree that trajectory-level memorization may be harder to observe as cleanly as in the small-data settings. However, this does not mean memorization disappears. Real datasets are highly non-uniform, so a model may still memorize sparse regions while generalizing well elsewhere. This is especially relevant in text-to-image models, where memorization is often associated with prompts or concepts supported by very few training images, sometimes even a single one [W2024]. Notably, Figure 1 of MemBench ([Hong2024]) shows that many of the known memorized prompts in Stable Diffusion v1.4 are about commercial product listings. We hypothesize that overly detailed listing information might create local sparse regions. We view extending our theory to such locally sparse regimes, and testing it in settings such as text-to-image models, as an important direction for future work.
>
> [P1998] Percus, Martin. Scaling universalities of kth-nearest neighbor distances on closed manifolds. Advances in Applied Mathematics, 21, 424-436, 1998.
>
> [W2024] Wen et al. Detecting, Explaining, and Mitigating Memorization in Diffusion Models. ICLR, 2024.
>
> [Hong2024] Hong, Chunsan, Tae-Hyun Oh, and Minhyuk Sung. "Membench: Memorized image trigger prompt dataset for diffusion models." arXiv 2024.

---

> > ### Author Rebuttal · Reviewer_F4Wy · 2026-04-02
> >
> > I thank the authors for their detailed rebuttal
> >
> > However, I agree with other reviewers (vxF3 and ohzn) that the reliance on small, artificially constrained datasets remains a significant limitation. I appreciate your theoretical argument that large-scale models (like Stable Diffusion) contain "locally sparse regions" corresponding to rare concepts, which might trigger the same memorization mechanics. This is a very sound and interesting hypothesis that requires validation.
> >
> > The new experiment on ImageNet does not empirically resolve this concern. Training a small model on 1300 ImageNet goldfish is essentially the same constrained, low-data regime as the 2k CIFAR-10 cars or 1024 CelebA images. While the theoretical connection to local sparsity in large datasets is logical, the paper still lacks an empirical demonstration of this framework scaling to truly data-rich, state-of-the-art training pipelines where memorization happens despite millions of data points.
> >
> > I will maintin my current score

---

> > > ### Author Response · Authors · 2026-04-06
> > >
> > > **Regarding the question of transfer beyond small-scale memorization-prone settings**
> > >
> > > We thank the reviewer for the engaging discussion. We emphasize that our controlled experiments are designed to validate the main methodological contribution: that coverage and max posterior weight jointly identify the noise levels responsible for memorization. The settings in the paper provided a clean testbed and supported our claims.  We now provide preliminary evidence that this framework extends to large-scale text-to-image model finetuning. Additional figures (figN.md) referenced below can be found in our anonymous repository: https://anonymous.4open.science/r/27904/.
> > >
> > > Finetuning Stable Diffusion on a small number of image-prompt pairs can induce memorization (see e.g. [1]). We finetune SD v1.4 on 200 memorized image-prompt pairs for 160 epochs; the baseline achieves a mean SSCD similarity of 0.42, indicating memorization [2]. We analyze this setting through the lens of our coverage-weight framework. It turned out that in this dataset, each prompt maps to exactly one training image. In such a case, the conditional posterior places all its mass on that image: $W_\sigma = 1$ for every noise level $\sigma$. This eliminates the large-$\sigma$ protection that exists in the multi-image unconditional case, where the posterior becomes diffuse at high noise. Coverage, on the other hand, still depends on the geometry of the data distribution. To approximate the coverage transition, we encode 50K LAION-Aesthetic images through the SD VAE and compute when the Gaussian shell around a single training point begins to overlap with the broader population. This unconditional estimate places the onset of non-trivial coverage around $\sigma \approx 3$, see `Fig9.md`. (A tighter analysis would condition on each prompt's text embedding, which we leave to future work.)
> > >
> > > In our framework, the danger zone is the noise region where both $W_\sigma$ and coverage are simultaneously high. Here $W_\sigma = 1$ everywhere, so the danger zone is determined entirely by where coverage becomes non-trivial, yielding a danger zone of the form $[\sigma_{\mathrm{gap}}, \infty)$ with $\sigma_{\mathrm{gap}} \approx 3$. The natural intervention is therefore to cap the training noise schedule at $\sigma_{\max} \leq \sigma_{\mathrm{gap}}$, preventing the model from training on the memorization-prone high-coverage region. The $\sigma$-capping experiments provide strong validation. The baseline finetuning (no capping) yields mean SSCD $= 0.42$ at 160 epochs. Capping at $\sigma < 5$ reduces SSCD to $0.29$; capping at $\sigma < 3$ brings SSCD down to $0.22$; and capping at $\sigma < 1$ keeps SSCD near $0.10$ throughout training, see `Fig10.md` for the full curves and `Fig11.md` for qualitative samples. The fact of low SSCD score when the danger zone is eliminated aligns with our framework.  We also note that Jain et al. [3] arrive at a complementary inference-time memorization mitigation strategy: withholding text-conditioned guidance at high noise levels, which in our language corresponds to gapping the danger-zone $\sigma$ levels during sampling.
> > >
> > > [1] Wen et al. Detecting, Explaining, and Mitigating Memorization in Diffusion Models. ICLR, 2024.
> > >
> > > [2] Somepalli, et al. "Understanding and mitigating copying in diffusion models." NeurIPS 2023
> > >
> > > [3] Jain, et al. "Classifier-free guidance inside the attraction basin may cause memorization." CVPR, 2025.

---

### Decision · Program_Chairs · 2026-04-30

**Decision:**

Accept (regular)

**Comment:**

This paper introduces a principled geometric framework, utilizing posterior weight concentration and Gaussian shell coverage, to analyze memorization in diffusion models. By identifying an intermediate noise "danger zone" that primarily drives memorization, the authors offer an original, per-noise-level perspective.

While reviewers initially questioned the reliance on artificially constrained low-data regimes and the practical scalability of the "gap training" mitigation strategy, the authors provided a robust rebuttal featuring large-scale text-to-image experiments (Stable Diffusion) that successfully validated their claims. Although refining these mitigation strategies for production-scale models remains future work, the core theoretical insight is strong and offers a valuable new lens for understanding and addressing memorization.

The paper is technically sound, insightful, and recommended for acceptance.